# Trust in scientists and their role in society across 68 countries

Science is crucial for evidence-based decision-making. Public trust in scientists can help decision makers act on the basis of the best available evidence, especially during crises. However, in recent years the epistemic authority of science has been challenged, causing concerns about low public trust in scientists. We interrogated these concerns with a preregistered 68-country survey of 71,922 respondents and found that in most countries, most people trust scientists and agree that scientists should engage more in society and policymaking. We found variations between and within countries, which we explain with individual- and country-level variables, including political orientation. While there is no widespread lack of trust in scientists, we cannot discount the concern that lack of trust in scientists by even a small minority may affect considerations of scientific evidence in policymaking. These findings have implications for scientists and policymakers seeking to maintain and increase trust in scientists.

Public trust in science provides many benefits to society. It helps people make informed decisions (for example, on health and nutrition) on the basis of the best available evidence, provides the foundation for evidence-based policymaking and facilitates government spending on research. Trust in science and scientists enables the management of global crises such as the COVID-19 pandemic and climate change. Societies with high public trust in science and scientists dealt with the COVID-19 pandemic more effectively, as citizens were more likely to comply with non-pharmaceutical COVID-19 interventions[1] and had higher vaccine confidence[2]. People with high trust in scientists are also more likely to engage in individual and collective action on climate change[3,4].

Studies find that most people trust science, and scientists are among the most trusted actors in society[5–7]. Despite these findings, there is a popular dominant narrative claiming that there is a crisis of trust in science and scientists[8,9]. This narrative well predates the COVID-19 pandemic and may alter people's views about scientists[10–12]. It is therefore important to revisit this narrative and provide robust empirical evidence on whether it is accurate.

Most previous studies have been limited to the Global North, typically the USA or Europe, including our own previous work (see, for example, refs. 13–19). A few studies have gone beyond these regions[5–7,20–23]. However, they assess a limited range of theoretical constructs. We address this limitation in two ways. First, we analyse the extent to which people believe that scientists should be involved in society and policymaking. We refer to this as 'normative perceptions of science in society and policymaking'. Second, we investigate which issues people want scientists to prioritize in their work and how such perceptions are related to their trust in scientists. Previous studies have shown that trust is affected by the perception of value alignment[24]. People who feel that their concerns and values are not reflected in the priorities of scientists may therefore doubt the trustworthiness of scientists.

Our large-scale, preregistered survey expands and strengthens previous work by offering a comprehensive dataset on trust in scientists after the COVID-19 pandemic[16] and by investigating the public's normative perceptions of the role of scientists in society and policymaking and their desired research priorities. We use a theoretically informed multidimensional trust measure[25] and examine relevant demographic, ideological, attitudinal and country-level factors to explain trust across countries[6]. We survey countries and individuals that are underrepresented in research[26], and, in almost all countries, we have worked with local research partners[27,28].

Our study answers the following questions. (1) How much do people around the world trust scientists, and how do levels of trust vary across countries? (2) How do demographic, ideological, attitudinal and

✉ e-mail: viktoriacologna@gmail.com

country-level factors relate to trust in scientists (see Supplementary Fig. 1a for a directed acyclic graph), and how do these relationships vary between countries? (3) What are people's normative perceptions of scientists in society and policymaking, and how do they differ across countries? (4) What issues do people want scientists to prioritize, and do they believe that scientists actually address these priorities? See the preregistration for more detailed research questions and hypotheses (https://osf.io/9ksrj/).

By investigating trust in scientists, we do not mean to imply that trust is always warranted. In some situations, low trust may be warranted. For example, science's fraught historical relationship with racism, its role in perpetuating racialized forms of knowledge production, sustaining racial paradigms[29] and disregarding ethical canons by experimenting on non-white human subjects[30], has reduced research participation in some populations[31]. Furthermore, the epistemic authority of science and scientists has been challenged by misinformation and disinformation[32,33], a "reproducibility crisis"[34], conspiracy theories[35,36] and science-related populist attitudes[37,38]. Science-related populism has been conceptualized as a perceived antagonism between 'the ordinary people' and common sense on one side and academic elites and scientific expertise on the other[37]. Unlike political populism, which criticizes political elites and their political power claims, science-related populism criticizes academic elites, challenges their decision-making authority in scientific research and suggests that their epistemic truth claims are inferior to the common sense of 'the people'[37]. Anti-science attitudes, even if held by only a minority of people, raise concerns about a potential crisis of trust in science, which could challenge the epistemic authority of science and the role of scientists in supporting evidence-based policymaking[20,37]. These concerns, which have been prominently discussed in leading news media, have been exacerbated as trust in scientists and their desired role in policymaking have become divided along partisan lines. Several studies show that in the USA and some other countries, conservatives and right-leaning individuals have low levels of trust in scientists, hold stronger anti-science attitudes and express low confidence that scientists act in the best interest of the public, provide benefits to society and apply reliable methods[19–21,39,40]. Empirical evidence is needed to determine how widespread such critical attitudes towards science are across countries and population groups.

Our survey goes beyond commonly studied correlates of trust in scientists in four important ways. First, we investigate how trust in scientists relates to science-related populist attitudes. Science-related populists deny that scientists are knowledgeable experts and believe that they do not act in the interest of the general public—two key aspects of trust in scientists[37]. Second, we investigate whether trust in scientists is related to people's social dominance orientation (SDO), which has been defined as "the degree to which individuals desire and support group-based hierarchy and the domination of 'inferior' groups by 'superior' groups"[41] (p. 48). Individuals high in SDO are arguably less likely to trust scientists, as they perceive universities as hierarchy-attenuating social institutions[42]. Previous research supports this, showing that high SDO is a predictor of low trust in scientists[43] and distrust in climate science[19,43]. However, it is unknown how SDO relates to trust in scientists across many countries. Third, we investigate what goals people want scientists to prioritize in their work and how this relates to trust. Fourth, we investigate whether people perceive that their desired priorities are tackled by science.

To answer our research questions, we conducted a crowdsourced Many Labs project with the same translated online questionnaire given to 71,922 respondents in 68 countries on all inhabited continents (Supplementary Fig. 2). The term 'country' in this Article refers to both sovereign states and territories not recognized as such. The survey covered 31% of the world's countries, which together make up 79% of the global population. The data were collected between November 2022 and August 2023, with quota samples that were weighted according

to national distributions of age, gender and education level, as well as country sample size. As recommended by other studies on trust in scientists[6], we provided the respondents with a definition of science and scientists to mitigate semantic variations across languages (Supplementary Information). We measured trust in scientists (instead of science) because 'science' is more abstract than 'scientists' and therefore makes a less clear referent: people may think of scientific institutions, scientific communities, scientific methods or individual scientists when being asked about their general perception of 'science'. However, these trust measures can be distinguished both conceptually and empirically[25,44]. For example, research has shown that less educated people trust scientific methods more than scientific institutions[44]. General measures that assess trust in the scientific community capture only some of the conceptually established dimensions of perceived trustworthiness (for example, expertise)[25]. We reduced this ambiguity by avoiding the abstract category 'science' and using the more concrete reference object 'scientists'[6]. We slightly deviated from the preregistration. We collapsed sparsely populated neighbouring strata for post hoc weighting, excluded confidence in science as a model covariate because of multicollinearity and included SDO as a covariate in the regression model testing predictors of normative perceptions of the role of science in society and politics (Supplementary Information). All analyses can be reproduced with the replication materials available at https://osf.io/wj34h/.

## Results
### Trust in scientists across the world
We employed an index composed from a 12-item scale measuring four established dimensions of trustworthiness: perceived competence, benevolence, integrity and openness[13,25,45]. This scale is based on a comprehensive review of trust measures used to assess trustworthiness perceptions of scientists[45]. It was pretested to confirm its reliability, relies on accepted conceptual assumptions that we validated in factor analyses and has high reliability across countries[46]. However, confirmatory factor analyses show that we can assume only configural invariance and no metric or scalar invariance[46]. This is a common caveat of multilingual survey research and is to some extent unavoidable[47]. When these components of trustworthiness perceptions are aggregated to a single score, the index represents an integrative measure of public trust in scientists with strong reliability (Cronbach's $\alpha = 0.93$ and $\omega = 0.95$). We therefore used the aggregate index for our main analyses (see Supplementary Information for additional analyses with individual trust dimensions).

Overall, trust in scientists is moderately high (grand mean, 3.62; s.d., 0.70; 1 = very low, 2 = somewhat low, 3 = neither high nor low, 4 = somewhat high, 5 = very high). No country shows low overall trust in scientists (Fig. 1). Across the globe, people perceive scientists as having high competence (mean, 4.02; s.d., 0.71), with 78% believing that scientists are qualified to conduct high-impact research (5% believe they are unqualified, and 16% selected the scale midpoint; Supplementary Fig. 3). People perceive scientists to have moderate integrity (mean, 3.58; s.d., 0.78) and benevolent intentions (mean, 3.55; s.d., 0.82; Supplementary Table 1). For example, 57% of people believe that most scientists are honest (11% believe they are dishonest, and 31% selected the scale midpoint), and 56% believe that most scientists are concerned about people's well-being (15% believe they are not concerned, and 29% selected the scale midpoint). Scientists' perceived openness to feedback is slightly lower (mean, 3.33; s.d., 0.86), with 42% believing that scientists pay attention to others' views somewhat or very much. Overall, 75% agree that scientific research methods are the best way to find out whether something is true or false. Trust in scientific methods moderately correlates with trust in scientists ($r_{69,516} = 0.473$; $P < 0.001$; $t = 128.45$; 95% confidence interval (CI), 0.468 to 0.478), supporting previous findings on the multidimensionality of trust in science[44].

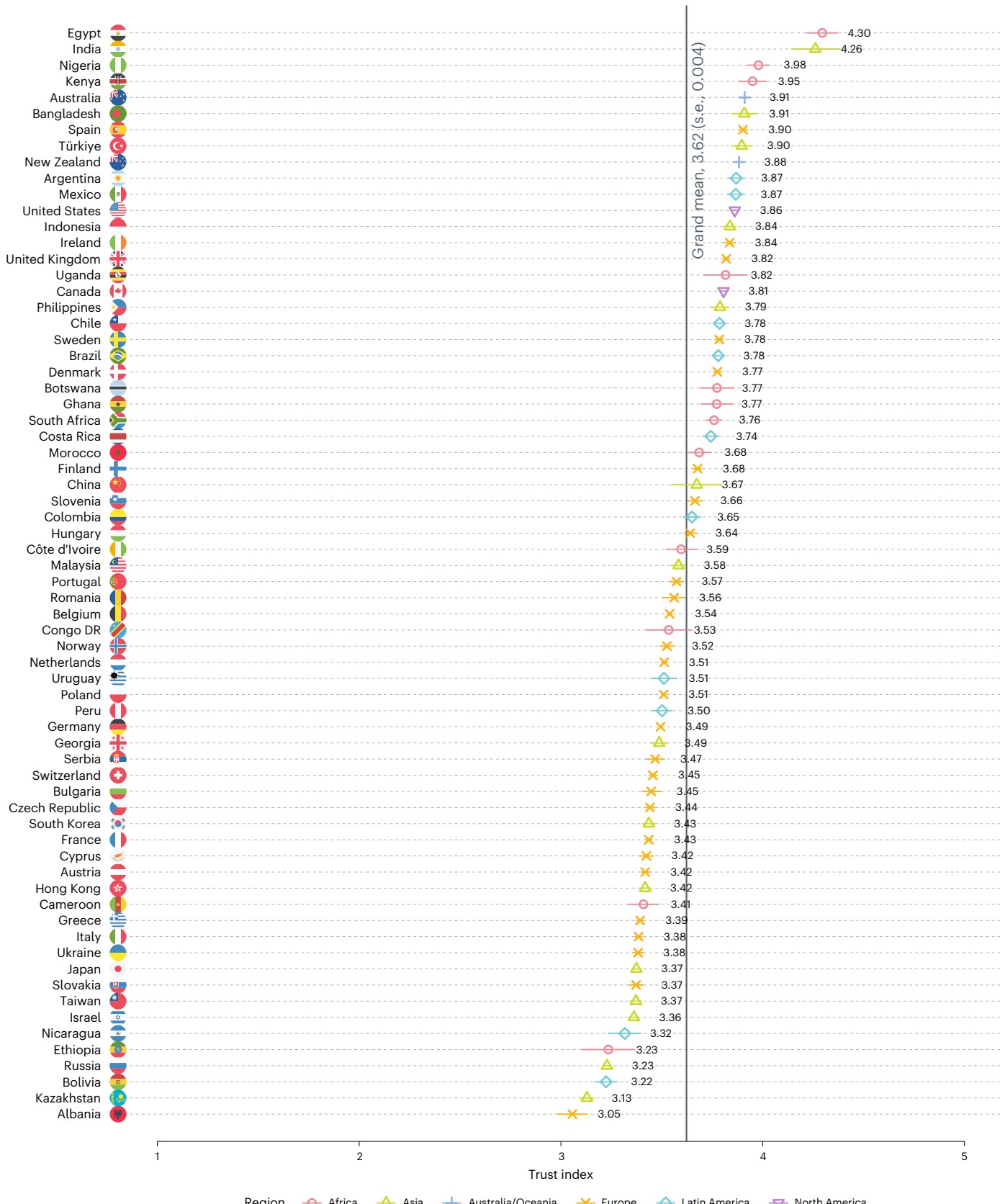

**Fig. 1 | Weighted means for trust in scientists across countries and regions (1 = very low, 3 = neither high nor low, 5 = very high).** Total $n$ = 69,527. Country $n$s range between 312 and 8,014 (see Supplementary Information for a detailed overview). The vertical line denotes the weighted grand mean. The horizontal lines indicate means ± standard errors. Country-level standard errors range between 0.008 and 0.133.

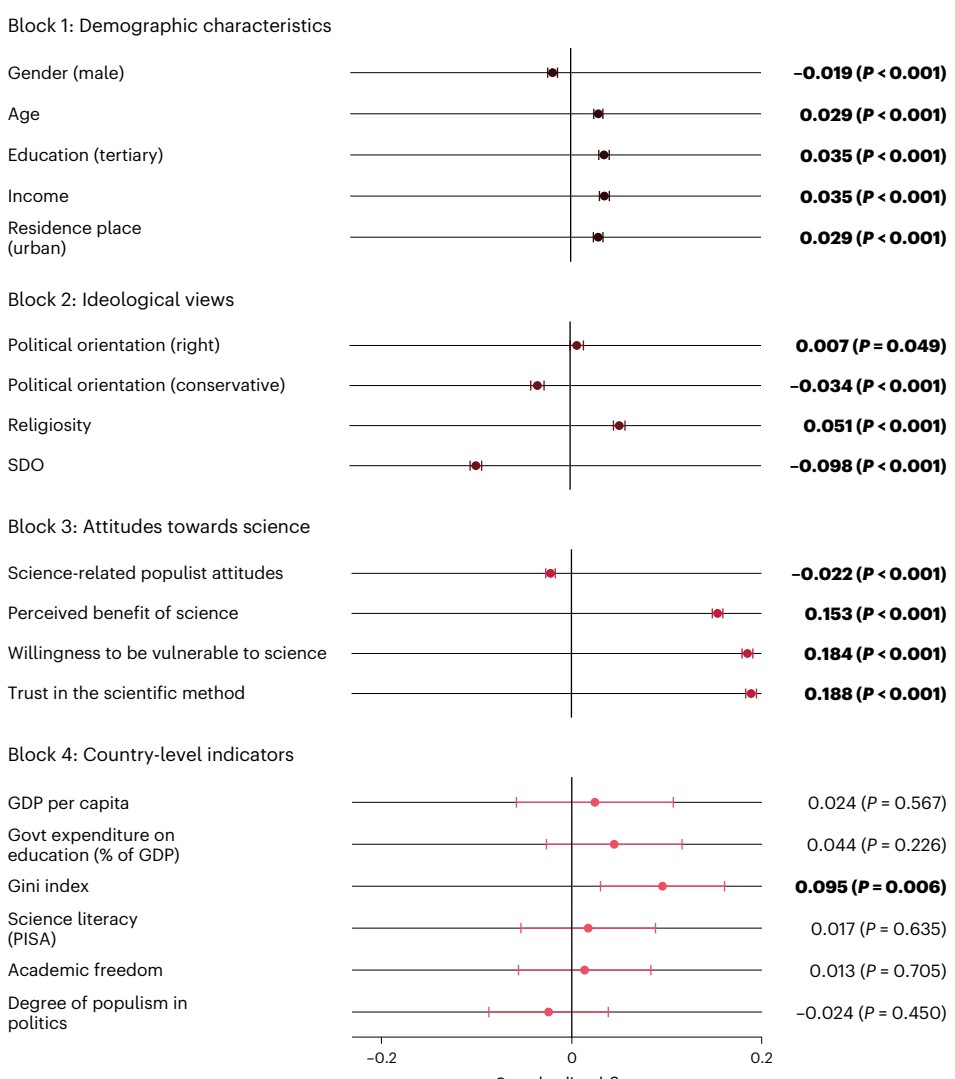

**Fig. 2 | Standardized estimates of weighted blockwise multilevel regression models testing the association of trust in scientists with demographic characteristics, ideological views, attitudes towards science and country-level indicators (random intercepts across countries).** The dots indicate point estimates of fixed effects, and the horizontal lines indicate 95% CIs based on two-sided *t*-tests. Estimates for gender (male) indicate the association of identifying as male and trust in scientists, where 0 = female and 1 = male. Estimates for education (tertiary) indicate the association of having tertiary education and trust in scientists, where 0 = no tertiary education and 1 = tertiary education. Estimates for residence place (urban) indicate the association of living in an urban vs rural place of residence, where 0 = rural and 1 = urban. Estimates for political orientation (right) indicate the association of right-leaning vs left-leaning political orientation and trust in scientists, where 1 = strongly left-leaning and 5 = strongly right-leaning. Estimates for political orientation (conservative) indicate the association of conservative vs liberal political orientation and trust in scientists, where 1 = strongly liberal and 5 = strongly conservative. Bold indicates effects significant at *P* < 0.05. Block 1 uses data from all 68 countries, block 2 uses data from 67 countries (all except Malaysia, where SDO was not measured), block 3 uses data from 66 countries (all except Malaysia and Mexico, where willingness to be vulnerable to science was not measured) and block 4 uses data from 51 countries (all except those where PISA's literacy scores were not available; Supplementary Information). The full regression results are reported in Supplementary Table 2. The results of exploratory analyses with individual trust dimensions are reported in Supplementary Figs. 4–7. GDP, gross domestic product; Govt, government.

While trust in science is moderately high overall, there are notable variations across countries and regions (Fig. 1). Contrary to previous studies[6,7], we did not find a clear pattern that scientists are less trusted in Latin American and African countries. However, we did find patterns within specific regions. For example, Russia as well as several former Soviet republics and satellite states (such as Kazakhstan) show relatively low trust in scientists.

## Correlates of trust in scientists
To identify correlates of trust in scientists, we fitted linear random-intercept regression models that included post-stratification weights to provide estimates that are nationally representative in terms of gender, age and education in almost all countries. To investigate how trust in scientists differs across population groups, we assessed several demographic variables and analysed their correlation with trust in scientists. We found higher levels of trust among many demographic groups: women, older people, residents of urban (versus rural) regions, people with high incomes, religious people, educated people, liberal people and left-leaning people (Fig. 2; see also Supplementary Table 2). Differences across countries and sociodemographic groups can be explored with an online data visualization tool developed for this project: https://tisp.shinyapps.io/TISP/.

One might assume that trust in science would correlate with tertiary education, as people with more years of schooling and university

education have had more chances to build a closer relationship with science and experience the competence and benevolence of scientists, for example[48]. However, our data show only a small positive relationship between tertiary education and trust in scientists on average ($\beta_{63,979}$ = 0.035; $P$ < 0.001; $t$ = 12.56; 95% CI, 0.029 to 0.040). In fact, in most countries we found little or no credible evidence for a relationship between tertiary education and trust (Supplementary Fig. 8). Overall, the relationships between demographic characteristics and trust in scientists are very small (marginal effects plots with unstandardized units are shown in Supplementary Fig. 9).

Many might also assume that religiosity is associated with lower trust in scientists, given that many studies conducted in Global North countries have found this relationship (see, for example, refs. 19,49). However, against this assumption, one previous study found that only 29% of people worldwide believe that science stands in disagreement with their religion[6]. Another study found that while religiosity is associated with negative attitudes towards science in the USA, the relationship is inconsistent across the world[50]. Indeed, we found that, overall, religiosity is positively associated with trust in scientists ($\beta_{47,597}$ = 0.051; $P$ < 0.001; $t$ = 16.68; 95% CI, 0.045 to 0.057). However, as previous studies have also shown, we found substantive differences across countries and regions[50,51]. In Muslim countries such as Türkiye, Bangladesh and Malaysia (Supplementary Fig. 10), trust is positively associated with religiosity. Qualitative interviews conducted by the Pew Research Center put these findings into context[52]. They found that most Muslim participants did not perceive a conflict between science and religion, because their holy text, the Quran, proclaims many principles of science. Conversely, some Christians perceive that science disagrees with their religion, even though there are pronounced variations across countries[52]. Our findings are consistent with these results.

Other positive correlates of trust in scientists include people's willingness to rely on scientific advice and thus make themselves vulnerable to scientists, the belief that science benefits people like them, and trust in scientific methods.

Our study also sheds light on individual attributes that are associated with lower trust in scientists—namely, conservative political orientation, higher SDO and science-populist attitudes. Previous studies, which mostly focused on North America and Europe, have found right-leaning and conservative political orientation to be negatively associated with trust in scientists[19,20]. Our study partly confirms these findings. We found a negative association between trust and conservative political orientation. However, we found a very small, positive relationship between right-leaning political orientation and trust. Given that some recent global social science studies used a left–right measure to assess political orientation while others used a liberal–conservative measure[53–55], we used both measures and analysed how the results vary depending on the measure in question. We found that the relationships between the two measures of political orientation and trust vary substantially across countries (Fig. 3a,b and Supplementary Figs. 11 and 12). For example, in the USA, trust is associated with a liberal orientation but not with one's self-placement on the left–right spectrum. More generally, right-leaning and conservative political orientation are negatively associated with trust in scientists in several European and North American countries, so previous research, which has disproportionally focused on these countries, has tended to stress right-leaning and conservative distrust. However, in most countries ($k$ = 41 for the left–right measure and $k$ = 48 for the liberal–conservative measure), our data do not show credible evidence of a relationship between political orientation and trust in scientists. Furthermore, in some Eastern European, Southeast Asian and African countries, right-leaning individuals have higher trust in scientists. These contrasting findings may be explained by the fact that in some countries right-leaning parties may have cultivated reservations against scientists among their supporters, while in other countries left-leaning parties may have done so[56] (Supplementary Fig. 11). In other words, the attitudes of political

leadership rather than peoples' political orientation may better explain politically correlated attitudes towards scientists (see Supplementary Information for selected country-specific explanations). We encourage future research to investigate differences in the two measures of political orientation on the country level (for broader discussions on these measures, see refs. 57–60).

Some studies have looked at SDO—that is to say, a preference for social hierarchy and inequality—and found it to be negatively associated with trust in scientists[19,43]. Our results confirm this: the low grand mean for SDO (mean, 3.62; s.d., 1.76; 1 = extremely oppose to 10 = extremely favour) is consistent with the overall moderately high trust in scientists. Moreover, we found that those who favour hierarchy enhancement (that is, more strongly endorse SDO) are less likely to trust scientists ($\beta_{47,602}$ = −0.098; $P$ < 0.001; $t$ = −31.98; 95% CI, −0.104 to −0.092). This may be because they see universities (that is, scientists) as institutions that weaken social hierarchies[42].

We also found that low trust in scientists is associated with science-related populist attitudes—that is, beliefs that people's common sense is superior to the expertise of scientists and scientific institutions. This corroborates findings on single countries[38] and provides evidence that populist resentment against science, a prevalent component of the trust crisis narrative, may undermine public trust in scientists.

We also tested preregistered hypotheses assuming that trust in scientists is linked to country-level indicators, including gross domestic product per capita, PISA's science literacy score and the Academic Freedom Index. Contrary to the finding of the Wellcome Global Monitor[6], we found that trust is weakly correlated with the Gini inequality index (that is, trust is higher in countries with more income inequality). One possible explanation for the discrepancy between the Monitor and our study is that urban populations—which are more likely to trust scientists (Fig. 2)—were overrepresented in our samples from countries with high Gini scores (for example, South Africa). However, a non-preregistered analysis advised against this explanation: the extent of oversampling urban participants (the difference of urban-residence individuals in the sample versus in the population) did not moderate the effect of the Gini index on trust in scientists. We found tentative support for another explanation: the relationship between income inequality and trust (Supplementary Fig. 13) is largely driven by countries with a high degree of corruption (primarily Latin American countries as well as sub-Saharan African countries), as indicated by a significant but very low-powered ($1 - \beta$ = 0.25 at $\alpha$ = 0.05) interaction effect of the Gini index and Transparency International's Corruption Perceptions Index[61] (Supplementary Table 3). This suggests that people in countries with high inequality may see scientists as a trustworthy alternative to perceivably corrupt governments and political and economic elites[62–64]. Comparing trust in scientists to trust in the national government (based on country estimates from the Wellcome Global Monitor) supports this assumption. Some countries with higher perceived corruption rank considerably lower in trust in the government than in trust in scientists, whereas the opposite applies to less corrupt countries with lower perceived corruption (Supplementary Fig. 14). Overall, we found no credible evidence that trust in the government and trust in scientists are correlated at the country level ($r_{63}$ = 0.138; $P$ = 0.274; $t$ = 1.104; 95% CI, −0.110 to 0.369). We did not find credible evidence that trust is higher in countries with higher average science literacy scores and government expenditures on education, which challenges assumptions that public understanding of science, and policy measures to increase such understanding, foster trust in scientists[65].

## Normative perceptions of scientists in society
Left–right divides in public opinion about science often centre on the question of whether scientists should take an active role in policymaking[66]. We found that people tend to agree that scientists should engage in society and policymaking (grand mean, 3.64; s.d., 0.87; 1 = strongly

**a** Country-level effects of right-leaning political orientation and trust in scientists

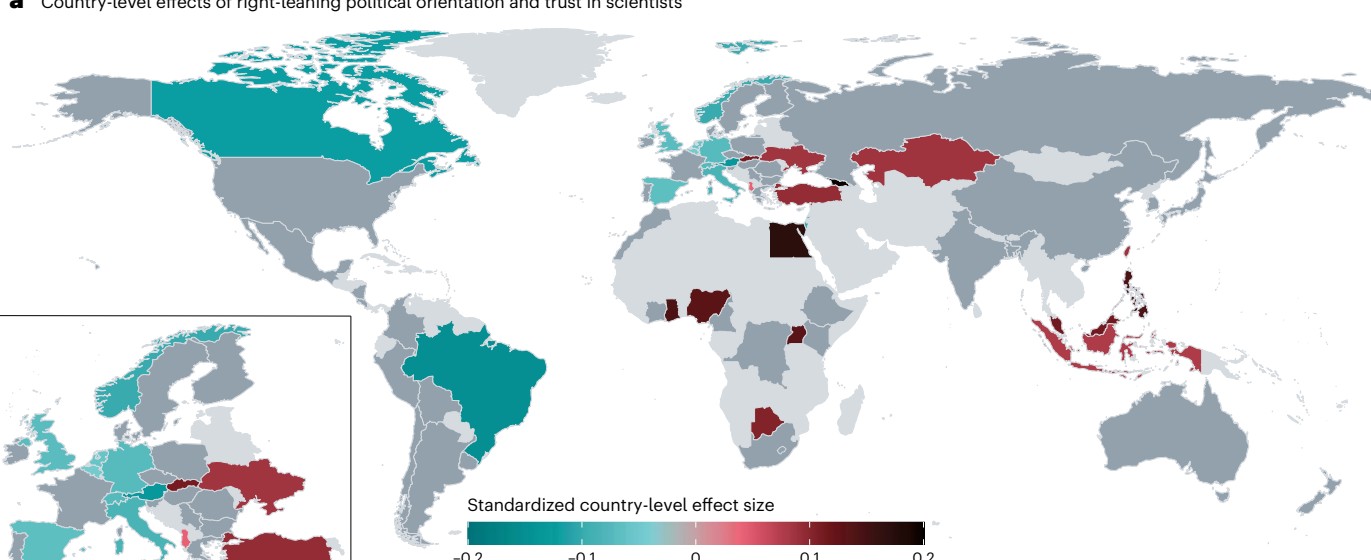

Standardized country-level effect size

−0.2 −0.1 0 0.1 0.2
(Right-leaning people have lower trust) (Left-leaning have lower trust)

**b** Country-level effects of conservative political orientation and trust in scientists

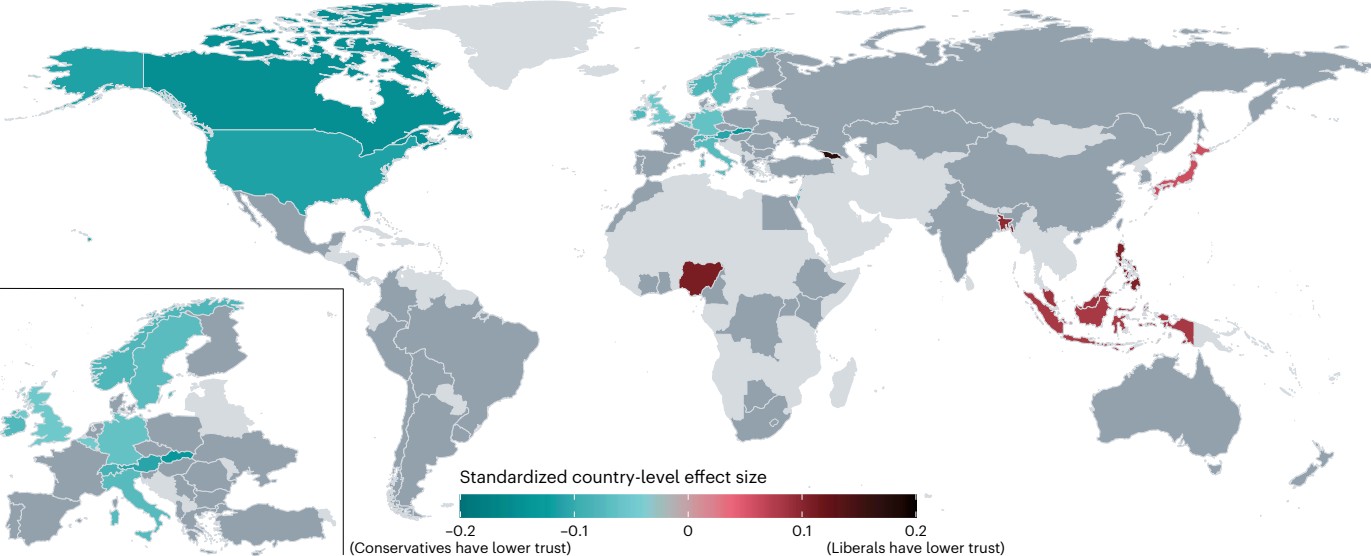

Standardized country-level effect size

−0.2 −0.1 0 0.1 0.2
(Conservatives have lower trust) (Liberals have lower trust)

**Fig. 3 | Relationship of political orientation measures and trust in scientists.** **a**,**b**, Standardized country-level effects of political orientation (in **a**, 1 = strongly left-leaning to 5 = strongly right-leaning; in **b**, 1 = strongly liberal to 5 = strongly conservative) on trust in scientists (1 = very low, 3 = neither high nor low, 5 = very high). These effects are sums of the grand effect for political orientation across all countries and the random effect within each country; they were estimated with weighted linear multilevel regressions that contained random intercepts and random slopes for political orientation (left–right in **a** and liberal–conservative in **b**) across countries. These models control for demographic characteristics.

Two-sided *t*-tests of the estimates used percentile bootstrapping. Countries with significant country-level effects (*P* < 0.05) are displayed in colours. Countries coloured in shades of blue show a positive country-level association of left-leaning (**a**) or liberal (**b**) orientation and trust in scientists (that is, right-leaning people or conservatives have lower trust). Countries coloured in shades of red show a positive country-level association of right-leaning (**a**) or conservative (**b**) orientation and trust in scientists (that is, left-leaning people or liberals have lower trust). Countries with non-significant effects are shaded in dark grey. Countries with no available data are shaded in light grey.

disagree to 5 = strongly agree). In the countries surveyed, a large majority (83%) agree that scientists should communicate about science with the public, particularly in African countries. Overall, only a minority disagree that scientists should actively advocate for specific policies (23%), communicate their findings to politicians (19%) and be more involved in the policymaking process (21%). However, perceptions differ across countries (Supplementary Fig. 15).

About a quarter of the sample selected the scale midpoints, neither agreeing nor disagreeing on whether scientists should be more involved in policymaking and society (Fig. 4). People with high

trust in scientists strongly favour scientists' engagement in society and policymaking ($\beta_{48}$ = 0.262; *P* < 0.001; *t* = 17.86; 95% CI, 0.232 to 0.291), especially in English-speaking countries (Supplementary Fig. 16). Support for scientists' engagement in society and policymaking also varies both between and within countries. People who are younger, have tertiary education and higher income, or live in urban areas generally approve of scientists' engagement in society and policymaking (Supplementary Table 4). Also, right-leaning people and conservatives disapprove of scientists' engagement in society and policymaking.

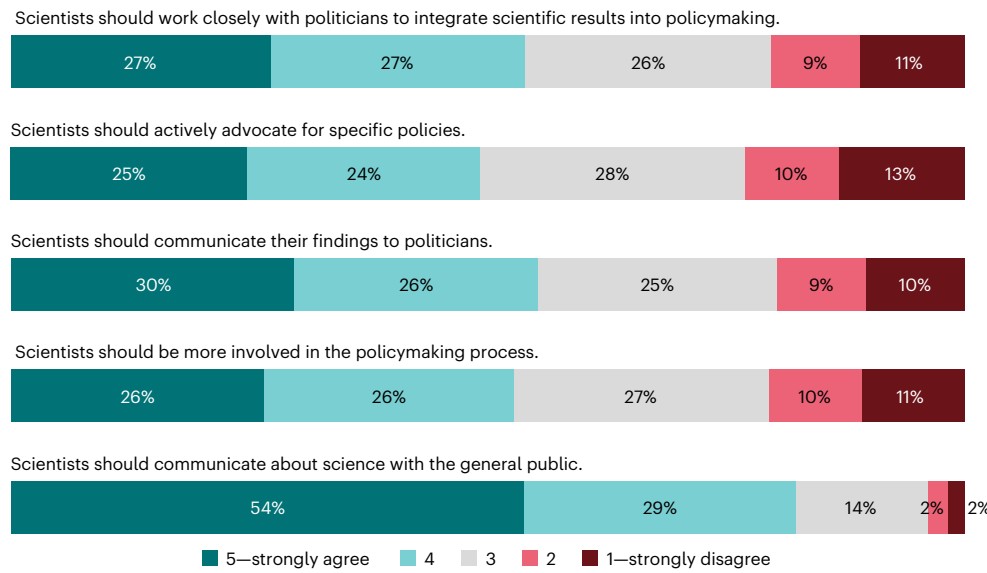

**Fig. 4 | Normative perceptions of scientists in society and policymaking.** Normative perceptions of scientists in society and policymaking using weighted response probabilities.

## Perceived and desired priorities of scientific research

We hypothesized that trust in scientists relates to another normative belief about science: expectations about which societal goals scientists should prioritize[67]. We compared whether people's expectations match their perceptions of whether scientists actually tackle the following goals: improving public health, solving energy problems, reducing poverty, and developing defence and military technology.

Overall, people assign the highest priority to improving public health (mean, 4.49; s.d. = 0.84; 1 = low to 5 = high), followed by solving energy problems (mean, 4.38; s.d. = 0.90) and reducing poverty (mean, 4.09; s.d. = 1.10). The responses suggest a substantial discrepancy between what they perceive science is currently prioritizing and what they expect scientists to prioritize, with poverty reduction showing the most substantial discrepancy (Fig. 5). The least desired research goal is developing defence and military technology (mean, 3.10; s.d. = 1.36). Again, there are large differences between global regions (ranging from a mean of 1.88 (s.d. = 1.21) in Uruguay to a mean of 4.07 (s.d. = 1.52) in the Democratic Republic of Congo). In African and Asian countries, people often demand high priority for developing defence and military technology, unlike people in most European and Latin American countries (Supplementary Fig. 17). Overall, people tend to think that science prioritizes developing defence and military technology more than they desire.

Further analyses show that the discrepancy between people's perceived and desired research priorities is associated with trust in scientists. On the one hand, higher trust is linked to a lower likelihood that scientists' efforts do not meet people's expectations for the following goals: improve public health, solve energy problems and reduce poverty (Supplementary Table 5; see exploratory analyses with reversed hypothesized causality in Supplementary Table 6). In other words, the more people trust scientists, the more they perceive that scientists' efforts exceed expectations. On the other hand, a higher likelihood that scientists' perceived efforts exceed people's expectations is associated with less trust in scientists in the case of developing defence and military technology (that is, those who think scientists are too focused on defence and military technology trust science less).

## Discussion

Our 68-country survey challenges the idea that there is a widespread lack of public trust in scientists. In most countries, scientists and scientific methods are trusted. This finding is in line with other international studies on trust in scientists[5–7]. Our study thus confirms, expands and strengthens previous work that refutes the narrative of a wide-ranging crisis of trust. We expand previous studies by providing a comprehensive dataset on trust in scientists post-pandemic and relying on a theoretically informed multidimensional trust measure. We also show that certain factors, such as being male, being conservative, having high SDO and having science-populist attitudes, are correlated with lower trust in scientists.

Public perception of scientific integrity—one of four components of trust—is somewhat high, but perceptions of scientists' openness are lower. Therefore, scientists wishing to gain public trust could work on being more receptive to feedback and more transparent about their funding and data sources, and invest more effort into communicating about science with the public—which we found to be desired by 83% of respondents. We recommend avoiding top-down communication but encouraging public participation in genuine dialogue, in which scientists seek to consider the insights and needs of other societal actors[68].

Trust differs considerably across countries (Fig. 1 and Supplementary Figs. 18–21), and there is substantial variation of the trust dimensions, which demonstrates the importance of using multidimensional trust measures like ours in comparative survey research[25]. This, in turn, will help scientists and science communicators better understand how to act in ways that increase different components of trustworthiness perceptions—that is, competence, integrity, benevolence and openness. Relatedly, trust in scientists varies across population groups, with women, older people and more educated people trusting scientists more. While demographic characteristics probably cannot cause views about scientists per se, they reflect shared direct or mediated experiences with scientists. For example, women's lived experiences with science are probably different from those of men. Media coverage disproportionally features male scientists and portrays them differently than female scientists, which may evoke different trustworthiness perceptions across genders[69,70].

This information can help scientists and science communicators better tailor their communication to different audiences. Our study assessed trust in scientists without distinguishing between different scientific fields. In some countries, trust may depend on the scientists' discipline and the potential impacts of science on public policy[18,71].

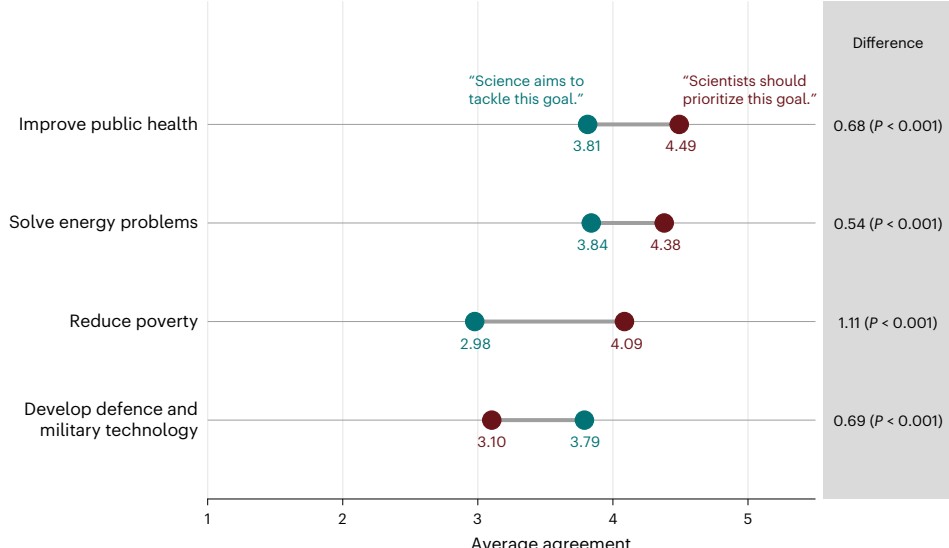

Perceived and desired priorities of scientific research
How strongly do you believe that scientists should prioritize these goals and that science actually tackles them?

**Fig. 5 | Perceived research priorities for four goals of scientific research and desired research priorities.** The grey horizontal lines indicate the discrepancy between perceived research priorities ("Science aims to tackle this goal"; blue) and desired priorities ("Scientists should prioritize this goal"; red). The P values indicate the results of weighted paired-samples, two-sided t-tests for significant differences between perceived priorities and desired priorities.

Trust and its correlates such as right-leaning and conservative political orientation, education and religiosity clearly vary across countries. This exemplifies the need for more international research that includes underrepresented countries and understudied subpopulations. Our findings also highlight the inconsistency in the association between political orientation depending on the measure used (left–right versus liberal–conservative) and the country of study, as well as the importance of ideology—specifically SDO—in relating to trust in scientists. While previous research finds that SDO is associated with the rejection of specific scientific information, such as the reality of climate change[72] or the safety and efficacy of vaccines[26], our findings support the idea that SDO may play a more fundamental role in undermining trust in scientists more generally. Scientists who challenge unjust social hierarchies might increase benevolence perceptions among some groups but would probably further decrease trust among people with SDOs. Interventions could be developed to build the perceived trustworthiness of scientists and involve trusted communicators outside of scientific institutions.

While no country has low trust in scientists on average, lack of trust in scientists by even a small minority needs to be taken seriously. Distrusting minorities may affect considerations of scientific evidence in policymaking, as well as decisions by individuals that can affect society at large, especially if they receive extensive news media coverage and include people in positions of power that can influence policymaking[11,73,74]. A minority of 10% can be sufficient to flip a majority[75], and when a critical mass value of 25% is reached, majority opinion can be tipped[76]. In the context of climate change, an agent-based model showed that an evidence-resistant minority can delay the process of public opinion converging with the scientific consensus[77]. Future research should investigate the size of these distrusting minorities across countries and their characteristics.

In most countries, a majority of people want scientists to take part in policymaking. Future international comparative research should analyse whether opinions differ depending on a scientist's expertise regarding a policy issue[78] and public support for the policy in question[15,79,80]. Future studies should also examine whether normative perceptions of science in policymaking shift when specific scientific disciplines or policy issues are mentioned in real-world settings.

A majority of the public wants scientists to prioritize research on public health and solving energy problems. Yet, most people believe that scientists are currently not tackling these issues sufficiently and think that defence and military technology are prioritized too much. As the perceived benefits of science are strongly correlated with trust in scientists, greater consideration of public research priorities by scientists, funding agencies and philanthropists presents an important avenue to increase trust. At the same time, science communication efforts could increasingly focus on highlighting ongoing research on public health and solving energy problems to elevate the prominence of this research in the minds of the public.

Our study has several limitations, mostly related to the international data collection effort. For example, our survey was fielded in English/French in some countries where English/French are not the most commonly spoken languages, including Botswana, Ethiopia, Ghana, India, Kenya and Nigeria. This probably resulted in the oversampling of more educated population segments. This limitation arises from the fact that our survey was fielded as an online survey, which considerably limited the representability of populations in certain countries with lower Internet penetration. Furthermore, we showed all participants the same definition of science and scientists at the beginning of the survey to make sure that the participants had a common definition in mind when answering the survey. A similar definition had been pretested and used by the Wellcome Global Monitor, one of the main global studies on trust in science. However, we are aware that introducing a very broad definition of science and scientists excludes other epistemological traditions, such as traditional and indigenous knowledges. While our definition of science reflects the dominant and Western conception of science, we want to acknowledge the importance of traditional and indigenous knowledges. It should further be noted that the words 'science' and 'scientists' may be interpreted slightly differently across countries. Thus, while we provided a definition of science and scientists at the beginning of the study, we cannot exclude the fact that translations of the words 'science' and 'scientists' might have slightly different connotations across countries. We provide a more detailed discussion of our limitations in the Supplementary Information.

Newspapers, opinion pieces and books[8] have spread narratives of low public trust in scientists. However, such claims remain largely unsubstantiated by empirical evidence[5–7]. Our Many Labs study provides decision makers, scientists and the public with large-scale and open public-opinion data on trust in scientists that can help these stakeholders maintain and potentially increase trust in scientists.

## Methods

### Overview

The data underlying the analyses were collected in an international pretested, preregistered, cross-sectional online survey ($n$ = 71,922 participants in $k$ = 68 countries) between November 2022 and August 2023 as part of the TISP Many Labs project ('Trust in Science and Science-Related Populism'). TISP is an international, multidisciplinary consortium of 241 researchers from 179 institutions across all continents. The researchers conducted surveys within 88 post hoc weighted quota samples in 68 countries, using the same questionnaire translated into 37 languages. In the following, we describe the procedures used to collect and analyse the data. Further details are available in the Supplementary Information and Mede et al.[46].

### Institutional review board approval

Our questionnaire was considered exempt from full institutional review board (IRB) review from the Harvard University Area Committee on the Use of Human Subjects (protocol no. IRB22-1046) in August 2022. A modified IRB application was submitted and considered exempt from full IRB review by the Harvard University Area Committee on the Use of Human Subjects in November 2022 (protocol no. IRB22-1046). All authors have informed themselves whether IRB approval was required from their institutions and obtained IRB approval if necessary. Supplementary Table 22 lists all IRB approvals that were obtained for this study.

### Pretest

A pretest with $n$ = 401 participants was conducted in the USA in October 2022. The average completion time was 14 min. After the pretest, the questionnaire was slightly modified, and two questions were added to the survey. The data from the pretest were not included in the final analyses.

### Questionnaire

In total, we measured 111 variables. No identifiable information was collected. In the following, we list the measures relevant for this study. The complete questionnaire (in English) is available via the Open Science Framework at https://osf.io/7y2br/. The participants were presented with these components in the order in which they are explained below, but the order of questions and items of multi-item scales was randomized.

**Consent form.** The participants were asked to carefully read a consent form (approved under IRB protocol no. IRB22-1046), which included some general information about the study and the anonymity of the data.

**Demographic data—part 1.** Participants who consented to participating in the study were then asked to indicate their gender (0 = female, 1 = male, 2 = prefer to self-describe, 99 = prefer not to say), age and level of education (1 = did not attend school, 2 = primary education, 3 = secondary education (for example, high school), 4 = higher education (for example, university degree or higher-education diploma)).

**Attention check 1.** The first attention check asked the participants to write the number 213 into a comment box. Participants who failed the attention check were redirected to the end of the survey and were not remunerated. See the Supplementary Information for details on how many respondents failed this attention check in the overall sample and across countries.

**Definition of science and scientists.** The participants were presented with a definition of science and scientists: "When we say 'science', we mean the understanding we have about the world from observation and testing. When we say 'scientists', we mean people who study nature, medicine, physics, economics, history, and psychology, among other things." This definition was based on the Wellcome Global Monitor[6]. We added it because in-depth interviews conducted by the Monitor[6] suggested that including a definition would improve the reliability of cross-country comparisons.

**Perceived benefits of science.** The participants were asked how much they perceived that scientific research benefits people like themselves in their country (1 = not at all, 5 = very strongly) and which geographic region benefits the most and the least from the work that scientists do (1 = Africa, 2 = Asia, 3 = Australia and Oceania, 4 = Europe, 5 = Latin America, 6 = North America).

**Desired and perceived goals of science.** The participants were asked what goals scientists should prioritize (four items; 1 = very low priority, 5 = very high priority) and how strongly they believed that science aims to tackle these goals (1 = not at all, 5 = very strongly).

**Normative perceptions of science and society.** The participants rated their agreement with six statements (for example, scientists should be more involved in the policymaking process) (1 = strongly disagree, 5 = strongly agree). Five of these statements were taken from ref. 66.

**Willingness to be vulnerable to scientists.** Participants' willingness to be vulnerable to scientific guidance was assessed with three items (1 = not at all, 5 = very strongly). Willingness to be vulnerable has been conceptualized as a behavioural trust measure, as it reflects the ceding of authority[48].

**Trust in scientists.** Trust in scientists was assessed with 12 questions on four different dimensions of trustworthiness (that is, competence, integrity, benevolence and openness) (1 = very [unqualified], 5 = very [qualified]), on the basis of Besley et al.[48]. Psychometric analyses (for example, scale reliability, exploratory and confirmatory factor analyses, and measurement invariance tests) can be found in the Supplementary Information.

**Trust in scientific methods.** The participants indicated their level of agreement on whether scientific research methods are the best way to find out if something is true or false (1 = strongly disagree, 5 = strongly agree).

**General trust in scientists.** A single question taken from Funk et al.[81] was used to measure the participants' level of confidence in scientists (1 = no confidence at all, 5 = a great deal of confidence).

**Science-related populism.** Science-related populist attitudes were assessed with the SciPop Scale[38], which comprises eight items (1 = strongly disagree, 5 = strongly agree).

**Attention check 2.** In the second attention check, the participants were instructed to select 'strongly disagree' to a question. Participants who did not select 'strongly disagree' were redirected to the end of the survey and were not remunerated. See the Supplementary Information for details on how many respondents failed this attention check in the overall sample and across countries.

**SDO.** To assess SDO, we used four items from Pratto et al.[82] (1 = extremely opposed, 10 = extremely favour).

**Demographic data—part 2.** The participants indicated their household's yearly net income (in local currency), their political orientation on a spectrum from liberal to conservative (1 = strongly liberal, 5 = strongly conservative, 99 = I don't know) and on a spectrum from left-leaning to right-leaning (1 = strongly left-leaning, 5 = strongly right-leaning, 99 = I don't know), their religiosity (1 = not religious at all, 5 = very strongly religious), and whether they live in a rural or urban area.

Collaborators were allowed to add questions at the end of the survey. Additional questions did not have to be approved by the lead author.

### Translations

The original English survey was translated into the local language where necessary. Translations were done by native speakers who were familiar with the study background and, in many cases, had expertise on survey research. Minor linguistic adjustments were made to the survey if deemed necessary. Major changes in the wording of the original survey instrument had to be approved by the project lead. In total, the survey instrument was translated into 36 languages and dialects[46].

### Preregistration

We submitted a comprehensive preregistration prior to the data collection to the Open Science Framework on 15 November 2022. It included detailed descriptions of our research questions and hypotheses, instruments, data collection, and analytical procedures and is available at https://osf.io/9ksrj. We slightly deviated from the preregistration: we collapsed sparsely populated neighbouring strata for post hoc weighting, excluded confidence in science as a model covariate because of multicollinearity and included SDO as a covariate in the regression model testing predictors of normative perceptions of the role of science in society and politics. Please see the Supplementary Information for deviations from the preregistration.

### Power analysis

To determine our minimum target sample size, we ran simulation-based power analyses using the R package simr (v.1.0.7)[83], which is designed to conduct power analyses for generalized linear mixed models such as those we used in the main study (for detailed information, see Supplementary Information).

### Data collection

Data were collected in surveys that used quotas for age (five bins: 20% 18–29 years, 20% 30–39 years, 20% 40–49 years, 20% 50–59 years and 20% 60 years and older) and gender (two bins: 50% male and 50% female). The participants had to be 18 years of age or older and provide informed consent to participate in the study. The data were collected between November 2022 and August 2023. See Mede et al.[46] for an overview of survey periods across countries. The median completion time was 18 min.

The surveys were programmed in Qualtrics. Participants that completed the survey were remunerated according to the market research company's local rates. All data were collected via online surveys, except in the Democratic Republic of Congo, where the participants were interviewed face-to-face and their responses were recorded in Qualtrics by the interviewers. The collaborators were instructed to work with the market research company Bilendi & respondi, except in most African countries, where collaborators collected data with MSi. Convenience samples were not accepted.

A total of n = 72,135 individuals from 88 samples across k = 68 countries completed the survey (n = 71,922 after the exclusion of duplicate respondents). See Mede et al.[46] for an overview of all included countries and valid sample sizes across countries (that is, after the exclusion of duplicate respondents) and the Supplementary Information for detailed characteristics of the final sample and the representativeness of the surveyed countries by income and region (Supplementary Tables 12 and 13).

### Preparing the dataset

**Exclusion of non-completes and data quality test.** We excluded all respondents who did not complete the survey, because they cancelled participation during the survey, were filtered as their gender × age quota was already full or did not pass one of the two attention checks. Overall, 4.24% of respondents who reached the first attention check did not pass it ("Please write the number 213 into the comment box"), and 24.38% of respondents who reached the second attention check did not pass it ("To show us that you are still paying attention, please select 'strongly disagree'"; ref. 46). We excluded all respondents (n = 213) who completed the survey more than once (for example, IP address checks).

**Outlier value removal.** We removed extreme outlier values for age and household income. Age outliers were defined as values smaller than 18 and bigger than 100. Income outliers were defined as values smaller than zero, equal to zero or outside five times the interquartile range of the log-transformed income distribution within each country after the exclusion of values smaller than or equal to zero. This led to the removal of the age values of 8 respondents and the removal of the income values of 2,457 respondents (1,365 respondents indicated income values equal to or smaller than zero; 1,092 respondents indicated income values outside five times the interquartile range of the log-transformed income distribution within each country after the exclusion of values equal to or smaller than zero).

**Post hoc weighting.** We computed post-stratification weights with the R package survey (v.4.4-2)[84] to ensure that our models would estimate parameters that are representative of the target populations in terms of gender, age and education and have more precise standard errors. We used raking[85] to compute four kinds of weights: (1) post-stratification weights at the country level, (2) sample size weights for each country, (3) post-stratification weights for the complete sample and (4) rescaled post-stratification weights for multilevel analyses[46].

**Scale reliability.** Scales were combined into indices, and psychometric properties were assessed for all indices (Supplementary Information), including scale reliability (Cronbach's $\alpha$ and $\omega$) and cross-country measurement invariances. Scale reliability was good for all scales[46].

### Analyses

**Factors explaining trust in scientists.** To investigate explanatory factors of trust in scientists and explore how their influence varies across countries, we ran a blockwise linear multilevel regression analysis with the R package lme4 (v.1.1-35)[86]. The model included rescaled post-stratification weights[87].

All independent variables in the first, second and third blocks were scaled by country means and country standard deviations. All independent variables in the fourth block were scaled by grand means and grand standard deviations.

We first tried to fit a model with random intercepts and random effects for all independent variables. However, this model failed to converge with three negative eigenvalues and also had a singular fit—that is, some random-effects correlations were close to −1/+1, and some random-effects variances were close to 0. This was probably because the random-effects structure was too complex. We therefore simplified the model as follows: to test the effects of the independent variables on trust in scientists, we fitted a model that contained random intercepts across countries (but no random effects) and inspected fixed-effects estimates. To investigate how the influence of the independent variables varies across countries, we fitted separate models, each of which contained random intercepts across countries and random effects

for one particular independent variable. This entire procedure was completely in line with our preregistration.

Before we fitted the multilevel models, we confirmed that they would fit the data better than fixed-effects models. First, we inspected intraclass correlations for trust in scientists (intraclass correlation coefficient (ICC), 0.170). Second, we ran a likelihood-ratio test. It showed that a random-intercept null model explaining trust in scientists had a significantly better fit than a fixed-effects null model ($\chi^2 = 6,024.9$, $P < 0.001$).

Moreover, we tested for multicollinearity of independent variables for the most complex model—that is, after the inclusion of all three blocks of independent variables (Supplementary Table 16). All variance inflation factors were well below even a very conservative threshold value of 4 (ref. [88]).

**Normative perceptions of science in policymaking.** To examine whether the public demands that scientists should take an active role in society and policymaking, we ran two analyses. First, we computed weighted probabilities of responses to the five items measuring these perceptions. This analysis provided estimates that are approximately representative with regard to gender, age, education and country sample size. Second, we tested explanatory factors of normative perceptions of science in policymaking and society: we fitted a linear multilevel regression model with the R package lme4 (ref. [86]), which explained the average agreement with the five individual items measuring those perceptions, included the rescaled post-stratification weights, and contained trust in scientists, science-related populist attitudes and sociodemographic variables as independent variables—that is, gender (binary; 1 = male), age (continuous), education (binary; 1 = tertiary education), annual household income in US dollars (continuous, log-transformed), place of residence (binary; 1 = urban), right-leaning political orientation (continuous), conservative political orientation (continuous) and religiosity. All independent variables were scaled by country means and country standard deviations.

We specified random intercepts across countries and random effects for trust in scientists and science-related populist attitudes. Significance tests of regression estimates relied on the Satterthwaite method[89]. Before we fitted the multilevel model, we confirmed that it would fit the data better than a fixed-effects model. First, we inspected the intraclass correlation of the normative perceptions index (ICC = 0.104). Second, we ran a likelihood-ratio test, which showed that a random-intercept null model had a significantly better fit than a fixed-effects null model ($\chi^2 = 3,780.6$, $P < 0.001$). Moreover, we tested for multicollinearity of independent variables (Supplementary Table 17). All variance inflation factors were well below even a very conservative threshold value of 4 (ref. [88]).

**Perceived and desired priorities of scientific research.** To explore desires that scientists should prioritize four specific goals (improving public health, solving energy problems, reducing poverty, and developing defences and military technology) as well as perceptions that science actually tackles these goals, we ran three analyses. First, we inspected weighted mean values of responses to the four items measuring priority desires as well as weighted mean values of responses to the four items measuring perceptions that science actually devotes efforts to the four goals.

Second, we ran weighted paired-samples $t$-tests to analyse whether mean values of desires and perceptions differed significantly from each other. These analyses provided estimates that are approximately representative with regard to gender, age, education and country sample size.

Third, we tested explanatory factors of the discrepancy between the desire that scientists should prioritize the four goals and perceptions that science actually tackles them. To do so, we ran four blockwise linear multilevel regression analyses with the R package lme4

(ref. [86]). Each model explained the discrepancy between desires that scientists should prioritize one of the four goals and perceptions that science actually tackles them, with higher outcome variable values indicating that perceptions are more likely to exceed desires and lower outcome variable values indicating that perceptions are more likely to stay behind desires. All models included rescaled post-stratification weights[87].

For each of the four models, we specified random intercepts across countries and random effects for trust in scientists and science-related populist attitudes. Significance tests of regression estimates relied on the Satterthwaite method[89]. Before we fitted the multilevel models, we confirmed that they would fit the data better than fixed-effects models. First, we inspected the intraclass correlations of the four discrepancy scores (health: ICC = 0.112; energy: ICC = 0.079; poverty: ICC = 0.134; defence: ICC = 0.107). Second, we ran likelihood-ratio tests, which showed that random-intercept null models had significantly better fit than fixed-effects null models (health: $\chi^2 = 3,246.2$, $P < 0.001$; energy: $\chi^2 = 2,264.4$, $P < 0.001$; poverty: $\chi^2 = 4,835.2$, $P < 0.001$; defence: $\chi^2 = 3,669.0$, $P < 0.001$). Moreover, we tested for multicollinearity of independent variables for the most complex models (Supplementary Table 18). All variance inflation factors were well below even a very conservative threshold value of 4 (ref. [88]).

**Reporting summary**

Further information on research design is available in the Nature Portfolio Reporting Summary linked to this article.

## Data availability

The dataset underlying this Article is publicly available at https://doi.org/10.17605/OSF.IO/5C3QD. Mede et al.[46] provide detailed information on the dataset, including data collection and preprocessing.

## Code availability

The code for replicating the analyses underlying this Article is publicly available at https://osf.io/wj34h/.

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

## Acknowledgements

We thank D. Lombardi (University of Zurich) for managing the author list and author contributions and P. Licari (Technite) for valuable methodological advice. The following funders had no role in study design, data collection and analysis, decision to publish or preparation of the manuscript: Swiss National Science Foundation Postdoc Mobility Fellowship (P500PS_202935) (V.C.); Harvard University Faculty Development Funds (V.C.); Swiss Federal Office of Energy (SI/502093–01) (S.B.); University of Zurich/IMKZ (M.S.S.); the HELTS Foundation (E.W.M.); School of Psychology, University of Sheffield (I.A. and H.G.); Beasiswa Pendidikan Indonesia Kemendikbudristek—LPDP provided by Balai Pembiayaan Pendidikan Tinggi (BPPT) Kemdikbudristek and LPDP Indonesia (I.A.); Department of Economics, University of Warwick (E.A.); John Templeton Foundation grant no. 61378 (M. Alfano); Australian Research Council grant no. DP190101507 (M. Alfano); Resnick Sustainability Institute (R.M.A.); Universitas Islam Negeri Sunan Kalijaga (D.A.); University 'Aleksandër Moisiu', Durrës (A. Bajrami and R.T.); Africa Albarado Fund (R. Bardhan); Cambridge Africa (R. Bardhan); ESRC GCRF (R. Bardhan); SNSF (VAR-EXP) (E.B., P.K. and A.Z.); German Research Foundation grant no. BE 3970/12-1 (C. Betsch);

Sloan School of Management, Massachusetts Institute of Technology (A.Y.B.); SWPS University (O. Białobrzeska and M. Parzuchowski); University of Warsaw (M.B. and P.H.); Harvey Mudd College (K. Breeden); Boston University (Startup Funds) (T.C. and M.M.); Jagiellonian University (G.C. and E. Szumowska); Quadrature Climate Fund (R.D. and E. Shuckburgh); Bill & Melinda Gates Foundation grant no. OPP1144 (R.D.); EDCTP2 Programme (TMA2020CDF-3171) (I.M.A.); Cambridge Humanities Research Grant (R.D.); CRASSH grant fund for climaTRACES lab (R.D.); Keynes Fund (R.D.); UKRI ODA International Partnership Fund (R.D. and R. Bardhan); COVID-19 Rapid Response grant from the University of Vienna (K.C.D., C.L., J.P.N., E.P. and B.T.); OptimAgent (German Federal Ministry of Education and Research, funding code 031L0299D) and University of Lübeck (A.C.V. and L. Kojan); Austrian Science Fund grant FWF, I3381 (K.C.D., C.L., J.P.N. and E.P.); the Austrian Science Fund FWF: W1262-B29 (C.L. and B.T.); Deutsche Forschungsgesellschaft grant no. RE 4752/1-1 (C.D. and F.G.R.); David and Claudia Harding Foundation (C.D. and F.G.R.); Basic Research Program at the National Research University Higher School of Economics (HSE University) (D.D., A.G., D.G. and E.K.); University of Lodz (M.D. and I. Warwas); Australian Research Council grant no. FT190100708 (U.K.H.E.); School of Economics Interdisciplinary funding at University of Birmingham (M.E.); Kieskompas.nl (T.W.E. and A. Krouwel); USAID (M. Facciani and T.W.); Aarhus University Research Foundation grant no. AUFF-E-2019-9-13 (A.F.-B. and S. Fuglsang); Carleton College (C. Farhart); internal project costs IWM (H.F.); Australian Research Council grant no. DP190101675 (O. Ghasemi); Government of Alberta Major Innovation Fund grant no. RES0049213 (E.G.); Conacyt grant no. A1S9013 (C.G.-B. and A.C.H.-M.); Simone Rödder (L.G.); Hixon Center for Climate and the Environment, Harvey Mudd College (L.N.H.); Faculty Research Grant of City University of Hong Kong grant no. PJ9618021 (G.H.); research grant from the College of Social Sciences, Kimep University (N.I. and Z.K.); Hitachi Fund Support for Research Related to Infectious Diseases (M.I. and M. Tanaka); JST-RISTEX ELSI grant no. JPMJRX20J3 (M.I. and M. Tanaka); School of Geography, Planning, and Spatial Sciences, University of Tasmania (C.A.J. and C.H.L.); Centre for Marine Socioecology, University of Tasmania (C.A.J. and C.H.L.); University of Bamberg (S.J. and S.M.); Nicolaus Copernicus University (D.J. and A.D.W.); Institute of Communication Studies and Journalism, Charles University (T.K.R. and K. Poliaková); the John Templeton Foundation, grant no. 61580 (M.A.P.); Zhangir Kabdulkair (H.K. and B. Scoggins); Concerted Research Action grant from the Fédération Wallonie-Bruxelles (Belgium) ('The Socio-Cognitive Impact of Literacy') (O.K.); Center for Climate and Energy Transformation, University of Bergen, Norway (H.H. and S. Kristiansen); University of Turku (A. Koivula and P.R.); Victoria University of Wellington (L.S.K.); NORFACE Joint Research Programme on Democratic Governance in a Turbulent Age (T.K., K. Petkanopoulou and J.v.N.); NWO (T.K., K. Petkanopoulou and J.v.N.); European Commission through Horizon 2020 grant no. 822166 (T.K., K. Petkanopoulou and J.v.N.); Australian Research Council grant no. DP180102384 (N.L. and R.M.R.); John Templeton Foundation grant no. 62631 (N.L., R.M.R. and M. Alfano); internal research/creative project grant (N.M.L.); Social Sciences and Humanities Research Council grant no. 430-2022-00711 (N.M.L.); School of Psychology and Public Health Internal Grant Scheme 2022 (M.D.M.); 'An Evolutionary and Cultural Perspective on Intellectual Humility via Intellectual Curiosity and Epistemic Deference' from the John Templeton Foundation (H.M.); SCALUP grant from the ANR grant no. ANR-21-CE28-0016-01 (H.M.); ANR grants to PSL and the DEC ANR-10-IDEX-0001-02, and ANR-17-EURE-0017 (H.M.); University of Delaware (J.M.); School of Medicine and Psychology, Australian National University (E.J.N. and S.K.S.); university research budget (T. Ostermann and J.R.P.); Trinity Western University (J.P.-H.); Swedish Research Council grant no. 2020-02584 (P.P.); University of Silesia in Katowice (M.P.-C. and K.P.-B.); John Templeton Foundation Academic Cross Training Fellowship grant no.

61580; University of Warsaw under the Priority Research Area V of the 'Excellence Initiative—Research University' programme (A.P. and E.Z.-P.); National Science and Technology Council, Taiwan (ROC) grant nos 111-2628-H-002-003- and 112-2628-H-002-002- (A.R.); the São Paulo Research Foundation—FAPESP grant no. 2019/26665-5 (G.R.); Deutsche Forschungsgemeinschaft (DFG, German Research Foundation)—458303980 (F.G.R. and C.D.); Aston University (J.P. Reynolds); ANR PICS (I.R.); Fundação para a Ciência e a Tecnologia, UIDB/04295/2020 and UIDP/04295/2020 (O. Santos and R.R.S.); European Union's Horizon 2020 research and innovation programme grant no. 964728 (JITSUVAX) (P.S.); Université Officielle de Bukavu (J.S.N.); ETH Zurich (J.S.); Swiss Agency for Development and Cooperation grant no. 7F09521 (L.S. and J.S.); NOMIS Foundation (M. Tsakiris); NOMIS Foundation (R.M.); School of Psychological Sciences, University of Melbourne (I. Walker); Observatory for Research on Media and Journalism, University of Louvain (G.L. and O. Standaert); European Research Council Advanced Grant 'Consequences of conspiracy theories—CONSPIRACY_FX' grant no. 101018262 (K.M.D.); Universität Hamburg (S. Schulreich); Genome Canada and Genome Alberta, Canada, LSARP Project Integrating Genomic Approaches to Improve Dairy Cattle Resilience: A Comprehensive Goal to Enhance Canadian Dairy Industry Sustainability (E.G.); Alberta Ministry of Technology and Innovation Canada, Major Innovation Fund Project AMR—One Health Consortium (E.G.); Aarhus University Research Foundation grant no. AUFF-E-2019-9-4 (P.M.); University of Hamburg (S.R., L.G. and M.J.); Resnick Sustainability Institute, Critical Zone Initiative, California Institute of Technology (R.M.A.); UK Research and Innovation under the UK government's Horizon Europe funding guarantee EP/X042758/1 (J.P. Reynolds); Spanish Foundation for Science and Technology (FECYT) (C.R.S., C.D.-C. and P.C.-Á.); CNPq—INCT (National Institute of Science and Technology on Social and Affective Neuroscience, grant no. 406463/2022-0) (G.R. and F.A.); Swiss National Science Foundation PRIMA Grant (no. PR00P1_193128) (J.L.G.); Economic and Social Research Council, UK (grant reference no. ES/X000702/1) (S.M.); and Leverhulme International Professorship Grant (no. LIP-2022-001) (R.M.); Slovak Research and Development Agency (APVV), grant no. APVV-22-0242 (O. Buchel).

## Author contributions

Conceptualization: V.C., N.G.M., S.B., J.B., C. Brick, M.J., E.W.M., S.M., N.O., M.S.S. and S.v.d.L. Data curation: N.G.M. Formal analysis: N.G.M. Methodology: V.C., N.G.M., S.B., J.B., C. Brick, M.J., E.W.M., S.M., N.O., M.S.S. and S.v.d.L. Project administration: V.C. Software: V.C. and N.G.M. Supervision: V.C. and N.G.M. Validation: V.C. and N.G.M. Visualization: V.C. and N.G.M. Investigation: V.C., N.G.M., N.I.A.A., S.A., N.A.S., B.A., I.A., E.A., A.A., M. Alfano, I.M.A., M. Altenmüller, R.M.A., R.A., T.A., D.A., F.A., A. Bajrami, R. Bardhan, K. Bati, E.B., A.Y.B., O. Białobrzeska, K. Breeden, A. Bret, O. Buchel, P.C.-Á., F.C., A.C.V., T.C., R.K.C., S.Ç., G.C., S.D.P., R.D., S. Delouvée, C.D.-C., L.D.S., K.C.D., S. Dohle, K.M.D., C.D., D.D., M.D., U.K.H.E., T.W.E., M. Facciani, A.F.-B., M.Z.F., X.F., C. Farhart, C. Feldhaus, M. Ferreira, S. Feuerriegel, H.F., J.F., M. Friese, S. Fuglsang, A.G., P.G.-V., M.E.G.V., W.G., O. Genschow, O. Ghasemi, T.G., J.L.G., E.G., M.G., C.G.-B., H.G., D.G., G.M.G., L.G., H.H., D.H., L.N.H., P.H., A.C.H.-M., A.H., G.H., M. Huff, M. Hurley, N.I., M.I., M.T.I., Y.J., T.J., C.A.J., S.J., D.J., Z.K., J.-J.K., S. Kavassalis, J.R.K., M.K., T.K.R., O.K., H.K., A. Koivula, L. Kojan, E.K., L. Koppel, K.K.N.C., A. Kosachenko, L.S.K., P.K., S. Kristiansen, A. Krouwel, T.K., E.A.K., C.L., A. Lantian, A. Lazić, J.-B.L., Z.L., N.L., A.M.L., G.L., A. Loeschel, A.L.O., C.L.-V., N.M.L., C.H.L., K.L.-T., M.D.M., S.J.M., H.M., J.M., T.L.M., J.M.M., P.M., F.M.-R., M.M., I.M., Z.M., J.N., E.J.N., J.P.N., N.-N.V.N., D.N., T. Ostermann, J.P.-H., M. Pantazi, P.P., P.P.S., M.P.-C., M. Parzuchowski, Y.G.P., A.R.P., M.A.P., K. Petkanopoulou, M.B.P., J.P., D.P., A.P., K. Poliaková, E.P., K.P.-B., D.M.A.Q., P.R., A.R., F.G.R., C.R.S., G.R., J.R., S.R., J.P. Röer, R.M.R., I.R., O. Santos, R.R.S., P.S., B. Scoggins, A.S., J.S.N., E. Shuckburgh, N. Solak, L.S., B. Spruyt, O. Standaert, S.K.S.,

**Article** https://doi.org/10.1038/s41562-024-02090-5

G.S., S. Syropoulos, B. Szaszi, E. Szumowska, M. Tanaka, C.T.E., C.T.-E., B.T., A.K.T., R.T., D.T.-F., M. Tyrala, Ö.M.U., I.C.U., J.v.N., C.V., S.V., I.V., A.v.B., I. Walker, I. Warwas, M. Weber, T.W., M. Westfal, F.W., A.D.W., Z.X., J.X., E.Z.-P., A.Z. and R.A.Z. Resources: N.I.A.A, S.A., N.A.S., B.A., I.A., E.A., A.A., M. Alfano, I.M.A., M. Alsobay, M. Altenmüller, R.M.A., R.A., T.A., D.A., F.A., A. Bajrami, R. Bardhan, K. Bati, E.B., A.Y.B., O. Białobrzeska, K. Breeden, A. Bret, O. Buchel, P.C.-Á., F.C., A.C.V., T.C., R.K.C., S.Ç., G.C., S.D.P., R.D., S. Delouvée, C.D.-C., L.D.S., K.C.D., S. Dohle, K.M.D., C.D., D.D., M.D., U.K.H.E., T.W.E., M. Facciani, A.F.-B., M.Z.F., X.F., C. Farhart, C. Feldhaus, M. Ferreira, S. Feuerriegel, H.F., J.F., M. Friese, S. Fuglsang, A.G., P.G.-V., M.E.G.V., W.G., O. Genschow, O. Ghasemi, T.G., J.L.G., E.G., M.G., C.G.-B., H.G., D.G., G.M.G., L.G., H.H., D.H., L.N.H., P.H., A.C.H.-M., A.H., G.H., M. Huff, M. Hurley, N.I., M.I., M.T.I, Y.J., T.J., C.A.J., S.J., D.J., Z.K., J.-J.K., S. Kavassalis, J.R.K., M.K., T.K.R., O.K., H.K., A. Koivula, L. Kojan, E.K., L. Koppel, K.K.N.C., A. Kosachenko, L.S.K., P.K., S. Kristiansen, A. Krouwel, T.K., E.A.K., C.L., A. Lantian, A. Lazić, J.-B.L., Z.L., N.L., A.M.L., G.L., A. Loeschel, A.L.O., C.L.-V., N.M.L., C.H.L., K.L.-T., M.D.M., S.J.M., H.M., J.M., T.L.M., J.M.M., P.M., F.M.-R., M.M., I.M., Z.M., J.N., E.J.N., J.P.N., N.-N.V.N., D.N., T. Ostermann, J.P.-H., M. Pantazi, P.P., P.P.S., M.P.-C., M. Parzuchowski, Y.G.P., A.R.P., M.A.P., C.R.P., K. Petkanopoulou, M.B.P., J.P., D.P., A.P., K. Poliaková, E.P., K.P.-B., D.M.A.Q., P.R., A.R., F.G.R., C.R.S., G.R., J.P. Reynolds, J.R., S.R., J.P. Röer, R.M.R., I.R., O. Santos, R.R.S., P.S., S. Schulreich, B. Scoggins, A.S., J.S.N., E. Shuckburgh, J.S., N. Solak, L.S., B. Spruyt, O. Standaert, S.K.S., G.S., S. Syropoulos, B. Szaszi, E. Szumowska, M. Tanaka, C.T.E., C.T.-E., B.T., A.K.T., R.T., D.T.-F., M. Tsakiris, M. Tyrala, Ö.M.U., I.C.U., J.v.N., C.V., S.V., I.V., M.V., A.v.B., I. Walker, I. Warwas, M. Weber, T.W., M. Westfal, F.W., A.D.W., Z.X., J.X., E.Z.-P., A.Z. and R.A.Z.

## Funding

## Competing interests

The authors declare no competing interests.

## Additional information

**Correspondence and requests for materials** should be addressed to Viktoria Cologna.

The full author contributions continue:

Funding acquisition: V.C., N.O., M.S.S., J.B., E.W.M., S.B., C.Brick, B.A., I.A., E.A., M.Alfano, M.Altenmüller, R.M.A., D.A., A.Bajrami, R.Bardhan, E.B., C.Betsch, A.Y.B., R.Bhui, O.Białobrzeska, M.B., A. Bouguettaya, K. Breeden, A. Bret, O. Buchel, P.C.-Á., F.C., A.C.V., T.C., S.Ç., G.C., R.D., S. Delouvée, C.D.-C., L.D.S., K.C.D., S. Dohle, K.M.D., C.D., D.D., M.D., U.K.H.E., M.E., B.E., T.W.E., M. Facciani, A.F.-B., X.F., C. Farhart, C. Feldhaus, M. Ferreira, S. Feuerriegel, H.F., J.F., M. Friese, S. Fuglsang, A.G., M.E.G.V., W.G., O. Genschow, O. Ghasemi, T.G., J.L.G., E.G., M.G., C.G.-B., H.G., D.G., G.M.G., L.G., H.H., D.H., L.N.H., P.H., A.C.H.-M., A.H., G.H., M. Huff, M. Hurley, N.I., M.I., C.A.J., S.J., D.J., Z.K., J.-J.K., S. Kavassalis, J.R.K., T.K.R., O.K., H.K., A. Koivula, L. Kojan, E.K., L. König, L. Koppel, A. Kosachenko, J.K., L.S.K., P.K., S. Kristiansen, A. Krouwel, T.K., E.A.K., C.L., A. Lantian, J.-B.L., Z.L., N.L., A.M.L., G.L., A. Loeschel, A.L.O., C.L.-V., N.M.L., C.H.L., K.L.-T., M.D.M., S.J.M., R.M., H.M., J.M., T.L.M., J.M.M., P.M., F.M.-R., M.M., E.J.N., J.P.N., T. Ostermann, T. Otterbring, J.P.-H., M. Pantazi, P.P., P.P.S., M.P.-C., M.P., Y.G.P., A.R.P., M.A.P., C.R.P., K. Petkanopoulou, J.P., D.P., A.P., K. Poliaková, E.P., K.P.-B., P.R., A.R., F.G.R., C.R.S., J.P. Reynolds, J.R., S.R., J.P. Röer, R.M.R., I.R., O. Santos, R.R.S., P.S., S. Schulreich, B. Scoggins, A.S., J.S.N., E. Shuckburgh, J.S., N. Solak, L.S., B. Spruyt, O. Standaert, S.K.S., G.S., S. Syropoulos, E. Szumowska, M. Tanaka, C.T.E., C.T.-E., B.T., R.T., D.T.-F., M. Tsakiris, M. Tyrala, Ö.M.U., I.C.U., J.v.N., C.V., S.V., I.V., A.v.B., I. Walker, I. Warwas, M. Weber, T.W., M. Westfal, F.W., A.D.W., Z.X., J.X., E.Z.-P., A.Z. and R.A.Z. Writing—original draft: V.C. Writing—review and editing: V.C., N.G.M., S.B., J.B., C. Brick, M.J., E.W.M., S.M., N.O., M.S.S., S.v.d.L., N.I.A.A, S.A., N.A.S., B.A., I.A., E.A., A.A., M. Alfano, I.M.A., M. Alsobay, M. Altenmüller, R.M.A., R.A., T.A., P.A., D.A., F.A., A. Bajrami, R. Bardhan, K. Bati, E.B., C. Betsch, A.Y.B., R. Bhui, O. Białobrzeska, M.B., A. Bouguettaya, K. Breeden, A. Bret, O. Buchel, P.C.-Á., F.C., A.C.V., T.C., R.K.C., S.Ç., G.C., S.D.P., R.D., S. Delouvée, C.D.-C., L.D.S., K.C.D., S. Dohle, K.M.D., C.D., D.D., M.D., U.K.H.E., M.E., B.E., T.W.E., M. Facciani, A.F.-B., M.Z.F., X.F., C. Farhart, C. Feldhaus, M. Ferreira, S. Feuerriegel, H.F., J.F., M. Friese, S. Fuglsang, A.G., P.G.-V., M.E.G.V.,

Viktoria Cologna [1,2,3 ✉], Niels G. Mede [2], Sebastian Berger [4], John Besley[5], Cameron Brick [6,7], Marina Joubert [8], Edward W. Maibach [9], Sabina Mihelj [10], Naomi Oreskes[1], Mike S. Schäfer [2], Sander van der Linden[11], Nor Izzatina Abdul Aziz [12], Suleiman Abdulsalam[13], Nurulaini Abu Shamsi[14], Balazs Aczel [15], Indro Adinugroho[16,17], Eleonora Alabrese[18], Alaa Aldoh[6], Mark Alfano [19], Innocent Mbulli Ali[20], Mohammed Alsobay[21], Marlene Altenmüller [22,23], R. Michael Alvarez[24], Richard Amoako[25], Tabitha Amollo[26], Patrick Ansah[25], Denisa Apriliawati [27], Flavio Azevedo[28,29], Ani Bajrami [30], Ronita Bardhan[31], Keagile Bati [32], Eri Bertsou[33], Cornelia Betsch [34], Apurav Yash Bhatiya [35], Rahul Bhui [21,36], Olga Białobrzeska [37], Michał Bilewicz [38], Ayoub Bouguettaya[39], Katherine Breeden[40], Amélie Bret[41], Ondrej Buchel [42], Pablo Cabrera-Álvarez[43], Federica Cagnoli[44], André Calero Valdez[45], Timothy Callaghan[46], Rizza Kaye Cases[47], Sami Çoksan [48,49], Gabriela Czarnek[50], Steven De Peuter [51], Ramit Debnath [24,52], Sylvain Delouvée [53], Lucia Di Stefano[44], Celia Díaz-Catalán [43,54], Kimberly C. Doell [55], Simone Dohle[56], Karen M. Douglas [57], Charlotte Dries[58], Dmitrii Dubrov [59], Małgorzata Dzimińska[60], Ullrich K. H. Ecker [61], Christian T. Elbaek [62], Mahmoud Elsherif[39], Benjamin Enke [63], Tom W. Etienne[64], Matthew Facciani[65], Antoinette Fage-Butler[66], Md. Zaki Faisal[67], Xiaoli Fan[68], Christina Farhart[69], Christoph Feldhaus[70], Marinus Ferreira[19], Stefan Feuerriegel [71], Helen Fischer[72], Jana Freundt[73], Malte Friese [74], Simon Fuglsang[75], Albina Gallyamova[59], Patricia Garrido-Vásquez[76], Mauricio E. Garrido Vásquez[76], Winfred Gatua[77], Oliver Genschow[78], Omid Ghasemi [79,80], Theofilos Gkinopoulos[50], Jamie L. Gloor[81], Ellen Goddard[68], Mario Gollwitzer [22], Claudia González-Brambila [82], Hazel Gordon[16], Dmitry Grigoryev [59], Gina M. Grimshaw[83], Lars Guenther[84], Håvard Haarstad[85,86], Dana Harari[87], Lelia N. Hawkins[88], Przemysław Hensel [89], Alma Cristal Hernández-Mondragón[90], Atar Herziger[87], Guanxiong Huang[91], Markus Huff[72,92], Mairéad Hurley [93], Nygmet Ibadildin[94], Maho Ishibashi [95], Mohammad Tarikul Islam[96], Younes Jeddi[13], Tao Jin[97], Charlotte A. Jones[98], Sebastian Jungkunz[99,100], Dominika Jurgiel [101], Zhangir Kabdulkair[94], Jo-Ju Kao[102], Sarah Kavassalis[88], John R. Kerr [103], Mariana Kitsa[104], Tereza Klabíková Rábová [105], Olivier Klein [106], Hoyoun Koh[107], Aki Koivula [108], Lilian Kojan[45], Elizaveta Komyaginskaya[59], Laura König[109,110], Lina Koppel [111], Kochav Koren Nobre Cavalcante[112], Alexandra Kosachenko[113], John Kotcher [9], Laura S. Kranz[83], Pradeep Krishnan[33], Silje Kristiansen[86,114], André Krouwel[115], Toon Kuppens[116], Eleni A. Kyza[117], Claus Lamm [55], Anthony Lantian [118], Aleksandra Lazić [119], Oscar Lecuona [120], Jean-Baptiste Légal [118], Zoe Leviston [121], Neil Levy [19,122], Amanda M. Lindkvist [111], Grégoire Lits[123], Andreas Löschel[70], Alberto López Ortega [115], Carlos Lopez-Villavicencio[124], Nigel Mantou Lou [125], Chloe H. Lucas [98], Kristin Lunz-Trujillo [126,127], Mathew D. Marques [128], Sabrina J. Mayer[99], Ryan McKay[129], Hugo Mercier [130], Julia Metag[131], Taciano L. Milfont [132], Joanne M. Miller [133], Panagiotis Mitkidis [62], Fredy Monge-Rodríguez[124], Matt Motta [46], Iryna Mudra[104], Zarja Muršič[134], Jennifer Namutebi[135], Eryn J. Newman[121], Jonas P. Nitschke [55], Ntui-Njock Vincent Ntui[136], Daniel Nwogwugwu [137], Thomas Ostermann [138], Tobias Otterbring [139], Jaime Palmer-Hague[140], Myrto Pantazi[106], Philip Pärnamets [141], Paolo Parra Saiani [44], Mariola Paruzel-Czachura[142,143], Michal Parzuchowski [37], Yuri G. Pavlov [144], Adam R. Pearson[145], Myron A. Penner[140], Charlotte R. Pennington [146], Katerina Petkanopoulou[147], Marija B. Petrović [119], Jan Pfänder [130], Dinara Pisareva[107], Adam Ploszaj[148], Karolína Poliaková[105], Ekaterina Pronizius [55], Katarzyna Pypno-Blajda [142], Diwa Malaya A. Quiñones[149], Pekka Räsänen [108], Adrian Rauchfleisch [102], Felix G. Rebitschek[58,150], Cintia Refojo Seronero[43], Gabriel Rêgo[29,151], James P. Reynolds[146], Joseph Roche[93], Simone Rödder[152], Jan Philipp Röer [138], Robert M. Ross [19], Isabelle Ruin[153], Osvaldo Santos[154], Ricardo R. Santos[154,155], Philipp Schmid [34,156,157], Stefan Schulreich[158,159], Bermond Scoggins[160], Amena Sharaf[161], Justin Sheria Nfundiko[162,163], Emily Shuckburgh[52], Johan Six[3], Nevin Solak [164], Leonhard Späth[3], Bram Spruyt[165], Olivier Standaert[123], Samantha K. Stanley [79,80,121], Gert Storms [51], Noel Strahm [4], Stylianos Syropoulos [166], Barnabas Szaszi [15], Ewa Szumowska [50], Mikihito Tanaka[167], Claudia Teran-Escobar[118,153], Boryana Todorova [55], Abdoul Kafid Toko[13], Renata Tokrri[168], Daniel Toribio-Florez [57], Manos Tsakiris [129,169], Michael Tyrala [170], Özden Melis Uluğ[171], Ijeoma Chinwe Uzoma[172], Jochem van Noord[116,165], Christiana Varda [117,173], Steven Verheyen[174], Iris Vilares[97], Madalina Vlasceanu [175], Andreas von Bubnoff[176], Iain Walker[121,177], Izabela Warwas[60], Marcel Weber[74], Tim Weninger[65], Mareike Westfal[78], Florian Wintterlin [131], Adrian Dominik Wojcik[178], Ziqian Xia [179], Jinliang Xie[180], Ewa Zegler-Poleska [148], Amber Zenklusen[33] & Rolf A. Zwaan [174]

[1]Department of the History of Science, Harvard University, Cambridge, MA, USA. [2]Department of Communication and Media Research, University of Zurich, Zurich, Switzerland. [3]Department of Environmental Systems Science, ETH Zurich, Zurich, Switzerland. [4]Institute of Sociology, University Bern, Bern, Switzerland. [5]Department of Advertising + Public Relations, Michigan State University, East Lansing, MI, USA. [6]Department of Psychology, University of Amsterdam, Amsterdam, the Netherlands. [7]Department of Psychology, Inland Norway University of Applied Sciences, Lillehammer, Elverum, Norway. [8]Centre for Research on Evaluation, Science and Technology, Stellenbosch University, Stellenbosch, South Africa. [9]Centre for Climate Change Communication, George Mason University, Fairfax, VA, USA. [10]Centre for Research in Communication and Culture, Department of Communication and Media, Loughborough University, Loughborough, UK. [11]Department of Psychology, University of Cambridge, Cambridge, UK. [12]Institute of Malaysian and International Studies, National University of Malaysia, Bangi, Malaysia. [13]School of Collective Intelligence, Mohammed VI Polytechnic University, Ben Guerir, Morocco. [14]Department of Science & Technology Studies, Faculty of Science, Universiti Malaya, Kuala Lumpur, Malaysia. [15]ELTE Institute of

Psychology, Eotvos Lorand University, Budapest, Hungary. [16]School of Psychology, University of Sheffield, Sheffield, UK. [17]Faculty of Psychology, Atma Jaya Catholic University of Indonesia, Jakarta, Indonesia. [18]Department of Economics, University of Bath, Bath, UK. [19]Department of Philosophy, Macquarie University, Sydney, New South Wales, Australia. [20]Department of Biochemistry, Faculty of Science, University of Dschang, Dschang, Cameroon. [21]Sloan School of Management, Massachusetts Institute of Technology, Cambridge, MA, USA. [22]Department of Psychology, Ludwig-Maximilians-Universität München, Munich, Germany. [23]Leibniz Institute for Psychology, Trier, Germany. [24]Linde Center for Science, Society, and Policy, Division of Humanities and Social Science, California Institute of Technology, Pasadena, CA, USA. [25]Department of Communication, George Mason University, Fairfax, VA, USA. [26]Department of Physics, Egerton University, Egerton, Kenya. [27]Department of Psychology, Universitas Islam Negeri Sunan Kalijaga, Yogyakarta, Indonesia. [28]Department of Interdisciplinary Social Science, University of Utrecht, Utrecht, the Netherlands. [29]National Institute of Science and Technology on Social and Affective Neuroscience, São Paulo, Brazil. [30]Museum of Natural Sciences 'Sabiha Kasimati', University of Tirana, Tirana, Albania. [31]Department of Architecture, University of Cambridge, Cambridge, UK. [32]Department of Biomedical Sciences, University of Botswana, Gaborone, Botswana. [33]Institute of Political Science, University of St. Gallen, St. Gallen, Switzerland. [34]Institute for Planetary Health Behaviour, University of Erfurt, Erfurt, Germany. [35]Department of Economics, University of Birmingham, Birmingham, UK. [36]Institute for Data, Systems, and Society, Massachusetts Institute of Technology, Cambridge, MA, USA. [37]Institute of Psychology, SWPS University, Warsaw, Poland. [38]Faculty of Psychology, University of Warsaw, Warsaw, Poland. [39]School of Psychology, University of Birmingham, Birmingham, UK. [40]Computer Science Department, Harvey Mudd College, Claremont, CA, USA. [41]Department of Psychology, Nantes Université, LPPL, Nantes, France. [42]Institute for Sociology of the Slovak Academy of Sciences, Bratislava, Slovakia. [43]Department of Scientific and Innovation Culture, Spanish Foundation for Science and Technology, Madrid, Spain. [44]Department of International and Political Sciences, University of Genoa, Genoa, Italy. [45]Institute of Multimedia and Interactive Systems, University of Lübeck, Lübeck, Germany. [46]Department of Health Law, Policy, and Management, Boston University School of Public Health, Boston, MA, USA. [47]Department of Sociology, University of the Philippines Diliman, Quezon City, Philippines. [48]Department of Psychology, Erzurum Technical University, Erzurum, Turkey. [49]Network for Economic and Social Trends, Western University, London, Ontario, Canada. [50]Behavior in Crisis Lab, Institute of Psychology, Jagiellonian University, Cracow, Poland. [51]Department of Psychology, KU Leuven, Leuven, Belgium. [52]Cambridge Zero, University of Cambridge, Cambridge, UK. [53]LP3C (Psychology Laboratory), Université Rennes 2, Rennes, France. [54]TRANSOC, Complutense University of Madrid, Madrid, Spain. [55]Department of Cognition, Emotion, and Methods in Psychology, Faculty of Psychology, University of Vienna, Vienna, Austria. [56]Institute of General Practice and Family Medicine, University of Bonn, University Hospital Bonn, Bonn, Germany. [57]School of Psychology, University of Kent, Canterbury, UK. [58]Harding Center for Risk Literacy, University of Potsdam, Potsdam, Germany. [59]Center for Sociocultural Research, HSE University, Moscow, Russia. [60]Department of Labor and Social Policy, University of Lodz, Lodz, Poland. [61]School of Psychological Science & Public Policy Institute, University of Western Australia, Perth, Western Australia, Australia. [62]Department of Management, Aarhus University, Aarhus, Denmark. [63]Department of Economics, Harvard University, Cambridge, MA, USA. [64]Department of Political Science & Annenberg School for Communication, University of Pennsylvania, Philadelphia, PA, USA. [65]Department of Computer Science and Engineering, University of Notre Dame, Notre Dame, IN, USA. [66]School of Communication and Culture, Aarhus University, Aarhus, Denmark. [67]a2i Programme of ICT Division and UNDP Bangladesh, Dhaka, Bangladesh. [68]Department of Resource Economics and Environmental Sociology, University of Alberta, Edmonton, Alberta, Canada. [69]Department of Political Science and International Relations, Carleton College, Northfield, MN, USA. [70]Faculty of Management and Economics, Ruhr-University Bochum, Bochum, Germany. [71]LMU Munich School of Management, LMU Munich, Munich, Germany. [72]Leibniz Institut für Wissensmedien, Tübingen, Germany. [73]School of Social Work, Lucerne University of Applied Sciences and Arts, Lucerne, Switzerland. [74]Department of Psychology, Saarland University, Saarbrücken, Germany. [75]Department of Political Science, Aarhus University, Aarhus, Denmark. [76]Department of Psychology, Universidad de Concepción, Concepción, Chile. [77]Faculty of Health Sciences, University of Bristol, Bristol, UK. [78]Institute for Management & Organization, Leuphana University, Lueneburg, Germany. [79]School of Psychology, University of New South Wales, Sydney, New South Wales, Australia. [80]UNSW Institute for Climate Risk & Response, University of New South Wales, Sydney, New South Wales, Australia. [81]Research Institute for Responsible Innovation, School of Management, University of St. Gallen, St. Gallen, Switzerland. [82]Department of Business Administration, Instituto Técnológico Autónomo de México, Mexico City, Mexico. [83]School of Psychology, Victoria University of Wellington, Wellington, New Zealand. [84]Department of Media and Communication, LMU Munich, Munich, Germany. [85]Department of Geography, University of Bergen, Bergen, Norway. [86]Centre for Climate and Energy Transformation, University of Bergen, Bergen, Norway. [87]Faculty of Data and Decision Sciences, Technion—Israel Institute of Technology, Haifa, Israel. [88]Hixon Center for Climate and the Environment, Harvey Mudd College, Claremont, CA, USA. [89]Faculty of Management, University of Warsaw, Warsaw, Poland. [90]Centro de Investigación y de Estudios Avanzados del Instituto Politícnico Nacional, Mexico City, Mexico. [91]Department of Media and Communication, City University of Hong Kong, Hong Kong, Hong Kong. [92]Department of Psychology, Eberhard Karls Universität Tübingen, Tübingen, Germany. [93]School of Education, Trinity College Dublin, Dublin, Ireland. [94]Department of Political Science and International Relations, KIMEP University, Almaty, Kazakhstan. [95]Center for Integrated Disaster Information Research, Interfaculty Initiative in Information Studies, University of Tokyo, Tokyo, Japan. [96]Department of Government & Politics, Jahangirnagar University, Dhaka, Bangladesh. [97]Department of Psychology, University of Minnesota, Minneapolis, MN, USA. [98]School of Geography, Planning, and Spatial Sciences, University of Tasmania, Tasmania, Australia. [99]Institute of Political Science, University of Bamberg, Bamberg, Germany. [100]Institute of Political Science and Sociology, University of Bonn, Bonn, Germany. [101]Institute of Psychology, Nicolaus Copernicus University, Toruń, Poland. [102]Graduate Institute of Journalism, National Taiwan University, Taipei, Taiwan. [103]Department of Public Health, University of Otago, Wellington, New Zealand. [104]Department of Journalism and Mass Communication, Lviv Polytechnic National University, Lviv, Ukraine. [105]Institute of Communication Studies and Journalism, Charles University, Prague, Czech Republic. [106]Center for Social and Cultural Psychology, Université Libre de Bruxelles, Brussels, Belgium. [107]Department of Political Science and International Relations, School of Sciences and Humanities, Nazarbayev University, Astana, Kazakhstan. [108]Department of Social Research, University of Turku, Turku, Finland. [109]Faculty of Life Sciences: Food, Nutrition and Health, University of Bayreuth, Kulmbach, Germany. [110]Department of Clinical and Health Psychology, Faculty of Psychology, University of Vienna, Vienna, Austria. [111]Division of Economics, Department of Management and Engineering, Linköping University, Linköping, Sweden. [112]Faculty of Polish and Classical Philology, University of Adam Mickiewicz, Poznań, Poland. [113]Department of Psychology, Ural Federal University, Yekaterinburg, Russia. [114]Department of Information Science and Media Studies, University of Bergen, Bergen, Norway. [115]Department of Communication Science and Political Science, Vrije Universiteit Amsterdam, Amsterdam, the Netherlands. [116]Faculty of Behavioural and Social Sciences, University of Groningen, Groningen, the Netherlands. [117]Department of Communication and Internet Studies, Cyprus University of Technology, Limassol, Cyprus. [118]Laboratoire Parisien de Psychologie Sociale, Université Paris Nanterre, Nanterre, France. [119]Laboratory for Research of Individual Differences, University of Belgrade, Belgrade, Serbia. [120]Department of Psychobiology and Methodology, Faculty of Psychology, Universidad Complutense de Madrid, Madrid, Spain. [121]School of Medicine and Psychology, Australian National University, Canberra, Australian Capital Territory, Australia. [122]Uehiro Centre for Practical Ethics, University of Oxford, Oxford, UK. [123]Institut Langage et Communication, University of Louvain, Louvain-la-Neuve, Belgium. [124]Departamento de Psicología, Universidad Peruana Cayetano Heredia, La Molina, Peru. [125]Department of Psychology,

University of Victoria, Victoria, British Columbia, Canada. [126]Harvard Kennedy School's Shorenstein Center, Harvard University, Cambridge, MA, USA. [127]Network Science Institute, Northeastern University, Boston, MA, USA. [128]School of Psychology and Public Health, La Trobe University, Melbourne, Victoria, Australia. [129]Department of Psychology, Royal Holloway, University of London, Egham, UK. [130]Institut Jean Nicod, Département d'études cognitives, ENS, EHESS, PSL University, CNRS, Paris, France. [131]Department of Communication, University of Muenster, Münster, Germany. [132]School of Psychological and Social Sciences, University of Waikato, Tauranga, New Zealand. [133]Department of Political Science and International Relations, University of Delaware, Newark, DE, USA. [134]Office for Quality Assurance, Analyses and Reporting, Project EUTOPIA, University of Ljubljana, Ljubljana, Slovenia. [135]Department of Management and Supply Chain Studies, Nkumba University, Entebbe, Uganda. [136]Department of Biochemistry and Molecular Biology, University of Buea, Buea, Cameroon. [137]Communication Arts Programme, Bowen University, Iwo, Nigeria. [138]Department of Psychology and Psychotherapy, Witten/Herdecke University, Witten, Germany. [139]Department of Management, University of Adger, Kristiansand, Norway. [140]Faculty of Humanities and Social Sciences, Trinity Western University, Langley, British Columbia, Canada. [141]Department of Clinical Neuroscience, Karolinska Institutet, Stockholm, Sweden. [142]Institute of Psychology, University of Silesia in Katowice, Katowice, Poland. [143]Penn Center for Neuroaesthetics, University of Pennsylvania, Philadelphia, PA, USA. [144]Institute of Medical Psychology, University of Tübingen, Tübingen, Germany. [145]Department of Psychological Science, Pomona College, Claremont, CA, USA. [146]School of Psychology, Aston University, Birmingham, UK. [147]Department of Psychology, University of Crete, Rethymno, Greece. [148]Science Studies Laboratory, University of Warsaw, Warsaw, Poland. [149]Department of Psychology, University of the Philippines Diliman, Quezon City, Philippines. [150]Max Planck Institute for Human Development, Berlin, Germany. [151]Social and Cognitive Neuroscience Laboratory, Mackenzie Presbyterian University, São Paulo, Brazil. [152]Department of Social Sciences, University of Hamburg, Hamburg, Germany. [153]Institut des Géosciences de l'Environnement, University Grenoble Alpes, CNRS, IRD, Grenoble-INP, Grenoble, France. [154]Institute of Environmental Health, Faculty of Medicine, University of Lisbon, Lisbon, Portugal. [155]NOVA Institute of Communication, NOVA University of Lisbon, Lisbon, Portugal. [156]Department of Implementation Research, Bernhard-Nocht-Institute for Tropical Medicine, Hamburg, Germany. [157]Centre for Language Studies, Radboud University Nijmegen, Nijmegen, the Netherlands. [158]Department of Nutritional Sciences, University of Vienna, Vienna, Austria. [159]Department of Cognitive Psychology, Universität Hamburg, Hamburg, Germany. [160]School of Politics and International Relations, Australian National University, Canberra, Australian Capital Territory, Australia. [161]Independent Researcher, Cairo, Egypt. [162]Département de Sociologie, Université Officielle de Bukavu, Bukavu, Democratic Republic of the Congo. [163]Faculté des Sciences Sociales, Université Catholique de Bukavu, Bukavu, Democratic Republic of the Congo. [164]Psychology Department, TED University, Ankara, Turkey. [165]Sociology Department, Vrije Universiteit Brussel, Brussels, Belgium. [166]Department of Psychology and Neuroscience, Boston College, Chestnut Hill, MA, USA. [167]Faculty of Political Science and Economics, Waseda University, Tokyo, Japan. [168]Department of Civil Law, Faculty of Law, University of Tirana, Milto Tutulani, Tirana, Albania. [169]Centre for the Politics of Feelings, University of London, London, UK. [170]Division of Public Policy, Hong Kong University of Science and Technology, Hong Kong, Hong Kong. [171]School of Psychology, University of Sussex, Falmer, UK. [172]Molecular Haematology and Immunogenetics Laboratory, Department of Medical Laboratory Science, Faculty of Health Sciences and Technology, College of Medicine, University of Nigeria Nsukka, Nsukka, Nigeria. [173]School of Arts, Media and Communiation, UCLan Cyprus, Pyla, Cyprus. [174]Department of Psychology, Education and Child Studies, Erasmus University Rotterdam, Rotterdam, the Netherlands. [175]Department of Psychology, New York University, New York, NY, USA. [176]Faculty of Technology and Bionics, Rhine-Waal University, Kleve, Germany. [177]Melbourne Centre for Behaviour Change, University of Melbourne, Melbourne, Victoria, Australia. [178]Faculty of Philosophy and Social Science, Nicolaus Copernicus University, Toruń, Poland. [179]School of Economics and Management, Tongji University, Shanghai, China. [180]School of Environment, Tsinghua University, Beijing, China. ✉e-mail: viktoriacologna@gmail.com

# Reporting Summary

## Statistics

For all statistical analyses, confirm that the following items are present in the figure legend, table legend, main text, or Methods section.

| n/a | Confirmed | |
|---|---|---|
| ☐ | ☒ | The exact sample size (*n*) for each experimental group/condition, given as a discrete number and unit of measurement |
| ☐ | ☒ | A statement on whether measurements were taken from distinct samples or whether the same sample was measured repeatedly |
| ☐ | ☒ | The statistical test(s) used AND whether they are one- or two-sided *Only common tests should be described solely by name; describe more complex techniques in the Methods section.* |
| ☐ | ☒ | A description of all covariates tested |
| ☐ | ☒ | A description of any assumptions or corrections, such as tests of normality and adjustment for multiple comparisons |
| ☐ | ☒ | A full description of the statistical parameters including central tendency (e.g. means) or other basic estimates (e.g. regression coefficient) AND variation (e.g. standard deviation) or associated estimates of uncertainty (e.g. confidence intervals) |
| ☐ | ☒ | For null hypothesis testing, the test statistic (e.g. *F*, *t*, *r*) with confidence intervals, effect sizes, degrees of freedom and *P* value noted *Give P values as exact values whenever suitable.* |
| ☒ | ☐ | For Bayesian analysis, information on the choice of priors and Markov chain Monte Carlo settings |
| ☐ | ☒ | For hierarchical and complex designs, identification of the appropriate level for tests and full reporting of outcomes |
| ☐ | ☒ | Estimates of effect sizes (e.g. Cohen's *d*, Pearson's *r*), indicating how they were calculated |

*Our web collection on statistics for biologists contains articles on many of the points above.*

## Software and code

Policy information about availability of computer code

| Data collection | Qualtrics software, versions 11-2022 throughout 08-2023. Qualtrics, Provo, UT, USA. |
|---|---|
| Data analysis | R v4.4.1 (2024-06-14 ucrt), platform: x86_64-w64-mingw32/x64 under Windows 11 x64 (build 22631). Name and version of packages: broom 1.0.7 broom.mixed 0.2.9.5 car 3.1-3 carData 3.0-5 corrplot 0.94 countrycode 1.6.0 data.table 1.16.0 datawizard 0.12.3 dplyr 1.1.4 forcats 1.0.0 ggalt 0.4.0 ggbump 0.1.0 ggdag 0.2.13 ggflags 0.0.4 ggplot2 3.5.1 ggpubr 0.6.0 GPArotation 2024.3-1 gridExtra 2.3 |

```
haven 2.5.4
Hmisc 5.1-3
insight 0.20.4
jtools 2.3.0
lavaan 0.6-19
lme4 1.1-35.5
lmerTest 3.1-3
lubridate 1.9.3
magrittr 2.0.3
maps 3.4.2
marginaleffects 0.22.0
Matrix 1.7-0
moments 0.14.1
MVN 5.9
optimx 2023-10.21
performance 0.12.3
psych 2.4.6.26
purrr 1.0.2
quantmod 0.4.26
readr 2.1.5
readxl 1.4.3
remotes 2.5.0
reshape2 1.4.4
rlang 1.1.4
rsample 1.2.1
rstudioapi 0.16.0
scales 1.3.0
sjlabelled 1.2.0
sjPlot 2.8.16
srvyr 1.3.0
stringr 1.5.1
survey 4.4-2
survival 3.6-4
tibble 3.2.1
tidyr 1.3.1
tidyverse 2.0.0
TTR 0.24.4
viridis 0.6.5
viridisLite 0.4.2
weights 1.0.4
xts 0.14.0
zoo 1.8-12
```

For manuscripts utilizing custom algorithms or software that are central to the research but not yet described in published literature, software must be made available to editors and reviewers. We strongly encourage code deposition in a community repository (e.g. GitHub). See the Nature Portfolio guidelines for submitting code & software for further information.

## Data

Policy information about availability of data

All manuscripts must include a data availability statement. This statement should provide the following information, where applicable:
- Accession codes, unique identifiers, or web links for publicly available datasets
- A description of any restrictions on data availability
- For clinical datasets or third party data, please ensure that the statement adheres to our policy

The dataset underlying this article is publicly available at: https://doi.org/10.17605/OSF.IO/5C3QD Mede46 provide detailed information on the dataset, including data collection and pre-processing.

## Research involving human participants, their data, or biological material

Policy information about studies with human participants or human data. See also policy information about sex, gender (identity/presentation), and sexual orientation and race, ethnicity and racism.

| Reporting on sex and gender | Gender was determined based on self-reporting. Participants were also given the option to select "Prefer not to say". |
|---|---|
| Reporting on race, ethnicity, or other socially relevant groupings | Race and ethnicity were not assessed in this study. All assessed socio-demographic variables were determined based on self-reporting. |
| Population characteristics | Participant's gender, age, education, income, place of residence, political orientation, and religiosity were determined based on self-reporting. |

| Recruitment | Data were collected in on line surveys that used quotas for age (five bins: 20% 18-29 years, 20% 30-39 years, 20% 40-49 l years, 20% 50-59 years, 20% 60 years and older) and gender (two bins: 50% male, 50% female). Participants had to be 18 years of age or older and provide informed consent to participate in the study. The surveys were programmed in Qualtrics. Participants that completed the survey were remunerated according to the market research company's local rates. All data was collected via on line surveys, except for the Democratic Republic of Congo, where participants were interviewed in face/to-face interviews and responses recorded in Qualtrics by the interviewers. Respondents were recruited from online panels of the market research companies Bilendi & respondi, MSi, Prolific, 2muse, and Kieskompas. They received vouchers/credit points for completing the full survey, which they could redeem and/or transfer into money. |
| --- | --- |
| Ethics oversight | Harvard University-Area Committee on the Use of Human Subjects (protocol# IRB22-1046). |

Note that full information on the approval of the study protocol must also be provided in the manuscript.

# Field-specific reporting

Please select the one below that is the best fit for your research. If you are not sure, read the appropriate sections before making your selection.

☐ Life sciences  ☒ Behavioural & social sciences  ☐ Ecological, evolutionary & environmental sciences

For a reference copy of the document with all sections, see nature.com/documents/nr-reporting-summary-flat.pdf

# Behavioural & social sciences study design

All studies must disclose on these points even when the disclosure is negative.

| Study description | This study uses quantitative data. They were collected in a global, pre-tested, pre-registered, cross-sectional on line survey (N = 71,922 participants in k = 68 countries) between November 2022 and August 2023 as part of the TISP Many Labs project ("Trust in Science and Science-Related Populism"). TISP is an international, multidisciplinary consortium of 241 researchers from 171 institutions across all continents. |
| --- | --- |
| Research sample | Researchers conducted on line surveys within 88 post-hoe weighted quota samples in 68 countries, using the same questionnaire translated into 37 languages. Data were collected in surveys that used quotas for age (five bins: 20% 18-29 years, 20% 30-39 years, 20% 40-49 years, 20% 50-59 years, 20% 60 years and older) and gender (two bins: 50% male, 50% female). Participants had to be 18 years of age or older and provide informed consent to participate in the study. Therefore the samples are not representative. We were interested in public attitudes towards scientists and their role in society and policy-making, hence we used samples of the general population. |
| Sampling strategy | To determine our minimum target sample size, we ran simulation-based power analyses using the R package simr (vl.0.7) which is designed to conduct power analyses for generalized linear mixed models such as those we used in the main study (for detailed information see SI). Based on these analyses we determined a minimum target sample size of 7,500, with n = 500 in k = 15 countries to detect fixed effects of trust in scientists and science-related populists as small as b = 0.10 and b = 0.05, respectively. Our final sample of 71,922 individuals with k = 68 countries is thus by far big enough to detect even smaller effects of trust in scientists and science-related populist attitudes. We used non-probability quota sampling, with balanced quotas for age (five bins: 20% 18-29 years, 20% 30-39 years, 20% 40-49 years, 20% 50-59 years, 20% 60 years and older) and gender (two bins: 50% male, 50% female). |
| Data collection | Data were collected in surveys that used quotas for age (five bins: 20% 18-29 years, 20% 30-39 years, 20% 40-49 years, 20% 50-59 years, 20% 60 years and older) and gender (two bins: 50% male, 50% female). Participants had to be 18 years of age or older and provide informed consent to participate in the study. The surveys were programmed in Qualtrics. Participants that completed the survey were remunerated according to the market research company's local rates. All data was collected via on line surveys, except for the Democratic Republic of Congo, where participants were interviewed in face-to-face interviews and responses recorded in Qualtrics by the interviewers. Participants were recruited with the market research company Bilendi & Respondi, except for most African countries, where data was collected with the market research company MSi.<br>The researchers were not blinded to the research questions and hypotheses. |
| Timing | Data were collected between November 2022 and August 2023. |
| Data exclusions | We excluded all respondents who did not complete the survey, because they cancelled participation during the survey, were filtered as their gender x age quota was already full, or because they did not pass one of the two attention checks. The TISP dataset contains complete records of N = 71,922 participants from 88 samples across k = 68 countries. Overall, we collected a total of N = 72,135 complete responses but had to delete 213 records from duplicate respondents. |
| Non-participation | We are not aware of how many participants that were invited to participate by the market research company declined participation. |
| Randomization | Participants were not allocated into experimental groups. |

# Reporting for specific materials, systems and methods

We require information from authors about some types of materials, experimental systems and methods used in many studies. Here, indicate whether each material, system or method listed is relevant to your study. If you are not sure if a list item applies to your research, read the appropriate section before selecting a response.

## Materials & experimental systems

| n/a | Involved in the study |
|-----|----------------------|
| ☒ ☐ | Antibodies |
| ☒ ☐ | Eukaryotic cell lines |
| ☒ ☐ | Palaeontology and archaeology |
| ☒ ☐ | Animals and other organisms |
| ☒ ☐ | Clinical data |
| ☒ ☐ | Dual use research of concern |
| ☒ ☐ | Plants |

## Methods

| n/a | Involved in the study |
|-----|----------------------|
| ☒ ☐ | ChIP-seq |
| ☒ ☐ | Flow cytometry |
| ☒ ☐ | MRI-based neuroimaging |

## Plants

| | |
|---|---|
| Seed stocks | *Report on the source of all seed stocks or other plant material used. If applicable, state the seed stock centre and catalogue number. If plant specimens were collected from the field, describe the collection location, date and sampling procedures.* |
| Novel plant genotypes | *Describe the methods by which all novel plant genotypes were produced. This includes those generated by transgenic approaches, gene editing, chemical/radiation-based mutagenesis and hybridization. For transgenic lines, describe the transformation method, the number of independent lines analyzed and the generation upon which experiments were performed. For gene-edited lines, describe the editor used, the endogenous sequence targeted for editing, the targeting guide RNA sequence (if applicable) and how the editor was applied.* |
| Authentication | *Describe any authentication procedures for each seed stock used or novel genotype generated. Describe any experiments used to assess the effect of a mutation and, where applicable, how potential secondary effects (e.g. second site T-DNA insertions, mosiacism, off-target gene editing) were examined.* |

