## [Peer Review File · Nature Human Behaviour]

Trust in scientists and their role in society across 68 countries

Corresponding Author: Dr Viktoria Cologna

This manuscript has been previously reviewed at another journal. This document only contains information relating to versions considered at Nature Human Behaviour.

Version 0:

Decision Letter:

6th March 2024

Dear Dr Cologna,

Thank you once again for your manuscript, entitled "Trust in scientists and their role in society across 67 countries", and for your patience during the peer review process.

Your Article has now been evaluated by 3 referees. You will see from their comments copied below that, although they find your work of potential interest, they have raised quite substantial concerns. In light of these comments, we cannot accept the manuscript for publication, but would be interested in considering a revised version if you are willing and able to fully address reviewer and editorial concerns.

We hope you will find the referees' comments useful as you decide how to proceed. If you wish to submit a substantially revised manuscript, please bear in mind that we will be reluctant to approach the referees again in the absence of major revisions. We are committed to providing a fair and constructive peer-review process. Do not hesitate to contact us if there are specific requests from the reviewers that you believe are technically impossible or unlikely to yield a meaningful outcome.

To guide the scope of the revisions, the editors discuss the referee reports in detail within the team, including with the chief editor, with a view to (1) identifying key priorities that should be addressed in revision and (2) overruling referee requests that are deemed beyond the scope of the current study. We hope that you will find the prioritised set of referee points to be useful when revising your study. Please do not hesitate to get in touch if you would like to discuss these issues further.

1. Reviewer 2 raises important concerns about the measurement of your dependent variable trust in scientists. We ask that you address these concerns and demonstrate the validity of your measurement.
2. Reviewer 3 raises some concerns about the regression analyses. We ask that you revise your manuscript in response to these concerns and follow the advice of the reviewer to allow for interpretability of the small effect sizes.
3. Our reviewers ask you to provide a clearer rationale for your study and revise the interpretation of your findings to be more specific and nuanced. Please ensure that all claims are supported by empirical evidence and avoid speculation.
4. We ask that you explicitly motivate all methodological and technical choices. Preregistered analyses should be reported in the manuscript and any deviations be described and explicitly motivated.

If you wish to submit a suitably revised manuscript, we would hope to receive it within 3 months. I would be grateful if you could contact us as soon as possible if you foresee difficulties with meeting this target resubmission date.

- Include a "Response to the editors and reviewers" document detailing, point-by-point, how you addressed each editor and referee comment. If no action was taken to address a point, you must provide a compelling argument. When formatting this document, please respond to each reviewer comment individually, including the full text of the reviewer comment verbatim followed by your

response to the individual point. This response will be used by the editors to evaluate your revision and sent back to the reviewers along with the revised manuscript.

- Highlight all changes made to your manuscript or provide us with a version that tracks changes.

Link Redacted

Thank you for the opportunity to review your work. Please do not hesitate to contact me if you have any questions or would like to discuss the required revisions further.

Sincerely,

[REDACTED]

Reviewer expertise:

Reviewer #1: trust in scientists

Reviewer #2: large-scale surveys ; social psychology ; (changes in) public opinion

Reviewer #3: trust in scientists

REVIEWER COMMENTS:

Reviewer #1:

Remarks to the Author:

This is an exceptionally conducted and reported piece of collaborative social science. There are many strengths to emphasize in this manuscript. I'll not list them all, but would like to especially acknowledge (1) the clearly articulated study rationale, (2) the expansive data set and associated analyses, (3) the high quality and accessible data visualizations, (4) the command of extant literature, and (5) the clear writing. What this paper represents -- a systematic, empirical, global, and timely assessment of the state of trust in science -- is, in my view, desperately needed and can help countervail (non-empirical) anecdotes that commonly mischaracterize and/or overstate levels of distrust in science and scientists.

I have only a handful of small considerations to share with the authors as they seek to revise their manuscript:

(1) P9, L323 - the authors refer to "historical failings of science" but I think this choice of wording is inaccurate and obscures the more historically correct characterization: science has perpetuated racism. The authors speak of the need for science/scientists to be more transparent as a means through which to engender increased societal trust. With that piece of advice in mind (which I strongly agree with), I encourage the authors to acknowledge more thoughtfully and honestly science's fraught history with race and to state that plainly in the literature review. Relatedly, listing this issue alongside other issues like mis/disinformation, conspiracy theories, science-related populist attitudes, etc. has the effect of unfairly and inappropriately minimizing it--science's complicated history with racism is a legitimate reason for trust deficits in ways that zany conspiracy theories are not. Beware of conveying a false equivalency.

(2) P24, L550 - I encourage the authors to expand this paragraph; readers will want to hear more from the authors regarding how/why "small minorities" of distrusting individuals/groups should be taken seriously and studied. I agree with the authors position that they should; I just don't think the authors sufficiently explain this contention.

(3) P24, L 55 - This sentence is awkward and needs a rewrite.

(4) I encourage the authors to acknowledge Dan Kahan's work on "cultural cognition." Specifically, I would like them to consider how/if that work connects to their findings, implications, and suggestions for future research.

Reviewer #2:

Remarks to the Author:

I am grateful for the opportunity to review this global study of trust in scientists. The study has some considerable strengths. First, its sheer size is very impressive as it covers 67 countries with samples of many hundred participants per country (in some places several thousand). Second, the survey included a rich set of outcome measures and predictor measures that allow many analyses to be performed to address how variables relate to each other. Third, the reporting is ambitious: 48 pages in the main paper and another 86 pages of supplementary material.

Despite these strengths, the paper left me disappointed. The introduction does not build up to a clear research question, hence no clear answer emerges from the study. My judgment is that for a paper lacking theory to fit in Nature Human Behaviour, it must at

least have a clear take-home message. To my eyes, this message is lacking.

Unfortunately, I do not think this problem can be remedied by rewriting the text. It seems to me there are weaknesses in the study design. For one thing, it is not clear why the authors choose to study trust in scientists rather than trust in science, which is the concept they first bring up as a background to their study.

To motivate the study, the authors write that “there is scant robust global comparative evidence on trust in science, and, in particular, the extent to which concern over a lack of trust in scientists is justified. This raises the risk of ill-informed science policies, and misconceptions about the state of science in society. These concerns call for a global study...” Apparently, the authors fear that there is a mismatch between the actual trust in scientists that people have and policy-makers’ beliefs about this trust. However, the study makes no attempt at documenting policy-makers’ beliefs about trust in scientists and the policies they consider based on such beliefs policy-makers are considering. Hence, the study does not tell us anything about the possible mismatch or its possible consequences.

It is a very expensive undertaking to conduct a global study. Therefore, it surprised me that there was no clear research question connected with this aspect of the study. The authors emphasize the study’s finding that the political left-right gradient in their measure of trust in scientists is inconsistent across countries but (1) I am not aware of any claim to the contrary, (2) a global study is not required to reach this conclusion as two countries would suffice, (3) it is well-known that in the 1970s the political left was less trusting in science than the political right in many Western countries, so we know already that there is no consistent gradient, and (4) no attempt is made to test any hypothesis about the cause of the current global variation.

My biggest concerns about the study are about the measure of trust in scientists, a 12-item scale aiming to cover four dimensions of trust, such as competence and openness. First, as the items are written it seems that they ask about scientists as individuals, not about their actions as scientists, which seems to be a much more relevant thing to ask about. Relatedly, I don’t understand how some of these items can even be said to address trust in scientists. In their scientific work, I do NOT trust scientists who are considerate of others’ interests and pay very much attention to others’ views, yet these items are counted as high trust. Similarly, in most scientific disciplines listed in the study’s definition of science (e.g., nature, physics, history), it is not beneficial to the scientist’s work to be very eager to improve others’ lives; other drives are preferable.

The conclusions of the study are weak. The first conclusion is based on the finding that scientists were rated lower on openness than on competence. The authors suggest that scientists therefore may gain more public trust by being more open (more receptive to feedback) - but that suggestion is only valid if we accept that it is desirable for scientists to maximize the scores on the authors’ scale. The study does not show that the public desires scientists to be more receptive to feedback, nor that it would be good for science itself. The authors’ second conclusion is that as the different dimensions of their scale yielded differences between and within countries they “demonstrate the importance of using multidimensional trust measures”. We are not told which finding actually demonstrates this importance or how it is important. The third conclusion is that, as the individual-level correlations between the measure of trust in scientists and other individual-level variables vary across countries, scholars should be cautious when generalizing findings from Western countries. As a general conclusion, this has been said uncountable times before. On the particular topic at hand, I did not see that the authors provided any evidence that scholars tend to make such unfounded generalizations.

In sum, I do not think this paper has a compelling message about trust in scientists. However, as the data is very rich, I encourage the authors to take my comments as an inspiration to think again about what their ambitious study can tell us. I note that the study included other data too, some of which is mentioned as sideshows in this paper and some of which is not analyzed here at all. As the main topic of the paper is trust in scientists, I do not discuss these aspects in this review.

Reviewer #3:

Remarks to the Author:

NB This review was conducted by a tenured faculty member and a graduate student. The faculty member takes full responsibility for the content of this review.

• Key results: Please summarise what you consider to be the outstanding features of the work.

This paper reports the findings of a large-scale international survey of public trust in science. The survey is outstanding for its international scope and the authors’ commitments to open science and transparency practices via the OSF preregistration and repository.

• Validity: Does the manuscript have flaws which should prohibit its publication? If so, please provide details.

We note a number of concerns with the regression models (under “Appropriate use of statistics” below) and interpretation of the findings (“Conclusion”). We recommend these issues be addressed before acceptance for publication.

• Originality and significance: If the conclusions are not original, please provide relevant references. On a more subjective note, do you feel that the results presented are of immediate interest to many people in your own discipline, and/or to people from several disciplines?

We are not aware of similar surveys of public trust in science on the same global scale, and consider the project highly original in that respect. The finding that trust in science is moderately high is particularly important, given a widespread misperception that there is a general crisis of trust in science. The results will be of immediate interest to scholars of public trust in science in a variety of fields, including philosophy of science, psychology, science communication, and so on.

- **Data & methodology:** Please comment on the validity of the approach, quality of the data and quality of presentation. Please note that we expect our reviewers to review all data, including any extended data and supplementary information. Is the reporting of data and methodology sufficiently detailed and transparent to enable reproducing the results?

The data and pre-processing and analysis scripts appear to be available in the osf.io repository. Because the repository is currently set to “view only” (downloading requires non-anonymous access), we did not attempt to reproduce the analysis. The project organization is perspicuous and the code itself is fairly easy to read. We believe that, with access to the data, it would be straightforward to attempt to computationally reproduce the analysis.

We do have a few grumbles about the code style. For future projects, we would encourage the authors to (1) use `lintr` or a similar style checking tool, and (2) embrace the DRY principle: Don’t Repeat Yourself. For example, `purrr::map()` calls should be used in place of copy-pasting.

We also note that the data does not include a codebook or standalone instructions for pre-processing and assembling the combined dataset. To facilitate re-use of the data, the authors might consider creating a separate repository, with just the data and appropriate metadata and instructions.

- **Appropriate use of statistics and treatment of uncertainties:** All error bars should be defined in the corresponding figure legends; please comment if that’s not the case. Please include in your report a specific comment on the appropriateness of any statistical tests, and the accuracy of the description of any error bars and probability values.

The authors use regression analysis to identify potential causes and effects of trust in science. We do not object to the use of non-experimental data as provisional, defeasible evidence for causal claims. But we do have a number of concerns about how these particular regression models are specified and interpreted.

First, the authors highlight statistically significant results that are of little or no practical significance. For example, they highlight global differences in trust associated with gender, age, urbanicity, income, religion, and education (lines 403-405), all of which are about 0.02-0.03 of a standard deviation. To prevent the overinterpretation of such very small effect sizes, we recommend adding marginal effects plots that use unstandardized (and therefore more interpretable) units.

Second, using a simple DAG would improve the specification and interpretation of the regression models. For one example, in the analysis of trust (table S2), a simple DAG might look like this:

country indicators  individual demographics  attitudes to science  trust

with additional direct effects running in the same direction, e.g., from country indicators to attitudes and trust. This DAG suggests that, for identifying the effects of demographics on trust, it is important to condition on country indicators (as potential confounders) but to not condition on attitudes (as potential mediators). This would explain why the coefficients on several demographic variables have substantial changes between block 1 and block 2. Or, for identifying the effects of country indicators, none of the other sets of covariates should be included (since they’re all potential mediators). Specifically, it seems plausible that individual-level education is a mediator between country literacy and education expenditure, on the one hand, and trust, on the other. So the lack of association between country literacy and education and trust (lines 471-472) is plausibly due to model misspecification error. Of the three models reported in S1, only block attitude (attitudes) is appropriately specified for analyzing the effects of the covariates of interest.

For another example where some simple causal analysis would be useful, in the analysis of research priorities (table S4), the difference between perceived and desired priorities is plausibly a cause, rather than an effect, of trust. If this is right, then conditioning on trust in the regression models introduces collider bias. At a minimum, the authors should compare regression models with and without trust to assess for the possibility of collider bias.

Third, the researchers measured several variables that do not appear to be included in any of their analyses. In particular, the social dominance orientation variable, which was one variable the researchers included hypotheses about in their pre-registration, is missing from their reported analyses on explanatory factors of trust in scientists. If this was a deliberate deviation from their pre-registered plans, an explanation should be included in their supplemental information alongside their explanations for their other deviations.

Fourth, the authors do not provide a rationale for their selection of demographic variables or the relationship between these variables and trust in scientists. In the absence of a plausible mechanism by which, for example, being older or being a woman disposes a person to trust scientists more than people who are younger or a man, this runs of risk of essentializing these demographic groups — especially since the authors highlight such relationships despite the very small effect sizes. We do think it is plausible that several of the demographic variables are convenient proxies for other constructs that are of more direct interest but difficult to measure. But, whatever the authors’ rationale, it should be noted in the methods.

- **Conclusions:** Do you find that the conclusions and data interpretation are robust, valid and reliable?

In places, the authors present a rather superficial interpretation of their findings.

For example, with regards to political orientation and trust in scientists, the authors’ analyses (figs. 3 and S5) show a positive association between trust in scientists and right-leaning political orientation in 8 countries – Georgia, Egypt, Philippines, Nigeria, Taiwan, Indonesia, Slovakia, and Greece – and the opposite association in 16 countries – China, Brazil, Austria, Israel, Canada, Norway, Italy, Germany, Netherlands, Portugal, Spain, Switzerland, United Kingdom, Belgium, Denmark, Sweden, and the United States. That is, twice as many countries show one relationship rather than the other; and the larger subset represents both a very

large proportion of the world's population and a majority of the world's scientific output. Thus, the consequences of a negative association between right-leaning political orientation and trust in scientists at both the country-level and worldwide level are likely quite different and more substantial than the opposite association. But the authors' conclusion here is simply that there is variation between Western and "non-Western" countries (lines 427-443). (Also, from the text on lines 825-830, it's unclear which covariates were included in these random slope models, and so whether this covariate selection was appropriate to estimate these effects.)

Second, the authors recommend that "scholars ... be cautious when generalizing findings from Western or Anglophone countries." (lines 558-9). It is always good to advise caution concerning the appropriate generalizability of findings, but are any scholars making such generalizations? References 7, 8, 9, 10, 19, 22 and 23 all clearly report their results in the context of variance across countries when discussing and summarizing their findings, and 20 and 21 do limit their discussions and interpretations to what their results mean in the context of the United States (the only country they sampled from). The authors' own literature review does not seem to indicate a problem of inappropriate generalizations on the part of scholars.

Third, the authors discuss disparities between participants' desired and perceived research priorities, and conclude that "greater consideration of public research priorities in scientific research and public funding presents an important avenue to increase trust" (lines 568-570). While we agree with this policy recommendation, a great deal of research that aligns with the desired priorities is already happening. There is substantial research to improve public health, solve energy problems, and reduce poverty. This suggests that the perception gap might primarily be a communications issue. So a more pointed recommendation would be expand and improve upon this aspect of science communication, to elevate the prominence such research has in the minds of members of the general public. This would be a specific recommendation that follows directly from the kind of general recommendation the authors suggest on lines 536-542. Alternatively, participants may be answering the question of how much science aims to tackle those goals by reframing it as what they perceive are actual policies to address those issues, and the reported gap is between what participants desire scientists should prioritize and what they perceive are actual tangible efforts to tackle those goals.

Suggested improvements: Please list additional experiments or data that could help strengthening the work in a revision. In the Methods subsection "Desired and perceived goals of science," it would be helpful if the authors listed the goals presented to participants.

- References: Does this manuscript reference previous literature appropriately? If not, what references should be included or excluded?

The authors report that only a minority of their sample expressed low trust in scientists, but their discussion on the potential consequences of even a minority lacking trust in scientists (lines 550-554) seems superficial. We (the co-reviewers) disagree among ourselves about the significance of minority distrust. But simulations have suggested that even a minority that is biased against science can have substantial consequences for the public's view on the scientific consensus (Lewandowsky et al., 2019; full citation below). We would encourage the authors to include this in the Discussion and Recommendations.

Lewandowsky, S., Pilditch, T. D., Madsen, J. K., Oreskes, N., & Risbey, J. S. (2019). Influence and seepage: An evidence-resistant minority can affect public opinion and scientific belief formation. *Cognition*, 188, 124-139.

- Clarity and context: Is the abstract clear, accessible? Are abstract, introduction and conclusions appropriate?

We find the writing to be clear and accessible throughout, and the abstract and introduction to be appropriate. Our concerns with the conclusion are discussed above.

- Inflammatory material: Does the manuscript contain any language that is inappropriate or potentially libelous?

We do not believe any of the language or claims made in the paper are inappropriate or potentially libelous.

- Springer Nature is committed to diversity, equity and inclusion; please raise any concerns that may in your view have an impact on this commitment.

We have no DEI-related concerns about the paper.

- Please indicate any particular part of the manuscript, data, or analyses that you feel is outside the scope of your expertise, or that you were unable to assess fully.

None to report.

- Please address any other specific question asked by the editor via email.

NA

Version 1:

Decision Letter:

12th July 2024

Dear Dr Cologna,

Thank you once again for your revised manuscript, entitled "Trust in scientists and their role in society across 68 countries," and for your patience during the re-review process.

Your manuscript has now been evaluated by Reviewers 1 and 3 from the original round of review as well as a new Reviewer 4 with expertise in large-scale surveys and trust research. All reviewer feedback is included at the end of this letter. Although the reviewers found your manuscript to have improved during revision, they also raise some important outstanding concerns. We remain interested in the possibility of publishing your study in Nature Human Behaviour, but would like to consider your response to these outstanding concerns in the form of a revised manuscript before we make a decision on publication.

1. Reviewer 4 mentions to find the work to lack a theoretical framework or rationale, resulting in the findings appearing to be disconnected. Please address these concerns in full and explain the practical or theoretical relevance of your findings.
2. Reviewer 4 suggests that your findings seem to largely overlap with the existing literature on trust in institutions in general. The reviewer asks that you explore this further. Please provide additional analyses and discussion to address this point.
3. Please follow Reviewer 4's suggestion and empirically test explanations for the result that country level inequality is associated with trust in scientists.
4. Reviewer 2 remains unconvinced by some aspects of your rebuttal. Carefully review the persisting concerns raised by Reviewer 2, and provide the requested analyses.

In sum, we invite you to revise your manuscript taking into account all reviewer and editor comments. We are committed to providing a fair and constructive peer-review process. Do not hesitate to contact us if there are specific requests from the reviewers that you believe are technically impossible or unlikely to yield a meaningful outcome.

We hope to receive your revised manuscript within 4-8 weeks. I would be grateful if you could contact us as soon as possible if you foresee difficulties with meeting this target resubmission date.

- Include a "Response to the editors and reviewers" document detailing, point-by-point, how you addressed each editor and referee comment. If no action was taken to address a point, you must provide a compelling argument. This response will be used by the editors and reviewers to evaluate your revision.
- Highlight all changes made to your manuscript or provide us with a version that tracks changes.

Link Redacted

We look forward to seeing the revised manuscript and thank you for the opportunity to review your work. Please do not hesitate to contact me if you have any questions or would like to discuss these revisions further.

Sincerely,

[REDACTED]

Reviewer expertise:

Reviewer #1: trust in scientists

Reviewer #3: trust in scientists

Reviewer #4: large-scale surveys, trust research

REVIEWER COMMENTS:

Reviewer #1:

Remarks to the Author:

The authors' revisions are thoughtful, thorough, and responsive and have ultimately made the manuscript even more robust. Thank you for your efforts to produce this highly salient research and for being so willing to grapple with the many insightful critiques shared by the reviewers.

Reviewer #3:

Remarks to the Author:

We do not feel that the following issue has been adequately addressed:

- Research priorities and trust: The issue here was that the authors had trust as a IV/cause of (perceived) research priority gaps, and we said this should be reversed. In the response, the authors reject this suggestion, giving two arguments: (a) trust is "a fundamental perception of science that shapes other such perceptions"; (b) making trust the DV in this model would deviate from the preregistration (17).

For (a), we don't know what the authors mean by a "fundamental perception." Whatever it means, it seems to entail that trust cannot be treated as a DV/effect in a regression model. But the authors use trust as a DV/effect in the figure 2 models. Notably, these models include a block of "attitudes to science," including the perceived benefit of science. We do not understand why trust can be treated as an effect of views on the perceived benefit of science, but not as an effect of perceived research priority gaps. Indeed, it seems to us like perceived research priority gaps are likely to be causally upstream from the perceived benefit of science: all else being equal, insofar as someone thinks science does not address important research priorities, then they will perceive the benefit of science to be lower.

For (b), the preregistration for RQ5 is phrased in terms of correlations, so swapping trust and priorities between the IV and DV sides of a regression model would not be a deviation.

We are satisfied on balance with the authors' changes for the following issues, but feel that the paper might benefit from some further revision:

- Social dominance orientation: The authors explain that SDO was omitted by accident, and it has now been included in, e.g., block 2 in the figure 2 models. But the finding — that high SDO is associated with low trust — isn't mentioned in the Discussion. We feel that this finding has the important implication that scientists and science communicators could increase different components of trustworthiness by challenging unjust hierarchies and social stratification.

- Rationale for demographics: The authors explain in the response that the chosen demographic variables plausibly "reflect shared direct or mediated experiences with scientists" (18). We recommend including this rationale in the text of the paper itself.

We are satisfied with the changes made in response to the following issues:

- Practical significance vs. statistical significance: The authors have noted in the text that the effects are very small, and included marginal effects plots in the supplement as we suggested.

- Using a DAG to select covariates: The authors have added a DAG in the supplement. It differs from the one that we suggested, and mostly justifies the order of blocks of covariates as used in the previous version. There's a slight modification, with "demographics" now split into "demographics" and "ideological views" blocks. While this isn't the DAG that we would use, we're satisfied that at least the authors' implicit DAG is now explicit, and the data will be available for others to conduct an alternative analysis if they'd like.

- Negative association between trust and right-leaning political orientation: Due to changes in the data and regression model specification, this point no longer applies.

- Cautioning against WEIRD generalizations: The authors have dropped the relevant sentence.

- Science communication on research priorities: The authors have added a sentence in line with our suggestion.

- List all the research goals: The authors have added this in line with our suggestion.

- Minority distrust: The authors have added the citation we suggested.

Reviewer #4:

Remarks to the Author:

I'm a new reviewer on this manuscript. The paper employs a large, global, research team to address the extent to which people across the world trust scientists. This is an ambitious data collection effort on an important topic. Below I list some things I think the authors could/should consider doing to improve the paper. They appear in the order in which they occurred in the manuscript, rather than by importance.

Before I get to these issues, however, I'll mention what I see as the most significant limitation of the paper, as currently written. After

reading the paper, I found it difficult to remember the answers to several of the paper's four key questions/goals. It *is* clear that trust in scientists is reasonably high (higher than popular wisdom would suggest) and that there are country-level differences in trust in scientists. The authors do a good job communicating these things. What is difficult to hang on to (or recall) after reading the paper, however, are answers to the other questions – perhaps especially Question/Goal 2, which is “How do demographic, ideological, attitudinal, and country-level factors relate to trust in scientists and how do these relationships vary between countries?” After reading the paper, I tried to write down the answer(s) to this question and was mostly at a loss. The reason is that the paper is highly descriptive and does not have much in the way of theory – or even tentative explanations– to make sense of any given result, much less anything that ties the many results together into a coherent framework or narrative. As a result, much of the Results section comes off as a laundry list of largely disconnected findings.

To be clear, I am not suggesting that the authors need to begin with some overarching theory – I doubt any such theory exists for what they are trying to explain. Rather, I am encouraging the authors to avoid simply tossing out result after result and propose a (possible) explanation or interpretation of any given result or set of results that they present. This will allow readers to connect the findings to some framework and take away - actually remember - more from the paper. I think this is important to do, even if it means some findings need to be relegated to an SI to make space. Otherwise, the answers to the key questions motivating the paper are muddled or unclear. Several of the specific points below touch on this more general point.

i) P. 11. I know the authors added the “why do we study scientists rather than science” to address one of the previous reviewer's concerns. But the new discussion (on p. 11) seems out of place and will probably appear to readers like it comes out of nowhere. Either placing this discussion later - or, perhaps even better, after the next paragraph that outlines the research questions - would be more logical.

ii) Likewise, the authors write “However, research has shown that it makes a substantial difference whether one is being asked about trust in science, scientists, or scientific methods.” Readers will probably wonder how it makes a difference. After all, the types of differences it makes and the extent of those differences will have important implications for what readers can/should take away from this paper, given that the authors measured trust in scientists rather than science, scientific methods, etc. A brief clarification and explanation would help better situate the authors' contributions.

iii) I don't think readers can understand from the authors' text why they favor the directed relationships in figure S1 (indeed, the reviewer that prompted the addition of that figure proposed an altogether different DAG). In their responses to that reviewer, the authors point to some rationale for their favored DAG (though I personally did not find that rationale very convincing), but no rationale at all is given in the paper. If there is a good argument for these directed relationships over others, the authors should spell them out (even if happens in the SI). Otherwise, I don't think this figure is particularly useful and will probably only confuse readers.

iv) Given that NHB papers should be readable by educated non-specialists, in the introduction, the authors should clarify/explain two key concepts in their research questions, namely “people's normative perceptions of scientists in society and policymaking” and “people's perceived research priorities.” (I also think the latter issue could be briefly motivated/anticipated a bit better, as it seems to come out of nowhere.)

v) On p. 12 the authors introduce four dimensions of trust in science but then (as far as I can tell) this distinction is immediately dropped in the next paragraph (without warning or explanation) and isn't revisited again until the Discussion. I'd encourage the authors to i) warn readers that they are collapsing them and explain why, and ii) tell us whether the main results from the aggregate measure (largely) hold up with the individual dimensions.

vi) There is a large literature on trust in institutions in general. Just eyeballing Figure 1, it seems likely that the results would overlap a great deal with international differences in trust in institutions. Shouldn't the authors address it? Even if the authors did not measure it, they could perhaps correlate it with country-level responses from the World Values Survey or elsewhere. Even if this is not possible, it may warrant a discussion of whether the authors view country level differences in (dis)trust of science/scientists as part of a more general (dis)trust of institutions more broadly.

vii) As mentioned earlier, following Reviewer 2's comments, this paper is at times extremely descriptive. I think I view this as somewhat less problematic than R2's comments seem to suggest. But the results section goes through many findings, often without couching those findings in broader theoretical terms (just as one example, see the discussion of country-level determinants of how religiosity is associated with trust in science on pp. 15-16). Without some context or explanation, it makes it harder for reader to recall or “hold on” to particular findings. I'm not suggesting the paper needs an all-encompassing theory – rather just basic theoretical explanation of the (likely) cause of a given pattern of results they present. Without this, the paper often comes off as a long list of largely disconnected findings.

viii) Related to the previous point, the authors should explain the concepts underlying the two different measures of political orientation. The authors devote quite a bit of space (pp. 17-18) to how one measure matters (in some way) in some countries and the other measure matters more (in some way) in other countries, without ever explaining the underlying constructs (and how the measures, e.g., the left/right distinction make sense in a global context). As a result, that entire discussion doesn't make much sense.

ix) The authors do quickly entertain several explanations for the (somewhat surprising) result that country level inequality is associated with trust in scientists. But couldn't both these explanations (percent urban population in a given country and corruption perceptions) be tested with their data? (The authors point to Figure S9 as suggestive evidence for the corruption explanation, but I don't see how eyeballing the Figure can actually tell us anything about this explanation. Why not test it statistically and put that test in the SI?)

x) The authors motivate the paper and then conclude with the point that the media and anecdotal evidence suggest that trust in scientists is very low (and imply that this paper offers a corrective to that misperception). This paper offers somewhat surprising results, from the perspective of these popular perceptions. But as the introduction notes, there are prior cross-cultural studies of trust in science and scientists. It may be useful to note in the Discussion whether the findings of this paper are also surprising from prior scientific work on this. If not, this is important to know (since as the authors note, this paper offers design improvements over prior work and employs a larger sample).

Version 2:

Decision Letter:

Our ref: NATHUMBEHAV-24010048B

12th September 2024

Dear Dr. Cologna,

Thank you for submitting your revised manuscript "Trust in scientists and their role in society across 68 countries" (NATHUMBEHAV-24010048B). It has now been seen by the original referees and their comments are below. As you can see, the reviewers find that the paper has improved in revision. We will therefore be happy in principle to publish it in Nature Human Behaviour, pending revisions to comply with our editorial and formatting guidelines.

We are now performing detailed checks on your paper and will send you a checklist detailing our editorial and formatting requirements within two weeks. Please do not upload the final materials and make any revisions until you receive this additional information from us.

Sincerely,

[REDACTED]

Reviewer #3 (Remarks to the Author):

We are satisfied with the changes made by the authors in response to our feedback, and are happy to recommend moving the paper forward.

Reviewer #4 (Remarks to the Author):

I think the authors did a really good job addressing my comments. I have no further comments or suggestions.

Version 3:

Decision Letter:

Dear Dr Cologna,

We are pleased to inform you that your Article "Trust in scientists and their role in society across 68 countries", has now been accepted for publication in Nature Human Behaviour.

Please note that *Nature Human Behaviour* is a Transformative Journal (TJ). Authors may publish their research with us through the traditional subscription access route or make their paper immediately open access through payment of an article-processing charge (APC). Authors will not be required to make a final decision about access to their article until it has been accepted. [Find out more about Transformative Journals](https://www.springernature.com/gp/open-research/transformative-journals)

Acceptance of your manuscript is conditional on all authors' agreement with our publication policies (see

<http://www.nature.com/nathumbehav/info/gta>). In particular your manuscript must not be published elsewhere and there must be no announcement of the work to any media outlet until the publication date (the day on which it is uploaded onto our web site).

With best regards,

[REDACTED]

P.S. Click on the following link if you would like to recommend Nature Human Behaviour to your librarian <http://www.nature.com/subscriptions/recommend.html#forms>

** Visit the Springer Nature Editorial and Publishing website at http://editorial-jobs.springernature.com?utm_source=ejp_NHumB_email&utm_medium=ejp_NHumB_email&utm_campaign=ejp_NHumB for more information about our career opportunities. If you have any questions please click [here](mailto:editorial.publishing.jobs@springernature.com).

Open Access This Peer Review File is licensed under a Creative Commons Attribution 4.0 International License, which permits use, sharing, adaptation, distribution and reproduction in any medium or format, as long as you give appropriate credit to the original author(s) and the source, provide a link to the Creative Commons license, and indicate if changes were made. In cases where reviewers are anonymous, credit should be given to 'Anonymous Referee' and the source. The images or other third party material in this Peer Review File are included in the article's Creative Commons license, unless indicated otherwise in a credit line to the material. If material is not included in the article's Creative Commons license and your intended use is not permitted by statutory regulation or exceeds the permitted use, you will need to obtain permission directly from the copyright holder.

EDITOR COMMENTS:

To guide the scope of the revisions, the editors discuss the referee reports in detail within the team, including with the chief editor, with a view to (1) identifying key priorities that should be addressed in revision and (2) overruling referee requests that are deemed beyond the scope of the current study. We hope that you will find the prioritised set of referee points to be useful when revising your study. Please do not hesitate to get in touch if you would like to discuss these issues further.

1. Reviewer 2 raises important concerns about the measurement of your dependent variable trust in scientists. We ask that you address these concerns and demonstrate the validity of your measurement.
2. Reviewer 3 raises some concerns about the regression analyses. We ask that you revise your manuscript in response to these concerns and follow the advice of the reviewer to allow for interpretability of the small effect sizes.
3. Our reviewers ask you to provide a clearer rationale for your study and revise the interpretation of your findings to be more specific and nuanced. Please ensure that all claims are supported by empirical evidence and avoid speculation.
4. We ask that you explicitly motivate all methodological and technical choices. Preregistered analyses should be reported in the manuscript and any deviations be described and explicitly motivated.

We thank the editors for discussing the referee reports in detail and providing us with a prioritized set of referee points. We were pleased to see that especially Reviewers 1 and 3, who indicated to have expertise in trust in scientists, evaluated our manuscript very positively. We also appreciate the helpful methodological advice of Reviewer 2. We addressed all reviewer comments with great care, which we believe has improved the quality of our manuscript. For example, we considered Reviewer 2's concerns about the measurement of trust in scientists by providing information on the theoretical and empirical validity of our measure. We also made considerable changes to our Introduction to provide a clearer rationale for our study and described the contribution of our study and the interpretation of our findings in a more nuanced way. We carefully addressed Reviewer 3's concerns, by rerunning our regression analyses to include an additional measure as highlighted by the reviewer.

Please note that the revised version includes data from one additional country, Cameroon, for which IRB approval had been pending but not yet granted when we first submitted the article. We added our collaborators from Cameroon to the author list, changed the manuscript file, re-ran all analyses, and updated all tables and figures in the main text and supplementary information accordingly. We also made considerable changes to the supplementary information to reduce overlap with the dataset descriptor

article currently under review at Scientific Data as per the request of the Scientific Data editor. The supplementary information now focuses on the subset of the TISP dataset used for the analyses presented in this article. All removed information is still available in the public preprint of the dataset descriptor article: <https://osf.io/preprints/psyarxiv/jktsy>

If you wish to submit a suitably revised manuscript, we would hope to receive it within 3 months. I would be grateful if you could contact us as soon as possible if you foresee difficulties with meeting this target resubmission date.

Reviewer expertise:

Reviewer #1: trust in scientists

Reviewer #2: large-scale surveys; social psychology; (changes in) public opinion

Reviewer #3: trust in scientists

REVIEWER COMMENTS:

Reviewer #1:

Remarks to the Author:

This is an exceptionally conducted and reported piece of collaborative social science. There are many strengths to emphasize in this manuscript. I'll not list them all, but would like to especially acknowledge (1) the clearly articulated study rationale, (2) the expansive data set and associated analyses, (3) the high quality and accessible data visualizations, (4) the command of extant literature, and (5) the clear writing. What this paper represents -- a systematic, empirical, global, and timely assessment of the state of trust in science -- is, in my view, desperately needed and can help countervail (non-empirical) anecdotes that commonly mischaracterize and/or overstate levels of distrust in science and scientists.

We thank the reviewers for their very positive evaluation of our work, which was very encouraging to us.

I have only a handful of small considerations to share with the authors as they seek to revise their manuscript:

(1) P9, L323 - the authors refer to "historical failings of science" but I think this choice of

wording is inaccurate and obscures the more historically correct characterization: science has perpetuated racism. The authors speak of the need for science/scientists to be more transparent as a means through which to engender increased societal trust. With that piece of advice in mind (which I strongly agree with), I encourage the authors to acknowledge more thoughtfully and honestly science's fraught history with race and to state that plainly in the literature review. Relatedly, listing this issue alongside other issues like mis/disinformation, conspiracy theories, science-related populist attitudes, etc. has the effect of unfairly and inappropriately minimizing it--science's complicated history with racism is a legitimate reason for trust deficits in ways that many conspiracy theories are not. Beware of conveying a false equivalency.

Thank you for drawing our attention to these important considerations. We agree that the role of science in perpetuating racism is a legitimate reason for low trust and that this should not be listed alongside other issues like mis/disinformation. We added the following sentence on page 9:

“While science and scientists are generally held in high esteem, there are legitimate reasons for low trust among certain population groups and countries. For example, science’s fraught historical relationship with racism, its role in perpetuating racialized forms of knowledge production, sustaining racial paradigms¹¹, and disregarding ethical canons by experimenting on non-White human subjects¹², has reduced research participation in some populations¹³. Further, the epistemic and cultural authority of science and scientists has been challenged by mis- and disinformation^{14,15}, an alleged “reproducibility crisis”¹⁶, conspiracy theories^{17,18}, and science-related populist attitudes^{19,20}.”

(2) P24, L550 - I encourage the authors to expand this paragraph; readers will want to hear more from the authors regarding how/why "small minorities" of distrusting individuals/groups should be taken seriously and studied. I agree with the authors position that they should; I just don't think the authors sufficiently explain this contention.

Thank you for mentioning this. Reviewer 3 also found our explanation insufficient, so we added the following:

“When even reasonably small minorities of the public distrust science, it can have a considerable impact on society, especially if they receive extensive news media coverage and include people in positions of power that can influence policymaking^{54–56}. Studies have also shown that a minority of 10% can be sufficient to flip a majority⁵⁷ and that when a critical mass value of 25% is reached, majority opinion could be tipped⁵⁸. In the context of climate change, an agent-based model showed that an evidence-resistant minority can delay the process of public opinion converging with the scientific consensus⁵⁹.”

(3) P24, L 55 - This sentence is awkward and needs a rewrite.

We completely agree, thank you for spotting this. We rewrote the sentence as follows:

“Not only the levels of trust but also its correlates—such as right-leaning and conservative political orientation, education, and religiosity—vary clearly across countries.”

(4) I encourage the authors to acknowledge Dan Kahan's work on "cultural cognition." Specifically, I would like them to consider how/if that work connects to their findings, implications, and suggestions for future research.

We thank the reviewer for their comment and their suggestion to refer to the literature on cultural cognition. We gave this considerable thought, and after careful consideration, we decided not to engage with this literature on cultural cognition. First, we suspect that introducing another concept—which, in this case, has received substantial conceptual criticism (e.g., van der Linden, 2016)—would weaken the coherence and comprehensibility of the narrative. Second, we consider the cultural cognition literature less useful in global comparative research given several failed replication attempts of cultural cognition. As the literature stands now, Dan Kahan’s studies on cultural cognition did not, or only weakly, replicate across 23 countries, both in the US and Europe. See for example:

Pröpper, H. Y. L., Geiger, S., Blanken, T. F., & Brick, C. (2022). Truth over identity? Cultural cognition weakly replicates across 23 countries. *Journal of Environmental Psychology*, 83, 101865. <https://doi.org/10.1016/j.jenvp.2022.101865>

Stagnaro, M. N., Tappin, B. M., & Rand, D. G. (2023). No association between numerical ability and politically motivated reasoning in a large US probability sample. *Proceedings of the National Academy of Sciences*, 120(32), e2301491120. <https://doi.org/10.1073/pnas.2301491120>

Connor, P., Sullivan, E., Alfano, M., & Tintarev, N. (2020). Motivated Numeracy and Active Reasoning in a Western European Sample. *Behavioral Public Policy*, 1. <https://philarchive.org/rec/CONMNA>

Van der Linden, S. (2016). A Conceptual Critique of the Cultural Cognition Thesis. *Science Communication*, 38(1), 128–138. <https://doi.org/10.1177/1075547015614970>

Reviewer #2:

Remarks to the Author:

I am grateful for the opportunity to review this global study of trust in scientists. The study has some considerable strengths. First, its sheer size is very impressive as it covers 67 countries with samples of many hundred participants per country (in some places several thousand). Second, the survey included a rich set of outcome measures and predictor measures that allow many analyses to be performed to address how variables relate to each other. Third, the reporting is ambitious: 48 pages in the main paper and another 86 pages of supplementary material.

We thank the reviewers very much for their positive feedback. We also appreciate the comprehensive, constructive criticism.

Despite these strengths, the paper left me disappointed. The introduction does not build up to a clear research question, hence no clear answer emerges from the study. My judgment is that for a paper lacking theory to fit in Nature Human Behaviour, it must at least have a clear take-home message. To my eyes, this message is lacking.

We agree that the research questions could have been mentioned more clearly in the introduction. We restructured parts of the introduction and now include the explicit research questions, which provides readers with a precise understanding of the contribution and take-home messages:

“Specifically, our study seeks to answer the following research questions: 1) How much do people around the world trust scientists, and how do levels of trust vary across countries? 2) How do demographic, ideological, attitudinal, and country-level factors (i.e., income inequality) relate to trust in scientists, and how do these relationships vary between countries? 3) What are people’s normative perceptions of scientists in society and policymaking, and how do they differ across countries? 4) What are people’s desired research priorities, and do they believe that scientists actually address these priorities? See the preregistration for more detailed research questions and hypotheses: https://osf.io/9ksrj/?view_only=065f36b1340648209062765c0a6f091d”

Our take-home message appears in the first sentence of the Discussion: “Our global, 68-country survey challenges concerns of a widespread lack of public trust in scientists. In most countries surveyed, scientists and scientific methods are trusted.”

We suggest this is an important and clear take-home message. There are many concerns about distrust in scientists, but in this study, we do not find widespread lack of trust in scientists. However, as argued above, distrust by even small minorities should be taken seriously. The importance of this take-home message is also acknowledged by the other two reviewers:

Reviewer 1: “What this paper represents -- a systematic, empirical, global, and timely

assessment of the state of trust in science -- is, in my view, desperately needed and can help countervail (non-empirical) anecdotes that commonly mischaracterize and/or overstate levels of distrust in science and scientists.”

Reviewer 3: “The finding that trust in science is moderately high is particularly important, given a widespread misperception that there is a general crisis of trust in science.”

Unfortunately, I do not think this problem can be remedied by rewriting the text. It seems to me there are weaknesses in the study design. For one thing, it is not clear why the authors choose to study trust in scientists rather than trust in science, which is the concept they first bring up as a background to their study.

We agree that the previous draft was not precise enough with the reference object of trust, i.e. “science” or “scientists”. This study intentionally focused on scientists for the reasons explained below, so we revised the Introduction accordingly. Some confusion may have arisen because previous studies use these terms interchangeably and inconsistently. For example, the previous most comprehensive global study on the topic—conducted by the Wellcome Global Monitor—also assessed trust in scientists, but often refers to trust in science when reporting their results. Other studies also similarly assessed trust in scientists, but then report it as trust in science.

We chose to measure trust in scientists because:

1) Science is a vague reference object of trust: People may think of scientific institutions, scientific methods, or individual scientists when being asked about their general perception of “science”. However, research has shown that it makes a substantial difference whether one is being asked about trust in science, scientists, or scientific methods (e.g., Hoogeveen et al. 2022; Achterberg et al., 2018). We reduced this ambiguity by avoiding the abstract category “science” and using the more concrete and graspable reference object “scientists”.

2) This approach to studying trust in scientists stems from an organizational science perspective, which focuses on attributes typically ascribed to individuals (e.g., being open to feedback) rather than to complex societal systems like the scientific enterprise as a whole. Operationalizing this perspective thus requires using scientists as a reference object rather than science.

3) We argue that assessing trust in scientists (rather than trust in science more generally) provides more societally relevant implications. If one were to study trust in science and found trust in science to be low, the implications of this finding would not be clear given the abstract nature of the measure and multitude of facets of science that could potentially explain this low trust (e.g., low trust in scientific methods, scientists, or funding sources). On the other hand, the current measure of trust in scientists, which is based on four established dimensions of trustworthiness perceptions, allows us to draw societally relevant implications because the reasons for low or high trust in scientists can be analyzed and understood in detail.

4) The previous most comprehensive global study on the topic, conducted by the Wellcome Global Monitor, also assessed trust in scientists.

In sum, measuring trust in scientists instead of trust in science is in line with previous authoritative global studies on the topic and established conceptual approaches to studying trust.

To make it clear to our readers why we assessed trust in scientists, we added the following sentences on page 11:

“We decided to measure trust in scientists instead of trust in science because science is a vague reference object of trust: People may think of scientific institutions, scientific methods, or individual scientists when being asked about their general perception of “science”. However, research has shown that it makes a substantial difference whether one is being asked about trust in science, scientists, or scientific methods^{34,35}. We reduced this ambiguity by avoiding the abstract category “science” and using the more concrete and graspable reference object “scientists” as has been done in a previous global study⁷.”

References:

Achterberg, P., de Koster, W., & van der Waal, J. (2017). A science confidence gap: Education, trust in scientific methods, and trust in scientific institutions in the United States, 2014. *Public Understanding of Science*, 26(6), 704–720. <https://doi.org/10.1177/0963662515617367>

Hoogeveen, S., et al. (2022). The Einstein effect provides global evidence for scientific source credibility effects and the influence of religiosity. *Nature Human Behaviour*, 6(4), 523–535. <https://doi.org/10.1038/s41562-021-01273-8>

To motivate the study, the authors write that “there is scant robust global comparative evidence on trust in science, and, in particular, the extent to which concern over a lack of trust in scientists is justified. This raises the risk of ill-informed science policies, and misconceptions about the state of science in society. These concerns call for a global study...” Apparently, the authors fear that there is a mismatch between the actual trust in scientists that people have and policy-makers’ beliefs about this trust. However, the study makes no attempt at documenting policy-makers’ beliefs about trust in scientists and the policies they consider based on such beliefs policy-makers are considering. Hence, the study does not tell us anything about the possible mismatch or its possible consequences.

Yes, we agree with this disconnect the reviewer helpfully identified. To avoid confusion, we deleted parts of this paragraph, which now reads:

“However, there is scant robust global comparative evidence on trust in scientists, and in particular the extent to which concern over a lack of trust in scientists is justified. Indeed, there are only a few global studies on trust in scientists, and they have significant limitations^{6–10,21,22}.”

It is a very expensive undertaking to conduct a global study. Therefore, it surprised me that there was no clear research question connected with this aspect of the study. The authors emphasize the study’s finding that the political left-right gradient in their measure of trust in scientists is inconsistent across countries but (1) I am not aware of any claim to the contrary, (2) a global study is not required to reach this conclusion as two countries would suffice, (3) it is well-known that in the 1970s the political left was less trusting in science than the political right in many Western countries, so we know already that there is no consistent gradient, and (4) no attempt is made to test any hypothesis about the cause of the current global variation.

Thank you for these comments. Given the large financial investment and organizational effort involved in this research, we did preregister our research questions and a detailed protocol in a formal pre-registration (https://osf.io/9ksrj/?view_only=065f36b1340648209062765c0a6f091d). Based on revisions noted above, the new draft also more explicitly describes the research questions, which do connect to the global scope of the study. For example, they ask for differences in levels of trust between countries, and for reasons why trust may differ between countries. We investigated these differences by comparing means of trust between countries and testing how the relationship between trust and political orientation varies globally.

With regards to your four points:

(1 & 2) A global investigation is needed because there is no comparative evidence on how the relationship between political orientation and trust in scientists varies across countries, especially in countries outside of Europe and the US. Another important finding in this context is that in many countries, political orientation is not related to trust in scientists. However, based on the state of the field that is mostly based on the US and European countries, it would be plausible to assume that political orientation is a key explanatory factor for trust in scientists.

(3) This paper does not cover the historical development of the relationship between trust in scientists and political orientation, nor does it argue that these effects are or will be robust over time. This variance is why, in our view, it is important to analyze this relationship simultaneously across countries and at a point in history where trust in scientists is critical (i.e., the management of the COVID-19 pandemic and climate change). Being aware of whether and how participants’ political orientation in certain countries relates to trust in scientists is important for stakeholders engaging with the public.

(4) While an analysis of the cause of this global variation would certainly be interesting, it is beyond the scope of this article. To provide readers with more detailed and nuanced information on these causes within specific contexts, we discuss the relationship between left-leaning political orientation and lower trust in scientists in selected countries in the Supplementary Information on page 20. We also encourage future research to investigate the cause of global variations for political orientation and other independent variables included in our models.

My biggest concerns about the study are about the measure of trust in scientists, a 12-item scale aiming to cover four dimensions of trust, such as competence and openness. First, as the items are written it seems that they ask about scientists as individuals, not about their actions as scientists, which seems to be a much more relevant thing to ask about. Relatedly, I don't understand how some of these items can even be said to address trust in scientists. In their scientific work, I do NOT trust scientists who are considerate of others' interests and pay very much attention to others' views, yet these items are counted as high trust. Similarly, in most scientific disciplines listed in the study's definition of science (e.g., nature, physics, history), it is not beneficial to the scientist's work to be very eager to improve others' lives; other drives are preferable.

These are relevant concerns, and we thank the reviewer for sharing them. We agree that one may usefully measure trust in scientists by capturing perceptions of their actions and behaviors rather than perceptions of them as individuals. However, this trust measure focuses on perceptions of the characteristics of scientists as individuals. After all, these perceptions fundamentally inform whether one decides to trust someone beyond their actions. Reif and Guenther (2023, p. 96), in their instructive review article, elaborate: The decision to trust is "based on certain characteristics for which subjects of (dis)trust use heuristics or information shortcuts (Marques et al., 2015; Metzger & Flanagan, 2013). Together, characteristics and heuristics help to evaluate the trustworthiness or credibility of objects of (dis)trust (Bentele, 1998; Luhmann, 2014)."

That being said, we do include a three-item measure for behavioral trust in scientists in our survey, operationalized as the willingness to make oneself vulnerable to scientists' actions, which is included as a covariate in the models.

For multiple reasons, we are confident that our measures of trust in scientists are valid and reliable, and consistent with the state of the science. We designed this trust measure with a team of world-leading experts on trust in scientists and relied on one of the most comprehensive literature reviews of trust measures used to assess perceptions of scientists (Besley et al., 2021). Besley et al. (2021) reviewed articles published in five key journals over the last 5 years to gather all scales used to measure trust, credibility, and fairness perceptions of scientists. With this approach, the authors extracted 58 trust measures used to assess perceptions of scientists. The authors then ran a confirmatory

factor analysis that clearly distinguished four factors: competence, integrity, benevolence, and openness. Note that other research had also distinguished (some of) these factors before the study by Besley et al. (2021), see for example Hendriks et al. (2015) and, very recently, Reif et al. (2023). The findings by Besley et al., (2021) are also consistent with the other common approaches to measuring trustworthiness/credibility in domains other than science, see:

Schoorman, F. D., Mayer, R. C., & Davis, J. H. (2007). An integrative model of organizational trust: Past, present, and future. *The Academy of Management Review*, 32(2), 344-354. <https://doi.org/10.2307/20159304>

McCroskey, J. C., & Teven, J. J. (1999). Goodwill: A reexamination of the construct and its measurement. *Communication Monographs*, 66(1), 90-103. <https://doi.org/10.1080/03637759909376464>

Reif, A., Taddicken, M., Guenther, L., Schröder, J. T., & Weingart, P. (2023). The Public Trust in Science Scale (PuTS): A multilevel and multidimensional approach. <https://doi.org/10.31219/osf.io/bp8s6>

Our data support the four-factor model of competence, integrity, benevolence, and openness: the exploratory and confirmatory factor analyses (see <https://osf.io/preprints/psyarxiv/jktsy>) largely confirm this structure. Moreover, the validity of the trust measure is supported by responses to the open-ended question in which participants were asked to respond to before the trust items: “In your opinion, what makes a scientist trustworthy?”. While an in-depth analysis of open-ended survey responses is beyond the scope of this article, many participants mentioned individual characteristics of scientists that could be classified in benevolence and openness.

Regarding the second point, we are not suggesting that the items used in the scale (e.g., being considerate of others’ interest) are objectively beneficial to scientists, or normatively good for scientific work. Rather, they reflect the issues that have been empirically shown to be connected to higher trust in scientists among the general public. In other words, we do not assess whether participants believe that paying attention to others’ views is beneficial to scientists’ work, nor do we make any normative claim in support of this statement. Instead, we ask participants to respond to items that have been shown to influence people’s perceptions of scientists’ trustworthiness. Indeed, participants might trust scientists for reasons that are unrelated to a scientists’ ability to produce good scientific work. For example, many studies find that participants’ perceptions of scientists’ benevolence (i.e., that they act in the public interest) relate to perceptions of trustworthiness.

To lend further empirical evidence to this argument, we present correlations between individual items and dimensions and our overall trust score in the Supplementary Information Table S7. For example, we find that agreement with the item “How

considerate or inconsiderate are most scientists of others' interests?" is strongly correlated with overall trust ($r = .78$). This provides further empirical evidence that scientists' benevolence is an important reason for many to trust scientists. Naturally, a scientist concerned solely with their own career, rather than acting in the best interest of society, might still conduct high-quality research.

The conclusions of the study are weak. The first conclusion is based on the finding that scientists were rated lower on openness than on competence. The authors suggest that scientists therefore may gain more public trust by being more open (more receptive to feedback) - but that suggestion is only valid if we accept that it is desirable for scientists to maximize the scores on the authors' scale. The study does not show that the public desires scientists to be more receptive to feedback, nor that it would be good for science itself.

The authors' second conclusion is that as the different dimensions of their scale yielded differences between and within they "demonstrate the importance of using multidimensional trust measures". We are not told which finding actually demonstrates this importance or how it is important.

The third conclusion is that, as the individual-level correlations between the measure of trust in scientists and other individual-level variables vary across countries, scholars should be cautious when generalizing findings from Western countries. As a general conclusion, this has been said uncountable times before. On the particular topic at hand, I did not see that the authors provided any evidence that scholars tend to make such unfounded generalizations.

Thank you for mentioning these points, which helped us improve and strengthen the conclusions.

Regarding the comment on our first conclusion: Factors that inform public perceptions of scientists' trustworthiness should not be conflated with factors that are important/good for science itself. Whether openness is normatively desirable or not is a debate that has taken place elsewhere and is separate from our discussion of the findings. Accordingly, we do not make the normative claim that being more receptive to feedback is better for science but instead argue that being more receptive to feedback may increase scientists' perceived trustworthiness.

This view is also shared by Reviewer 1: *"The authors speak of the need for science/scientists to be more transparent as a means through which to engender increased societal trust. With that piece of advice in mind (which I strongly agree with)..."*

We are also grateful for your comment on the second conclusion, i.e. for rightly pointing out that we did not assess differences in trust dimensions within countries, but only between countries. We present figures that show between-country differences in mean levels of agreement for each of the four trust dimensions in the Supplementary Information (figs. S13-S16). Understanding scores on these individual dimensions is important because it can help scientists to better understand public perceptions and how

to act in ways that increase trustworthiness perceptions. For example, in Slovenia, average perceived competence is above the global mean ($M = 4.22$) and perceived benevolence below the global mean ($M = 3.49$). Through our data visualization tool, researchers will be able to obtain insights into public perceptions of trust in scientists in specific countries. This can help scientists to better understand how they could increase public trust. In the case of Slovenia, e.g., scientists might engage more with the public to increase perceptions of benevolence, using their expertise on specific contexts to inform their interventions and messages. We changed the text in the manuscript to reflect the fact that we only mentioned between-country differences and added a sentence that describes why it is important to assess individual trust dimensions and how they vary across countries:

“We find considerable differences of trust between countries (Fig. 1, figs. S13-16) as well as substantial country variations of the trust dimensions, which demonstrates the importance of using multidimensional trust measures in comparative survey research²⁷. This, in turn, will help scientists and science communicators to better understand how to act in ways that increase different components of trustworthiness perceptions, i.e. competence, integrity, benevolence, and openness.”

Lastly, we agree with the reviewer that the concern that findings from Western countries have been generalized to non-Western countries has been made (too) many times before. In line with the comment made by Reviewer 3, we decided to drop this sentence.

Another important conclusion of our article that is not mentioned by the reviewer is that the majority of the public agrees that scientists should be more involved in policymaking. This is the first study to analyze public perceptions of scientists' role in society and policymaking on a global scale. These findings have important implications as scientists are often called on to provide policy-relevant evidence (e.g., during the COVID-19 pandemic), but the extent to which such engagement might hurt the perceived trustworthiness of scientists remains understudied. Scientists will be able to use our data visualization tool to gain better insights into the public's normative perceptions of their role in society and policymaking.

To further strengthen the Conclusion section, we expanded the discussion of several findings as also suggested by the other reviewers. For example, we now discuss the implications of distrusting minorities (see above), and implications of our findings for science communication in more detail.

In sum, I do not think this paper has a compelling message about trust in scientists. However, as the data is very rich, I encourage the authors to take my comments as an inspiration to think again about what their ambitious study can tell us. I note that the study included other data too, some of which is mentioned as sideshows in this paper and some of which is not analyzed here

at all. As the main topic of the paper is trust in scientists, I do not discuss these aspects in this review.

Once more, thank you for your thoughtful comments, which helped us considerably in improving the paper. We have made extensive efforts to make our findings clearer and more useful for readers. We hope that these changes and responses, along with the greater enthusiasm of the other reviewers, help mitigate the previous concerns.

Reviewer #3:

Remarks to the Author:

NB This review was conducted by a tenured faculty member and a graduate student. The faculty member takes full responsibility for the content of this review.

- Key results: Please summarise what you consider to be the outstanding features of the work.

This paper reports the findings of a large-scale international survey of public trust in science. The survey is outstanding for its international scope and the authors' commitment to open science and transparency practices via the OSF preregistration and repository.

- Validity: Does the manuscript have flaws which should prohibit its publication? If so, please provide details.

We note a number of concerns with the regression models (under "Appropriate use of statistics" below) and interpretation of the findings ("Conclusion"). We recommend these issues be addressed before acceptance for publication.

We thank the reviewers for their careful evaluation of our work and the excellent suggestions on how to increase the analysis robustness. As explained below, we made thorough revisions to the main and supplementary texts.

- Originality and significance: If the conclusions are not original, please provide relevant references. On a more subjective note, do you feel that the results presented are of immediate interest to many people in your own discipline, and/or to people from several disciplines?

We are not aware of similar surveys of public trust in science on the same global scale, and consider the project highly original in that respect. The finding that trust in science is moderately high is particularly important, given a widespread misperception that there is a general crisis of trust in science. The results will be of immediate interest to scholars of public trust in science in a variety of fields, including philosophy of science, psychology, science communication, and so

on.

We thank the reviewers for their positive evaluation of our work.

- Data & methodology: Please comment on the validity of the approach, quality of the data and quality of presentation. Please note that we expect our reviewers to review all data, including any extended data and supplementary information. Is the reporting of data and methodology sufficiently detailed and transparent to enable reproducing the results?

The data and pre-processing and analysis scripts appear to be available in the osf.io repository. Because the repository is currently set to “view only” (downloading requires non-anonymous access), we did not attempt to reproduce the analysis. The project organization is perspicuous and the code itself is fairly easy to read. We believe that, with access to the data, it would be straightforward to attempt to computationally reproduce the analysis.

We thank the reviewers for their positive feedback on our analysis and project organization. It is possible to download all files needed to reproduce the analyses, including the data, even for view-only projects. Please go to https://osf.io/wj34h/?view_only=7d74fb462cb641c5ba58902bf76e9610, select “OSF Storage (Germany-Frankfurt)” and click “Download as zip”. This may take a while, because the precomputed regression models are very large files. Alternatively, one might only download the folders 01_code, 02_data, and 03_labels in the 03_main-analysis directory.

We do have a few grumbles about the code style. For future projects, we would encourage the authors to (1) use linter or a similar style checking tool, and (2) embrace the DRY principle: Don't Repeat Yourself. For example, `purrr::map()` calls should be used in place of copy-pasting.

We thank the reviewers for their advice on how to render our code more readable. We suggest that our code is fairly parsimonious thanks to the many user-defined functions (e.g., for computing the weights, fitting regression models, making plots, etc.). Using `purrr`'s `map()` function is a good suggestion. However, we prefer not to use it here, first, because it tends to make the code less easy to read for R beginners, and second, because `map()` – or `group_map()` – seem to have issues with handling the survey design objects we used for many analyses.

We also note that the data does not include a codebook or standalone instructions for pre-processing and assembling the combined dataset. To facilitate re-use of the data, the authors might consider creating a separate repository, with just the data and appropriate metadata and instructions.

Thank you for this important suggestion. We now provide a detailed stand-alone paper to facilitate re-use of the data. The paper contains a detailed description of the dataset

(including pre-processing). It is currently under review and available as a preprint: <https://osf.io/preprints/psyarxiv/jktsy>. We also created a separate OSF for the dataset and supplementary information including a codebook, which is linked in the data descriptor paper (URLs are redacted in the public preprint). We aim to publish the current study, the data descriptor paper, and the OSF repository at the same time, so that readers will find all materials clearly organized in a dedicated repository.

- Appropriate use of statistics and treatment of uncertainties: All error bars should be defined in the corresponding figure legends; please comment if that's not the case. Please include in your report a specific comment on the appropriateness of any statistical tests, and the accuracy of the description of any error bars and probability values.

The authors use regression analysis to identify potential causes and effects of trust in science. We do not object to the use of non-experimental data as provisional, defeasible evidence for causal claims. But we do have a number of concerns about how these particular regression models are specified and interpreted.

We appreciate the careful considerations and suggestions for our analyses. See below for our responses.

First, the authors highlight statistically significant results that are of little or no practical significance. For example, they highlight global differences in trust associated with gender, age, urbanicity, income, religion, and education (lines 403-405), all of which are about 0.02-0.03 of a standard deviation. To prevent the overinterpretation of such very small effect sizes, we recommend adding marginal effects plots that use unstandardized (and therefore more interpretable) units.

We thank the reviewers for this suggestion and agree that it is important not to overinterpret such small effect sizes given the large sample size. We followed the recommendations of the reviewers and added marginal effects plots for all predictors (demographic characteristics, ideological views, attitudes to science, country-level indicators) that use unstandardized units. These allow better interpretation of the effect sizes. The plots can be found in the Supplementary Information in Figure S5 on pages 13-16. Moreover, we now mention more explicitly in the main text that some effects are very small on page 15:

“The relationships between demographic characteristics and trust in scientists are very small (marginal effects plots with unstandardized units are provided in fig. S5).”

Second, using a simple DAG would improve the specification and interpretation of the regression models. For one example, in the analysis of trust (table S2), a simple DAG might look like this:

country indicators  individual demographics  attitudes to science  trust

with additional direct effects running in the same direction, e.g., from country indicators to attitudes and trust. This DAG suggests that, for identifying the effects of demographics on trust, it is important to condition on country indicators (as potential confounders) but to not condition on attitudes (as potential mediators). This would explain why the coefficients on several demographic variables have substantial changes between block 1 and block 2. Or, for identifying the effects of country indicators, none of the other sets of covariates should be included (since they're all potential mediators). Specifically, it seems plausible that individual-level education is a mediator between country literacy and education expenditure, on the one hand, and trust, on the other. So the lack of association between country literacy and education and trust (lines 471-472) is plausibly due to model misspecification error. Of the three models reported in S1, only block attitude (attitudes) is appropriately specified for analyzing the effects of the covariates of interest.

For another example where some simple causal analysis would be useful, in the analysis of research priorities (table S4), the difference between perceived and desired priorities is plausibly a cause, rather than an effect, of trust. If this is right, then conditioning on trust in the regression models introduces collider bias. At a minimum, the authors should compare regression models with and without trust to assess for the possibility of collider bias.

We thank the reviewer for mentioning DAGs as a helpful tool to organize and visualize relationships. We also appreciate the suggestions on how to improve the specification of the regression models, disentangling direct, mediation, and moderation effects and considering collider biases. We have now added a DAG in the Supplementary Material (fig. S1) and reference it in the main text on page 11:

"2) How do demographic, ideological, attitudinal, and country-level factors relate to trust in scientists (see fig. S1 for a directed acyclic graph visualizing these relationships), and how do these relationships vary across countries?"

This DAG precisely illustrates the preregistered model specifications. It slightly differs from the one(s) proposed by the reviewer. In particular, some of the suggestions were not in line with our model specification rationales:

1. We deem it less plausible to assume direct effects of country indicators on demographics and attitudes as suggested by the reviewers. We would argue that country indicators are a function rather than a predictor of demographics and attitudes to science. Take, for example, the PISA science literacy score: This score is arguably a function of a population's education level (i.e. demographic characteristic) and their sympathy for science (i.e. attitude). Therefore, our DAG does not include a direct effect from the country indicators to demographics, ideological views (which we now use as a new category for grouping the predictor variables, see below), and

- attitudes to science. However, we do assume direct effects of the indicators on trust in science.
2. Since we think that effects of country indicators on demographics and science attitudes are less plausible, we would not follow the reviewer's suggestion to condition on country indicators (as potential confounders) when analyzing effects of demographics on trust. However, we do condition on demographics when analyzing the effects of attitudes on trust, as is shown in the DAG.
 3. We argue that country indicators are an aggregate measure that consider demographics and attitudes. So, deviating from the reviewer's recommendation, we indeed conditioned on demographics and attitudes when analyzing their effects on trust in science (step 4). Our DAG visualizes this, because it assumes direct effects of demographics, ideological views, and attitudes to science on trust in science.

We hope that these explanations clarify the rationales of our model specifications.

We also appreciate the reviewer's suggestion to reconsider the model specification of the analysis of the research priorities. Here, however, we preferred to keep the original approach, because we are hesitant about assuming reversed causality of trust and research priorities. This would conflict with established assumptions to conceptualize trust in science as a fundamental perception of science that shapes other such perceptions, including expected and perceived research priorities. Moreover, running additional regression models would be stark deviations from the preregistration.

Third, the researchers measured several variables that do not appear to be included in any of their analyses. In particular, the social dominance orientation variable, which was one variable the researchers included hypotheses about in their pre-registration, is missing from their reported analyses on explanatory factors of trust in scientists. If this was a deliberate deviation from their pre-registered plans, an explanation should be included in their supplemental information alongside their explanations for their other deviations.

We are very grateful to the reviewers for reviewing our manuscript and the preregistration so carefully and for spotting this. Indeed, social dominance orientation (SDO) should have been included as an independent variable in our model as per our preregistrations: this was a mistake. We re-ran the models now including SDO, and decided to restructure the predictor variable blocks: *Demographic characteristics*, *Ideological views*, *Attitudes to science*, *Country-level indicators*. The block on ideological views now includes political orientation (right-leaning), political orientation (conservative), SDO, and religiosity. We also added the following paragraph on SDO in the introduction:

"We also assessed the role of social dominance orientation (SDO), which has been defined as "the degree to which individuals desire and support group-based hierarchy and the domination of 'inferior' groups by 'superior' groups"²⁵ (p. 48), because individuals high in SDO are arguably less likely to trust scientists as they perceive universities as

hierarchy-attenuating social institutions, for example²⁶. Research in the United States supports this, showing that SDO is a predictor of trust in scientists²⁴ and distrust in climate science^{23,24}. However, a comparative assessment of the relationship between SDO and trust in scientists across countries is lacking.”

We also added the following sentence in the results section on page 18:

“In line with previous studies^{23,24}, we find that social dominance orientation is negatively associated with trust in scientists, meaning that individuals who tend to prefer for social group-based hierarchy and inequality are less likely to trust scientists.”

We intended to also include SDO as a covariate in the regression model testing predictors of normative perceptions of the role of science in society and politics (RQ4). This omission was a mistake. Replicating the procedure of the RQ1, RQ2, RQ3, and RQ5 analyses, we also included social dominance orientation as a covariate in the RQ4 analyses. We make this slight deviation from the preregistration transparent in the Supplementary Information on page 54:

“2. The preregistration did not mention social dominance orientation as a covariate in the regression model testing predictors of normative perceptions of the role of science in society and politics (RQ4). This omission was a mistake. Replicating the procedure of the RQ1, RQ2, RQ3, and RQ5 analyses, we also included social dominance orientation as covariate in the RQ4 analyses.”

Fourth, the authors do not provide a rationale for their selection of demographic variables or the relationship between these variables and trust in scientists. In the absence of a plausible mechanism by which, for example, being older or being a woman disposes a person to trust scientists more than people who are younger or a man, this runs of risk of essentializing these demographic groups — especially since the authors highlight such relationships despite the very small effect sizes. We do think it is plausible that several of the demographic variables are convenient proxies for other constructs that are of more direct interest but difficult to measure. But, whatever the authors’ rationale, it should be noted in the methods.

We thank the reviewers for bringing up the selection and role of demographic variables for trust in scientists. As has been argued in previous publications on this subject (Besley et al., 2021), we argue that demographic variables cannot themselves cause views about scientists and instead likely reflect shared direct or mediated experiences with scientists by groups. For example, being a woman does not directly lead to more or less trust, but women’s lived experiences with science and scientists are likely different from those of men. Studies have shown that scientists are often presented as male in news and social media and that male and female scientists are often presented differently in these media (Freedman et al., 2023; Mitchell & McKinnon, 2019), which

could result in different levels of trust in scientists among women. Another example provided by Besley et al., (2021) is: *“Those with higher knowledge may have different beliefs about scientists because the process of obtaining that knowledge likely involved a deeper level of experience with scientists.”*

We included demographic variables to investigate whether trust in scientists varies across different population groups. We believe that this has important implications for scientists and science communication, for example, because it would help science communicators to better tailor their messaging to different audiences. For example, given that older population groups tend to be more trusting of scientists, scientists and science communicators may want to be especially careful in addressing young audiences in ways that increase trustworthiness perceptions.

We now added the following sentence to provide a rationale for the selection of demographic variables on page 15:

“To investigate how trust in scientists differs across population groups, we assessed several demographic variables and analyse their correlation with trust in scientists.”

We also add this sentence to the Discussion on page 26:

“Relatedly, we find that trust in scientists varies across population groups, depending on their age, gender and education, which can help scientists and science communicators to better tailor their communication to different audiences.”

Freedman, G., Moutoux, I., Hermans, I., & Green, M. C. (2023). “She made a mean beef stroganoff”: Gendered portrayals of women in STEM in newspaper articles and their effects. *Communication Monographs*, 1-21.
<https://doi.org/10.1080/03637751.2023.2285989>

Mitchell, M., & McKinnon, M. (2019). ‘Human’ or ‘objective’ faces of science? Gender stereotypes and the representation of scientists in the media. *Public understanding of science*, 28(2), 177-190. doi: 10.1177/0963662518801257

Besley, J. C., Lee, N. M., & Pressgrove, G. (2021). Reassessing the Variables Used to Measure Public Perceptions of Scientists. *Science Communication*, 43(1), 3–32.
<https://doi.org/10.1177/1075547020949547>

- Conclusions: Do you find that the conclusions and data interpretation are robust, valid and reliable?

In places, the authors present a rather superficial interpretation of their findings.

For example, with regards to political orientation and trust in scientists, the authors' analyses (figs. 3 and S5) show a positive association between trust in scientists and right-leaning political orientation in 8 countries – Georgia, Egypt, Philippines, Nigeria, Taiwan, Indonesia, Slovakia, and Greece – and the opposite association in 16 countries – China, Brazil, Austria, Israel, Canada, Norway, Italy, Germany, Netherlands, Portugal, Spain, Switzerland, United Kingdom, Belgium, Denmark, Sweden, and the United States. That is, twice as many countries show one relationship rather than the other; and the larger subset represents both a very large proportion of the world's population and a majority of the world's scientific output. Thus, the consequences of a negative association between right-leaning political orientation and trust in scientists at both the country-level and worldwide level are likely quite different and more substantial than the opposite association. But the authors' conclusion here is simply that there is variation between Western and “non-Western” countries (lines 427-443). (Also, from the text on lines 825-830, it's unclear which covariates were included in these random slope models, and so whether this covariate selection was appropriate to estimate these effects.)

We thank the reviewers for this comment. In the revised version, we no longer find that a negative association between trust in scientists and right-leaning political orientation is more common than a positive association. This is for three reasons. First, we included data from one additional country (Cameroon). Second, the model underlying this figure controls for demographic variables, and the variable religiosity is now included in the block “ideological views” rather than demographic characteristics as in the previous version. Third, we now present country-level effects (i.e., sums of the global/fixed effect and random effects for each country) instead of random effects (i.e., deviations of country-level effects from the global/fixed effect) in Fig 3, because this is more informative for assessing whether trust is negatively or positively related to political orientation in a given country. Therefore, this concern is no longer applicable.

Second, the authors recommend that “scholars ... be cautious when generalizing findings from Western or Anglophone countries.” (lines 558-9). It is always good to advise caution concerning the appropriate generalizability of findings, but are any scholars making such generalizations? References 7, 8, 9, 10, 19, 22 and 23 all clearly report their results in the context of variance across countries when discussing and summarizing their findings, and 20 and 21 do limit their discussions and interpretations to what their results mean in the context of the United States (the only country they sampled from). The authors' own literature review does not seem to indicate a problem of inappropriate generalizations on the part of scholars.

Thank you. Reviewer 2 also highlighted this sentence, and we decided to delete it.

Third, the authors discuss disparities between participants' desired and perceived research priorities, and conclude that “greater consideration of public research priorities in scientific research and public funding presents an important avenue to increase trust” (lines 568-570). While we agree with this policy recommendation, a great deal of research that aligns with the

desired priorities is already happening. There is substantial research to improve public health, solve energy problems, and reduce poverty. This suggests that the perception gap might primarily be a communications issue. So a more pointed recommendation would be expand and improve upon this aspect of science communication, to elevate the prominence such research has in the minds of members of the general public. This would be a specific recommendation that follows directly from the kind of general recommendation the authors suggest on lines 536-542. Alternatively, participants may be answering the question of how much science aims to tackle those goals by reframing it as what they perceive are actual policies to address those issues, and the reported gap is between what participants desire scientists should prioritize and what they perceive are actual tangible efforts to tackle those goals.

That is an excellent suggestion, thank you! We added the following recommendation on page 27:

“At the same time, science communication efforts could increasingly focus on highlighting ongoing research on public health and solving energy problems to elevate the prominence of this research in the minds of the public.”

Suggested improvements: Please list additional experiments or data that could help the work in a revision.

In the Methods subsection “Desired and perceived goals of science,” it would be helpful if the authors listed the goals presented to participants.

Good suggestion: we now list the goals presented to participants.

• References: Does this manuscript reference previous literature appropriately? If not, what references should be included or excluded?

The authors report that only a minority of their sample expressed low trust in scientists, but their discussion on the potential consequences of even a minority lacking trust in scientists (lines 550-554) seems superficial. We (the co-reviewers) disagree among ourselves about the significance of minority distrust. But simulations have suggested that even a minority that is biased against science can have substantial consequences for the public's view on the scientific consensus (Lewandowsky et al., 2019; full citation below). We would encourage the authors to include this in the Discussion and Recommendations.

Lewandowsky, S., Pilditch, T. D., Madsen, J. K., Oreskes, N., & Risbey, J. S. (2019). Influence and seepage: An evidence-resistant minority can affect public opinion and scientific belief formation. *Cognition*, 188, 124-139.

We thank the reviewers for mentioning this important and relevant publication. We now cite this publication:

“When even reasonably small minorities of the public distrust science, it can have a considerable impact on society, especially if they receive extensive news media coverage and include people in positions of power that can influence policymaking^{54–56}. Studies have also shown that a minority of 10% can be sufficient to flip a majority⁵⁷ and that when a critical mass value of 25% is reached, majority opinion could be tipped⁵⁸. In the context of climate change, an agent-based model showed that an evidence-resistant minority can delay the process of public opinion converging with the scientific consensus⁵⁹.”

- Clarity and context: Is the abstract clear, accessible? Are abstract, introduction and conclusions appropriate?

We find the writing to be clear and accessible throughout, and the abstract and introduction to be appropriate. Our concerns with the conclusion are discussed above.

- Inflammatory material: Does the manuscript contain any language that is inappropriate or potentially libelous?

We do not believe any of the language or claims made in the paper are inappropriate or potentially libelous.

- Springer Nature is committed to diversity, equity and inclusion; please raise any concerns that may in your view have an impact on this commitment.

We have no DEI-related concerns about the paper.

- Please indicate any particular part of the manuscript, data, or analyses that you feel is outside the scope of your expertise, or that you were unable to assess fully.

None to report.

Your manuscript has now been evaluated by Reviewers 1 and 3 from the original round of review as well as a new Reviewer 4 with expertise in large-scale surveys and trust research. All reviewer feedback is included at the end of this letter. Although the reviewers found your manuscript to have improved during revision, they also raise some important outstanding concerns. We remain interested in the possibility of publishing your study in Nature Human Behaviour, but would like to consider your response to these outstanding concerns in the form of a revised manuscript before we make a decision on publication.

We thank the editor for the opportunity to revise our manuscript. Please find a detailed description of our changes below.

1. Reviewer 4 find the work to lack a theoretical framework or rationale, resulting in the findings appearing to be disconnected. Please address these concerns in full and explain the practical or theoretical relevance of your findings.

We have addressed this comment by restructuring our introduction and results section. The introduction now provides a clear framework and rationale that connects our variables of interest in a coherent narrative. We also provide theoretical explanations for our findings, including the relationship between trust and gender, education, religiosity, income inequality, and social dominance orientation. We have also provided a rationale for our assessment of perceived and desired research priorities in the introduction as requested by the reviewer.

2. Reviewer 4 suggests that your findings seem to largely overlap with the existing literature on trust in institutions in general. The reviewer asks that you explore this further. Please provide additional analyses and discussion to address this point.

Using data from the Wellcome Global Monitor on trust in governments, we investigated the relationship between trust in scientists and trust in governments. We did not find credible evidence for a relationship between trust in scientists and trust in governments. These findings are presented in the main text and fig. S14.

3. Please follow Reviewer 4's suggestion and empirically test explanations for the result that country level inequality is associated with trust in scientists.

We followed Reviewer 4's suggestion and empirically tested two explanations for the result that country level inequality is associated with trust in scientists. We found that the association between income inequality and trust is moderated by the Perceived Corruptions Index. We report this finding in the main text and Table S3.

4. Reviewer 2 remains unconvinced by some aspects of your rebuttal. Carefully review the persisting concerns raised by Reviewer 2, and provide the requested analyses.

We addressed these concerns by running additional analyses, discussing the findings on social dominance orientation, and providing a rationale for demographic variables as requested.

Reviewer #1:

Remarks to the Author:

The authors' revisions are thoughtful, thorough, and responsive and have ultimately made the manuscript even more robust. Thank you for your efforts to produce this highly salient research and for being so willing to grapple with the many insightful critiques shared by the reviewers.

Thank you for your kind feedback.

Reviewer #3:

Remarks to the Author:

We do not feel that the following issue has been adequately addressed:

- Research priorities and trust: The issue here was that the authors had trust as a IV/cause of (perceived) research priority gaps, and we said this should be reversed. In the response, the authors reject this suggestion, giving two arguments: (a) trust is "a fundamental perception of science that shapes other such perceptions"; (b) making trust the DV in this model would deviate from the preregistration (17).

For (a), we don't know what the authors mean by a "fundamental perception." Whatever it means, it seems to entail that trust cannot be treated as a DV/effect in a regression model. But the authors use trust as a DV/effect in the figure 2 models. Notably, these models include a block of "attitudes to science," including the perceived benefit of science. We do not understand why trust can be treated as an effect of views on the perceived benefit of science, but not as an effect of perceived research priority gaps. Indeed, it seems to us like perceived research priority gaps are likely to be causally upstream from the perceived benefit of science: all else being equal, insofar as someone thinks science does not address important research priorities, then they will perceive the benefit of science to be lower.

For (b), the preregistration for RQ5 is phrased in terms of correlations, so swapping trust and priorities between the IV and DV sides of a regression model would not be a deviation.

Thank you for your perseverance on this point. We agree that conceptually, a reversed relationship between IV and DV is also plausible. We followed your suggestion and reran our analyses with trust in scientists as dependent variable and the discrepancy scores of research priorities as independent variables. We present these results in the Supplementary Information in Table S6. Indeed, the difference between perceived and desired research priorities was positively related to trust in scientists for the goals "Improve public health", "Solve energy problems", "Reduce poverty". We found the opposite for "Develop defense and military technology", where the perception that perceived priorities exceed desired priorities was negatively related to trust.

We describe these results on page 31:

"Further analyses showed that the discrepancy between people's perceived and desired research priorities is associated with trust in scientists. On the one hand, higher trust is linked to a lower likelihood that scientists' efforts do not meet people's expectations for the following goals: improve public health, solve energy problems, and reduce poverty (table S5; see exploratory analyses with reversed hypothesised causality in table S6). In other

words, the more people trust scientists, the more they perceive that scientists' efforts exceed expectations. On the other hand, a higher likelihood that peoples' perceived efforts exceed expectations is associated with less trust in scientists in the case of developing defence and military technology (i.e., those who think scientists are too focused on defence and military technology trust science less)."

For transparency, we added the following note in the Supplementary Information and report the results of both the preregistered and exploratory analysis:

"The preregistered analyses treated the discrepancy between perceived and desired priorities as the dependent variable and trust in scientists as an independent variable (Table S5). We ran an exploratory analysis with trust in scientists as the dependent variable and the discrepancy scores as independent variables. The results of these exploratory analyses are reported in the main text and Table S6."

We are satisfied on balance with the authors' changes for the following issues, but feel that the paper might benefit from some further revision:

- Social dominance orientation: The authors explain that SDO was omitted by accident, and it has now been included in, e.g., block 2 in the figure 2 models. But the finding — that high SDO is associated with low trust — isn't mentioned in the Discussion. We feel that this finding has the important implication that scientists and science communicators could increase different components of trustworthiness by challenging unjust hierarchies and social stratification.

Thank you for this suggestion. We now interpret the SDO findings in the results section on page 25 and the discussion section on page 34. In particular, we have added the potential implication that our findings point to a broader role of SDO in potentially undermining acceptance of scientific conclusions that have been observed in the literature (e.g., SDO relating to climate denial, vaccine hesitancy). We have added proposed applications of the findings, including the need to build perceived trustworthiness of scientists but also to involve non-scientist communicators who may already be trusted by those higher in SDO.

We have added the following paragraph in the results section on page 25:

"Some studies have looked at social dominance orientation—that is to say, a preference for social hierarchy and inequality—and found it to be negatively associated with trust in scientists^{19,43}. Our results confirm this: The low global mean for SDO ($M = 3.62$, $SD = 1.76$; 1 = extremely oppose – 10 = extremely favor) is consistent with the overall moderately high trust in scientists. Moreover, we find that those who favour hierarchy enhancement (i.e., more strongly endorse SDO) are less likely to trust scientists ($b = -0.098$, $p < .001$). This may be because they see universities (i.e., scientists) as institutions that weaken social hierarchies⁴²."

And the following paragraph in the discussion on page 34:

"Our findings also highlight the inconsistency in the association between political orientation depending on the measure used (left-right versus liberal-conservative) and country of study, as well as the importance of ideology—specifically SDO—in relating to trust in scientists. While previous research finds that SDO is associated with the rejection of

specific scientific information, such as the reality of climate change⁷³ or the safety and efficacy of vaccines²⁶, our findings support the idea that SDO may play a more fundamental role in undermining trust in scientists more generally. Scientists who challenge unjust social hierarchies might increase benevolence perceptions among some groups but would likely further decrease trust among people with social dominance orientations. Interventions could be developed to build perceived trustworthiness of scientists and involve trusted communicators outside of scientific institutions.”

- Rationale for demographics: The authors explain in the response that the chosen demographic variables plausibly "reflect shared direct or mediated experiences with scientists" (18). We recommend including this rationale in the text of the paper itself.

Thank you for this suggestion, we added this explanation in the discussion on page 33:

“While demographic characteristics likely cannot cause views about scientists per se, they reflect shared direct or mediated experiences with scientists. For example, women’s lived experiences with science are likely different from those of men. Media coverage disproportionately features male scientists and portrays them differently than female scientists, which may evoke different trustworthiness perceptions across genders^{70,71}.”

We are satisfied with the changes made in response to the following issues:

- Practical significance vs. statistical significance: The authors have noted in the text that the effects are very small, and included marginal effects plots in the supplement as we suggested.

- Using a DAG to select covariates: The authors have added a DAG in the supplement. It differs from the one that we suggested, and mostly justifies the order of blocks of covariates as used in the previous version. There’s a slight modification, with “demographics” now split into “demographics” and “ideological views” blocks. While this isn't the DAG that we would use, we're satisfied that at least the authors' implicit DAG is now explicit, and the data will be available for others to conduct an alternative analysis if they'd like.

Following the comment made by Reviewer 4, we now also present the alternative DAG that you suggested in the previous round of revisions in figure S1b. We also added the following sentence in the Supplementary Information on page 6:

“There are distinct causal pathways that could be hypothesized and tested. We focus on this particular causal model depicted in our directed acyclic graph (DAG; fig. S1a) which is in line with previous global studies on trust^{7,93,94}. For an alternative DAG proposed by an anonymous reviewer see fig. S1b. Our data is publicly available, and we encourage other researchers to explore this and other alternative models.”

- Negative association between trust and right-leaning political orientation: Due to changes in the data and regression model specification, this point no longer applies.

- Cautioning against WEIRD generalizations: The authors have dropped the relevant sentence.

- Science communication on research priorities: The authors have added a sentence in line with our suggestion.
- List all the research goals: The authors have added this in line with our suggestion.
- Minority distrust: The authors have added the citation we suggested.

Reviewer #4:

Remarks to the Author:

I'm a new reviewer on this manuscript. The paper employs a large, global, research team to address the extent to which people across the world trust scientists. This is an ambitious data collection effort on an important topic. Below I list some things I think the authors could/should consider doing to improve the paper. They appear in the order in which they occurred in the manuscript, rather than by importance.

Before I get to these issues, however, I'll mention what I see as the most significant limitation of the paper, as currently written. After reading the paper, I found it difficult to remember the answers to several of the paper's four key questions/goals. It *is* clear that trust in scientists is reasonably high (higher than popular wisdom would suggest) and that there are country-level differences in trust in scientists. The authors do a good job communicating these things. What is difficult to hang on to (or recall) after reading the paper, however, are answers to the other questions – perhaps especially Question/Goal 2, which is “How do demographic, ideological, attitudinal, and country-level factors relate to trust in scientists and how do these relationships vary between countries?” After reading the paper, I tried to write down the answer(s) to this question and was mostly at a loss. The reason is that the paper is highly descriptive and does not have much in the way of theory – or even tentative explanations– to make sense of any given result, much less anything that ties the many results together into a coherent framework or narrative. As a result, much of the Results section comes off as a laundry list of largely disconnected findings. To be clear, I am not suggesting that the authors need to begin with some overarching theory – I doubt any such theory exists for what they are trying to explain. Rather, I am encouraging the authors to avoid simply tossing out result after result and propose a (possible) explanation or interpretation of any given result or set of results that they present. This will allow readers to connect the findings to some framework and take away - actually remember - more from the paper. I think this is important to do, even if it means some findings need to be relegated to an SI to make space. Otherwise, the answers to the key questions motivating the paper are muddled or unclear. Several of the specific points below touch on this more general point.

Thank you for your thoughtful comment. We have carefully addressed all your comments and suggestions. We have rewritten parts of the introduction and restructured it to provide a more coherent narrative and clear rationale of our work. We added several theoretical explanations throughout the results and discussion sections to tie the results into a coherent narrative. We now clearly introduce relevant concepts in the introduction and motivate their inclusion in our models based on the trust literature. We then summarize our main findings in the results section, theoretically contextualize them, and compare them to the results of previous studies. Importantly, we have provided more explanations for our

results on Question/Goal 2, “How do demographic, ideological, attitudinal, and country-level factors relate to trust in scientists and how do these relationships vary between countries?”, including the relationships between trust in scientists and gender, education, religiosity, social dominance orientation, and income inequality. Finally, we ran additional analyses with individual trust dimensions and tested whether corruption and oversampling of urban populations in our sample moderate the relationship between income inequality and trust in scientists. We also analyzed the relationship between trust in government and trust in scientists and visualized these results in the Supplementary Information in fig. S14. We are grateful for your feedback, which we feel has made the manuscript more robust.

i) P. 11. I know the authors added the “why do we study scientists rather than science” to address one of the previous reviewer’s concerns. But the new discussion (on p. 11) seems out of place and will probably appear to readers like it comes out of nowhere. Either placing this discussion later - or, perhaps even better, after the next paragraph that outlines the research questions - would be more logical.

Thank you for this comment. We have restructured our introduction and moved this paragraph to the end of the introduction.

ii) Likewise, the authors write “However, research has shown that it makes a substantial difference whether one is being asked about trust in science, scientists, or scientific methods.” Readers will probably wonder how it makes a difference. After all, the types of differences it makes and the extent of those differences will have important implications for what readers can/should take away from this paper, given that the authors measured trust in scientists rather than science, scientific methods, etc. A brief clarification and explanation would help better situate the authors’ contributions.

Thank you for highlighting this point. We agree that a better explanation on how trust in science, scientists, and scientific methods differs will be helpful in interpreting our results. We edited the sentence cited above and added the following text on page 14:

“We measured trust in scientists (instead of science) because “science” is more abstract than “scientists” and therefore makes a less clear referent: People may think of scientific institutions, scientific communities, scientific methods, or individual scientists when being asked about their general perception of “science”. However, these trust measures can be distinguished both conceptually and empirically^{25,44}. For example, research has shown that less educated people trust scientific methods more than scientific institutions⁴⁴. General measures that assess trust in the scientific community only capture some of the conceptually established dimensions of perceived trustworthiness (e.g., expertise)²⁵. We reduced this ambiguity by avoiding the abstract category “science,” and using the more concrete reference object “scientists”⁷.”

iii) I don't think readers can understand from the authors’ text why they favor the directed relationships in figure S1 (indeed, the reviewer that prompted the addition of that figure proposed an altogether different DAG). In their responses to that reviewer, the authors point to some rationale for their favored DAG (though I personally did not find that rationale very convincing), but no rationale at all is given in the paper. If there is a good argument for these directed relationships over

others, the authors should spell them out (even if happens in the SI). Otherwise, I don't think this figure is particularly useful and will probably only confuse readers.

Thank you for pointing out that the relationship between the variables in our model shown in figure S1 was not clear. Since presenting a DAG was recommended by Reviewer 3, we decided to keep figure S1. However, we now further justify our directed relationships presented in figure S1 by citing several studies with comparable data structures that employ similar DAGs to study determinants/correlates of trust with multilevel models. We also encourage other researchers to explore alternative DAGs in future research.

We added the following text to the Supplementary Information on page 6:

“There are distinct causal pathways that could be hypothesized and tested. We focus on this particular causal model depicted in our directed acyclic graph (DAG; fig. S1a) which is in line with previous global studies on trust^{7,93,94}. For an alternative DAG proposed by an anonymous reviewer see fig. S1b. Our data is publicly available, and we encourage other researchers to explore this and other alternative models.”

iv) Given that NHB papers should be readable by educated non-specialists, in the introduction, the authors should clarify/explain two key concepts in their research questions, namely “people’s normative perceptions of scientists in society and policymaking” and “people’s perceived research priorities.” (I also think the latter issue could be briefly motivated/anticipated a bit better, as it seems to come out of nowhere.)

We agree that these concepts might not be fully understood by educated non-specialists and have provided more information. On page 10, we now write:

“First, we analyse the extent to which people believe that scientists should be involved in society and policymaking. We refer to this as normative perceptions of science in society and policymaking.”

We now also provide additional information on why we assess people’s perceived research priorities on pages 10-11:

“Second, we investigate which issues people want scientists to prioritise in their work, and how such perceptions are related to their trust in scientists. Previous studies have shown that trust is affected by the perception of value alignment²⁴. People who feel that their concerns and values are not reflected in the priorities of scientists may therefore doubt the trustworthiness of scientists.”

v) On p. 12 the authors introduce four dimensions of trust in science but then (as far as I can tell) this distinction is immediately dropped in the next paragraph (without warning or explanation) and isn’t revisited again until the Discussion. I’d encourage the authors to i) warn readers that they are collapsing them and explain why, and ii) tell us whether the main results from the aggregate measure (largely) hold up with the individual dimensions.

Thank you for mentioning this point. We now justify using the trust in scientists index instead of the individual dimensions and inform readers that we use the trust in scientists index for subsequent analyses (page 15):

“When these components of trustworthiness perceptions are aggregated to a single score, the index represents an integrative measure of public trust in scientists with strong reliability (Cronbach’s $\alpha = 0.93$ and $\omega = 0.95$). Therefore, we used the aggregate index for our main analyses (see SI for additional analyses with individual trust dimensions).”

To analyze whether the main results from the aggregate measure hold up with the individual dimensions, we reran the analyses underlying Figure 2 with each of the four trust dimensions (i.e., competence, integrity, benevolence, and openness) as outcome variable. As can be seen in the Supplementary Information (fig. S4-S7) results from the aggregate measure largely hold up with those of models using individual trust dimensions. These additional analyses say little about predictors of *trust in scientists*, and only provide information about different *dimensions* of trust in scientists.

We also refer to these exploratory analyses in the caption of Figure 2:

“Results of exploratory analyses with individual trust dimensions are reported in fig. S4-S7.”

vi) There is a large literature on trust in institutions in general. Just eyeballing Figure 1, it seems likely that the results would overlap a great deal with international differences in trust in institutions. Shouldn’t the authors address it? Even if the authors did not measure it, they could perhaps correlate it with country-level responses from the World Values Survey or elsewhere. Even if this is not possible, it may warrant a discussion of whether the authors view country level differences in (dis)trust of science/scientists as part of a more general (dis)trust of institutions more broadly.

Thank you for this suggestion. We agree that studying the relationship between trust in scientists and trust in institutions is an interesting research endeavor. We therefore inspected this relationship at the country level. To assess trust in institutions, we used data from the Wellcome Global Monitor, as this dataset contained data on trust in institutions for most of the countries sampled in our study. Specifically, we use the share of respondents who answered "a lot" or "some" to the question: "How much do you trust your national government?". Please note that this question was not assessed in Botswana, China, and Congo DR (for which we have data on trust in scientists). We decided not to use the Integrated Values Survey Data (World Value Survey and European Value Survey), because data on trust in institutions is not available for 13 countries sampled in our study.

We calculated the Pearson correlation between trust in scientists (weighted country mean) and trust in government and found that the two variables are not significantly correlated: $r = 0.14$, $t = 1.1$, $df = 63$, $p = .27$. We visualize the relationship between the two variables across countries in the Supplementary Information (fig. S14).

We added following sentences on page 27:

“Comparing trust in scientists to trust in the national government (based on country estimates from the Global Wellcome Monitor) supports this assumption. Some countries with higher perceived corruption rank considerably lower in trust in the government than in

trust in scientists, whereas the opposite applies to less corrupt countries with lower perceived corruption (see fig. S14). Overall, trust in the government and trust in scientists are not significantly correlated at the country level ($r = 0.138$, $t = 1.104$, $df = 63$, $p = .274$)."

vii) As mentioned earlier, following Reviewer 2's comments, this paper is at times extremely descriptive. I think I view this as somewhat less problematic than R2's comments seem to suggest. But the results section goes through many findings, often without couching those findings in broader theoretical terms (just as one example, see the discussion of country-level determinants of how religiosity is associated with trust in science on pp. 15-16). Without some context or explanation, it makes it harder for reader to recall or "hold on" to particular findings. I'm not suggesting the paper needs an all-encompassing theory – rather just basic theoretical explanation of the (likely) cause of a given pattern of results they present. Without this, the paper often comes off as a long list of largely disconnected findings.

We agree that better integrating theoretical explanations would improve comprehension and impact. We now provide additional theoretical explanations for several of our results, especially those pertaining to Research Question 2, including the relationship between trust in science and religion on page 21. The paragraph on science and religion now reads:

"Many might also assume that religiosity is associated with lower trust in scientists, given that many studies conducted in Global North countries find this relationship (see e.g.,^{19,49}). However, against this assumption, one previous study found that only 29% of people worldwide believe that science stands in disagreement with their religion⁷. Another study found that while religiosity is associated with negative attitudes towards science in the United States, the relationship is inconsistent across the world⁵⁰. Indeed, we find that, globally, religiosity is positively associated with trust in scientists ($b = 0.051$, $p < .001$). However, as previous studies have also shown, we find substantive differences across countries and regions^{50,51}. In Muslim countries such as Türkiye, Bangladesh, and Malaysia (fig. S10), trust is positively associated with religiosity. In contrast, in many Christian-majority European countries, and in Israel, trust is negatively associated with religiosity. Qualitative interviews conducted by the Pew Research Center put these findings into context⁵². They found that most Muslim participants did not perceive a conflict between science and religion, because their holy text, the Quran, proclaims many principles of science. Conversely, some Christians perceive that science disagrees with their religion, even though there are pronounced variations across countries⁵². Our findings are consistent with these results. "

Additionally, we added theoretical explanations for the relationships between gender and trust in the discussion on page 33:

"While demographic characteristics likely cannot cause views about scientists per se, they reflect shared direct or mediated experiences with scientists. For example, women's lived experiences with science are likely different from those of men. Media coverage disproportionately features male scientists and portrays them differently than female scientists, which may evoke different trustworthiness perceptions across genders^{70,71}."

As well as an explanation on the relationship between education and trust on page 20:

“One might assume that trust in science would correlate with tertiary education, as people with more years of schooling and university education have had more chances to build a closer relationship with science and experience the competence and benevolence of scientists, for example⁴⁸. However, our data shows only a small positive relationship between tertiary education and trust in scientists on average ($b = 0.035, p < .001$). In fact, in most countries we find little or no credible evidence for a relationship between tertiary education and trust (fig. S8).”

We also added the following sentences on the relationship between SDO and trust on page 25 and page 34 respectively:

“Some studies have looked at social dominance orientation—that is to say, a preference for social hierarchy and inequality—and found it to be negatively associated with trust in scientists^{19,43}. Our results confirm this: The low global mean for SDO ($M = 3.62, SD = 1.76$; 1 = extremely oppose – 10 = extremely favor) is consistent with the overall moderately high trust in scientists. Moreover, we find that those who favour hierarchy enhancement (i.e., more strongly endorse SDO) are less likely to trust scientists ($b = -0.098, p < .001$). This may be because they see universities (i.e., scientists) as institutions that weaken social hierarchies⁴².”

“Our findings also highlight the inconsistency in the association between political orientation depending on the measure used (left-right versus liberal-conservative) and country of study, as well as the importance of ideology—specifically SDO—in relating to trust in scientists. While previous research finds that SDO is associated with the rejection of specific scientific information, such as the reality of climate change⁷³ or the safety and efficacy of vaccines²⁶, our findings support the idea that SDO may play a more fundamental role in undermining trust in scientists more generally. Scientists who challenge unjust social hierarchies might increase benevolence perceptions among some groups but would likely further decrease trust among people with social dominance orientations. Interventions could be developed to build perceived trustworthiness of scientists and involve trusted communicators outside of scientific institutions.”

viii) Related to the previous point, the authors should explain the concepts underlying the two different measures of political orientation. The authors devote quite a bit of space (pp. 17-18) to how one measure matters (in some way) in some countries and the other measure matters more (in some way) in other countries, without ever explaining the underlying constructs (and how the measures, e.g., the left/right distinction make sense in a global context). As a result, that entire discussion doesn't make much sense.

Thank you for mentioning this point. The different measures were selected based on a review of the literature and careful deliberation with our interdisciplinary author team. Both measures have been used in large-scale international surveys. Some international studies used a measure of political orientation on a scale from *liberal* to *conservative* (Vlasceanu et al., 2024), while others used a measure from *left* to *right* (De keersmaecker et al., 2024; Inglehart et al., 2014) and *left-wing* to *right-wing* (van Bavel et al., 2022). These are just a few examples to show that both measures are used in global comparative, cross-national

surveys. Given that some scales might be better suited at measuring political ideology in some countries than others, we decided to assess both. This gave us the opportunity to provide more nuanced findings on the role of political orientation, going beyond some of the studies cited above. It also gives researchers interested in producing secondary publications with our data the option of choosing the measure that they perceive to be most suited for their national context.

In a previous manuscript version, we only reported findings for the left-right measure, but included results with the liberal-conservative measure in the last round of revisions to transparently report that the two measures lead to different results in some countries. A nuanced political science discussion on the two measures and a detailed analysis of which measure is best suited in individual countries are beyond the scope of this article, and we hope that future research will investigate and entangle these differences. In sum, we believe the two measures to be conceptually distinct, report both for transparency, and have empirically tested that the inclusion of both measures in our model does not lead to issues of multicollinearity (table S14).

To reflect the fact that both measures are used in contemporary international social science research, we added the following sentence on pages 22-23 where we also reference ongoing discussion on these measures:

“Given that some recent global social science studies used a left-right measure to assess political orientation while others used a liberal-conservative measure⁵³⁻⁵⁵, we used both measures and analysed how results vary depending on the measure in question. [...] We encourage future research to investigate differences in the two measures of political orientation on the country level (for broader discussions on these measures see⁵⁷⁻⁶⁰).”

De keersmaecker, J., Schmid, K., Sibley, C. G., & Osborne, D. (2024). The association between political orientation and political knowledge in 45 nations. *Scientific Reports*, 14(1), 2590. <https://doi.org/10.1038/s41598-024-53114-z>

Inglehart, R., C. Haerpfer, A. Moreno, C. Welzel, K. Kizilova, J. Diez-Medrano, M. Lagos, P. Norris, E. Ponarin & B. Puranen et al. (eds.). 2014. *World Values Survey: All Rounds - Country-Pooled Datafile Version*: <https://www.worldvaluessurvey.org/WVSDocumentationWVL.jsp>. Madrid: JD Systems Institute.

Van Bavel, J. J., Cichocka, A., Capraro, V., Sjøstad, H., Nezlek, J. B., Pavlović, T., Alfano, M., Gelfand, M. J., Azevedo, F., Birtel, M. D., Cislak, A., Lockwood, P. L., Ross, R. M., Abts, K., Agadullina, E., Aruta, J. J. B., Besharati, S. N., Bor, A., Choma, B. L., ... Boggio, P. S. (2022). National identity predicts public health support during a global pandemic. *Nature Communications*, 13(1), 517. <https://doi.org/10.1038/s41467-021-27668-9>

Vlasceanu, M., Doell, K. C., Bak-Coleman, J. B., Todorova, B., Berkebile-Weinberg, M. M., Grayson, S. J., Patel, Y., Goldwert, D., Pei, Y., Chakroff, A., Pronizius, E., van den Broek, K. L., Vlasceanu, D., Constantino, S., Morais, M. J., Schumann, P., Rathje, S., Fang, K., Aglioti, S. M., ... Van Bavel, J. J. (2024). Addressing climate change with behavioral science: A global intervention tournament in 63 countries. *Science Advances*, 10(6), eadj5778. <https://doi.org/10.1126/sciadv.adj5778>

Bauer, P. C., Barberá, P., Ackermann, K., & Venetz, A. (2017). Is the Left-Right Scale a Valid Measure of Ideology? *Political Behavior*, 39(3), 553–583. <https://doi.org/10.1007/s11109-016-9368-2>

Otjes, S., & Rekker, R. (2021). Socialised to think in terms of left and right? The acceptability of the left and the right among European voters. *Electoral Studies*, 72, 102365.

<https://doi.org/10.1016/j.electstud.2021.102365>

Thorisdottir, H., Jost, J. T., Liviatan, I., & Shrout, P. E. (2007). Psychological Needs and Values Underlying Left-Right Political Orientation: Cross-National Evidence from Eastern and Western Europe. *Public Opinion Quarterly*, 71(2), 175–203.

<https://doi.org/10.1093/poq/nfm008>

Weeden, J., & Kurzban, R. (2016). Do People Naturally Cluster into Liberals and Conservatives? *Evolutionary Psychological Science*, 2(1), 47–57.

<https://doi.org/10.1007/s40806-015-0036-2>

ix) The authors do quickly entertain several explanations for the (somewhat surprising) result that country level inequality is associated with trust in scientists. But couldn't both these explanations (percent urban population in a given country and corruption perceptions) be tested with their data? (The authors point to Figure S9 as suggestive evidence for the corruption explanation, but I don't see how eyeballing the Figure can actually tell us anything about this explanation. Why not test it statistically and put that test in the SI?)

Thank you for mentioning this important point, which we addressed by running additional analyses. We tested whether the level of perceived corruption and oversampling of urban populations in our sample affect the relationship between the Gini index and trust in scientists by adding two interaction terms to the regression model: 1) Gini index * Corruption Perceptions Index, 2) Gini index * Oversampling of urban population in our sample.

We retrieved the Perceived Corruptions Index from Transparency International. To assess whether urban populations were overrepresented in our sample, we calculated – for each country – the difference between the percentage of urban respondents in the sample and the percentage of urban population according to World Bank data. Larger values indicate oversampling of urban populations in our sample.

As can be seen in table S3 in the Supplementary Information, we find a significant interaction effect for Gini index * Corruption Perceptions Index. This supports our assumption that income inequality affects trust in scientists positively in countries where there is a high level of perceived corruption, potentially because people in these countries see scientists as a trustworthy alternative to perceivably corrupt (political) elites.

However, we do not find a significant interaction effect for Gini index * Overrepresentation of urban population in our sample. It should be noted that the statistical power of this interaction is low. We reported this in the text.

We now report these results on pages 26-27:

“One possible explanation for the discrepancy between the Monitor and our study is that urban populations—which are more likely to trust scientists (Fig. 2)—were overrepresented in our samples from countries with high Gini scores, e.g., South Africa. However, a non-

preregistered analysis advised against this explanation: The extent of oversampling urban participants (difference of urban-residence individuals in sample vs. in population) did not moderate the effect of the Gini index on trust in scientists. We found tentative support for another explanation: The relationship between income inequality and trust (fig. S13) is largely driven by countries with a high degree of corruption (primarily Latin American countries as well as Sub-Saharan African countries), as indicated by a significant, but very low-powered ($1 - \beta = 0.25$ at $\alpha = .05$) interaction effect of the Gini index \times Transparency International's Corruption Perceptions Index⁶¹ (table S3). This suggests that people in countries with high inequality may see scientists as a trustworthy alternative to perceivably corrupt governments and political and economic elites⁶²⁻⁶⁴. Comparing trust in scientists to trust in the national government (based on country estimates from the Global Wellcome Monitor) supports this assumption. Some countries with higher perceived corruption rank considerably lower in trust in the government than in trust in scientists, whereas the opposite applies to less corrupt countries with lower perceived corruption (see fig. S14). Overall, trust in the government and trust in scientists are not significantly correlated at the country level ($r = 0.138$, $t = 1.104$, $df = 63$, $p = .274$).

x) The authors motivate the paper and then conclude with the point that the media and anecdotal evidence suggest that trust in scientists is very low (and imply that this paper offers a corrective to that misperception). This paper offers somewhat surprising results, from the perspective of these popular perceptions. But as the introduction notes, there are prior cross-cultural studies of trust in science and scientists. It may be useful to note in the Discussion whether the findings of this paper are also surprising from prior scientific work on this. If not, this is important to know (since as the authors note, this paper offers design improvements over prior work and employs a larger sample).

This is an important point. Our results on the levels of trust in scientists are largely in line with previous global surveys on the topic. For example, the Wellcome Global Monitor (2018, p. 53) reports:

“Globally, 54% of people have a ‘medium’ level of trust in scientists, according to the Wellcome Global Monitor Trust in Scientists Index. Fewer than one in five people (18%) have a ‘high’ level of trust in scientists, and one in seven people (14%) have a ‘low’ level of trust in scientists. Another 13% of individuals did not offer an opinion on a majority of the individual questions asked as part of the Index”.

In their 2020 study, the Wellcome Global Monitor reports (p. 3):

“Globally, people were more likely to express a high degree of trust in science and scientists in 2020 than they were in 2018: there was a 10-percentage-point increase in people saying they trust science in general ‘a lot’, while the percentage who said they trust scientists in their country ‘a lot’ rose nine percentage points.”

Similarly, IPSOS finds that a majority (57%) of the public in 22 countries trusts scientists, with levels of trust in scientists being relatively stable between 2018-2022.

While comparing these results is not straightforward given the different trust measures used across studies, our findings are in line with those of other international: Only a minority of the public reports low levels of trust in scientists.

We added the following sentence to the discussion section on page 32:

“Our global, 68-country survey challenges the idea that there is a widespread lack of public trust in scientists. In most countries, scientists and scientific methods are trusted. This finding is in line with other international studies on trust in scientists⁶⁻⁸. Our study thus confirms, expands and strengthens previous work that refutes the narrative of a wide-ranging crisis of trust. We expand previous studies by providing the largest dataset on trust in scientists post-pandemic and relying on a theoretically informed multidimensional trust measure.”